# 2d dualities from 4d

**Jiaqun Jiang[⋆], Satoshi Nawata[†] and Jiahao Zheng[‡]**

Department of Physics and Center for Field Theory & Particle Physics, Fudan University,
20005, Songhu Road, 200438 Shanghai, China

⋆ jiangjiaqun@gmail.com , † snawata@gmail.com , ‡ azjh1997@gmail.com

## Abstract

We find new $\mathcal{N} = (2, 2)$ and $\mathcal{N} = (0, 2)$ dualities through the twisted compactifications of 4d supersymmetric theories on $S^2$. Our findings include dualities for both $\mathcal{N} = (2, 2)$ and $\mathcal{N} = (0, 2)$ non-Abelian gauge theories, as well as $\mathcal{N} = (0, 2)$ Gauge/Landau-Ginzburg duality.

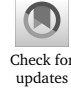

# 1 Introduction

Two-dimensional (2d) supersymmetric theories serve as simplified models that capture the essential features of broader quantum field theories. Additionally, many 2d supersymmetric theories exhibit exact solvability, allowing exact computation of physical observables such as correlation and partition functions. These exact results provide profound insights into the non-perturbative aspects of quantum field theory. They not only shed light on infrared physics but also reveal rich mathematical structures, including the geometry of the target space in non-linear sigma models and representations of infinite-dimensional symmetries.

The foundational groundwork for 2d $\mathcal{N} = (2,2)$ and $\mathcal{N} = (0,2)$ supersymmetric gauge theories was laid in [1]. Building on this foundation, the study of dualities in 2d supersymmetric theories has advanced significantly. In particular, $\mathcal{N} = (2,2)$ gauged linear sigma models have been instrumental in the study of Calabi-Yau sigma models and mirror symmetry. Most research has focused on Calabi-Yau manifolds in toric varieties, leading to the discovery of dualities and the understanding of phases in $\mathcal{N} = (2,2)$ Abelian gauge theories. Although progress on non-Abelian gauge theories has been limited, dualities for $\mathcal{N} = (2,2)$ non-Abelian gauge theories were proposed in [2,3]. These dualities are scrutinized from the viewpoint of exact partition functions (see [4] and references therein), and mathematics [5].

The $\mathcal{N} = (0,2)$ triality proposed in [6] is the first crucial progress along this line of investigation in 2d $\mathcal{N} = (0,2)$ gauge theories. Recently, a large class of dualities of $\mathcal{N} = (0,2)$ quiver gauge theories has been proposed in [7] through the twisted compactification of Lagrangian class $\mathcal{S}$ theories [8]. In this paper, we further explore this direction, finding new 2d $\mathcal{N} = (2,2)$ and $\mathcal{N} = (0,2)$ dualities from 4d supersymmetric theories.

The structure of this paper is as follows. In Section 2, we examine the twisted compactification of Lagrangian class $\mathcal{S}$ theories of type $A$, and generalize the construction to a wider family of 2d $\mathcal{N} = (2,2)$ quiver gauge theories. Through the computation of elliptic genera, we explicitly demonstrate that these theories are independent of the duality frame. In Section 3, we provide a detailed study of various 2d $\mathcal{N} = (0,2)$ dualities, beginning with a discussion on 2d (0,2) Seiberg-like dualities for SU gauge groups (§3.1). We then extend 2d (0,2) Seiberg-like dualities to new trialities in §3.2. Building on these results, we will uncover various new (0,2) dualities. First, we derive dualities between SU and Sp gauge theories in §3.3. The next two subsections (§3.4 and 3.5) systematically examine dualities in SU gauge theories with different types of chiral matter, including anti-symmetric, and symmetric representations. In §3.6, we investigate SO and Sp gauge theories with adjoint chiral matter and their dualities to free chiral theories. Finally, section 3.7 investigates theories with both symmetric and anti-symmetric chiral multiplets, providing a duality to a Landau-Ginzburg model. Various appendices are provided to supplement the main text.

In a sense, this paper serves as a sequel to [7]. Interested readers are encouraged to refer to [7] alongside this paper.

# 2 $\mathcal{N} = (2,2)$ dualities

In this section, we consider the twisted compactification of Lagrangian class $\mathcal{S}$ theories [8] on $S^2$, which leads to 2d $\mathcal{N} = (2,2)$ quiver theories. Specifically, we focus on a particular topological twist referred to as "flavored" reduction in [9].

We will perform a topological twist of the holonomy $U(1)_{S^2}$ of $S^2$ with a particular $U(1)_{\mathfrak{R}}$ $\mathfrak{R}$-symmetry of 4d $\mathcal{N} = 2$ SCFT. Treating the theory as a 4d $\mathcal{N} = 1$ theory, for the twisted compactification to be well-defined, the $U(1)_{\mathfrak{R}}$ charge $\mathfrak{r}$ of a 4d $\mathcal{N} = 1$ chiral multiplet must be integral [10]. Furthermore, if *all* charges are non-negative integers, we can focus on the

vanishing sector of the gauge magnetic flux on $S^2$ [9]. In this case, a 4d $\mathcal{N} = 1$ chiral multiplet with $U(1)_{\mathfrak{R}}$-charge $\mathfrak{r}$ becomes $(1 - \mathfrak{r})$ (0,2) chiral multiplets if $\mathfrak{r} < 1$, or $(\mathfrak{r} - 1)$ (0,2) Fermi multiplets if $\mathfrak{r} > 1$. However, the 4d chiral multiplet with $\mathfrak{r} = 1$ does not contribute to the 2d theory.

A 4d $\mathcal{N} = 2$ SCFT is endowed with $SU(2)_R \times U(1)_r$ $R$-symmetry. Here, we pick $\mathfrak{R} = 2R - f$ for the topological twist where $U(1)_R \subset SU(2)_R$ and $U(1)_f$ is a flavor symmetry that distinguishes the half-hypermultiplets $(q, \tilde{q})$. The symmetry analysis for supercharges and the fundamental fields is given in Tab. 1. Since the 4d $\mathcal{N} = 2$ supercharges are uncharged under the $U(1)_f$ flavor symmetry, the four supercharges $Q^1_-, Q^2_+, \tilde{Q}^1_+, \tilde{Q}^2_-$ are neutral under this twist. They have two opposite charges under the $U(1)_{T^2}$ rotation group of $\mathbb{R}^2$ (or $T^2$ in this paper) where the 2d theory lives, resulting in $\mathcal{N} = (2, 2)$ supersymmetry. Note that $U(1)_{-R+f/2}$ corresponds to the $U(1)_V$ vector $R$-symmetry in 2d while $U(1)_r$ is identified with the $U(1)_A$ axial $R$-symmetry in 2d.

The adjoint $\Phi$ in 4d $\mathcal{N} = 2$ vector multiplet has charge 0 under $U(1)_{2R-f}$ so that it becomes a 2d (0,2) adjoint chiral multiplet. Consequently, 4d $\mathcal{N} = 2$ vector multiplet gives rise to 2d $\mathcal{N} = (2, 2)$ vector multiplet. Under this twist, the half-hypermultiplets $(q, \tilde{q})$ transform into 2d (0,2) chiral and Fermi multiplet, forming a (2,2) chiral multiplet. Here, the $U(1)_f$ flavor symmetry plays an important role. In summary, under this twisted compactification, we obtain the following mapping

$$4\text{d } \mathcal{N} = 2 \text{ vector multiplet} \ \rightsquigarrow \ 2\text{d } \mathcal{N} = (2, 2) \text{ vector multiplet},$$
$$4\text{d } \mathcal{N} = 2 \text{ hypermultiplet} \ \rightsquigarrow \ 2\text{d } \mathcal{N} = (2, 2) \text{ chiral multiplet}.$$

In class $\mathcal{S}$ theories of type $A_{N-1}$, punctures are labeled by partitions of $N$, and these theories generally lack a Lagrangian description. To perform the (2,2) reduction, we focus on class $\mathcal{S}$ theories with Lagrangian descriptions. The fundamental building block, in this case, is a sphere with two maximal punctures and one minimal puncture, corresponding to $N^2$ hypermultiplets with a flavor symmetry group $U(N^2)$. This group contains a subgroup $SU(N)_a \times SU(N)_b \times U(1)_c$.

Table 1: Symmetries of 4d $\mathcal{N} = 2$ supercharges and fields. The 4d $\mathcal{N} = 1$ chiral fields $(q, \tilde{q})$ form an $\mathcal{N} = 2$ hypermultiplet, while $\Phi$ represents the $\mathcal{N} = 1$ adjoint chiral in an $\mathcal{N} = 2$ vector multiplet. The fifth column indicates the $U(1)_f$ flavor symmetry that differentiates between $q$ and $\tilde{q}$. The topological twist of $U(1)_{S^2}$ with $U(1)_{R+\frac{1}{2}(r-f)}$ gives rise to $\mathcal{N} = (0, 2)$ supersymmetry, whereas twisting with $U(1)_{2R-f}$ results in $\mathcal{N} = (2, 2)$ supersymmetry.

| | $SU(2)_1$ | $SU(2)_2$ | $SU(2)_R$ | $U(1)_r$ | $U(1)_f$ | $U(1)_{T^2}$ | $U(1)_{S^2}$ | $U(1)^{(0,2)}$ | $U(1)^{(2,2)}$ |
|---|---|---|---|---|---|---|---|---|---|
| $Q^1_-$ | $-\frac{1}{2}$ | 0 | $\frac{1}{2}$ | 1 | 0 | $-1$ | $-1$ | 0 | 0 |
| $Q^1_+$ | $\frac{1}{2}$ | 0 | $\frac{1}{2}$ | 1 | 0 | 1 | 1 | 2 | 2 |
| $Q^2_-$ | $-\frac{1}{2}$ | 0 | $-\frac{1}{2}$ | 1 | 0 | $-1$ | $-1$ | $-1$ | $-2$ |
| $Q^2_+$ | $\frac{1}{2}$ | 0 | $-\frac{1}{2}$ | 1 | 0 | 1 | 1 | 1 | 0 |
| $\tilde{Q}^1_-$ | 0 | $-\frac{1}{2}$ | $\frac{1}{2}$ | $-1$ | 0 | $-1$ | 1 | 1 | 2 |
| $\tilde{Q}^1_+$ | 0 | $\frac{1}{2}$ | $\frac{1}{2}$ | $-1$ | 0 | 1 | $-1$ | $-1$ | 0 |
| $\tilde{Q}^2_-$ | 0 | $-\frac{1}{2}$ | $-\frac{1}{2}$ | $-1$ | 0 | $-1$ | 1 | 0 | 0 |
| $\tilde{Q}^2_+$ | 0 | $\frac{1}{2}$ | $-\frac{1}{2}$ | $-1$ | 0 | 1 | $-1$ | $-2$ | $-2$ |
| $q$ | 0 | 0 | $\frac{1}{2}$ | 0 | 1 | 0 | 0 | 0 | 0 |
| $\tilde{q}$ | 0 | 0 | $\frac{1}{2}$ | 0 | $-1$ | 0 | 0 | 1 | 2 |
| $\Phi$ | 0 | 0 | 0 | 2 | 0 | 0 | 0 | 1 | 0 |

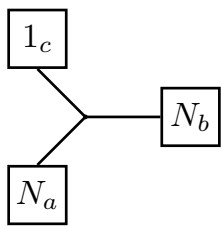

Figure 1: Basic building block $U_N^{(2,2)}$ for SU($N$) theory.

In the (2,2) reduction, $U(1)_c$ plays the role of the $U(1)_f$ introduced earlier, used in the topological twist. Under this twisted compactification, it reduces to $N^2$ $\mathcal{N} = (2,2)$ chiral multiplets. This reduced (2,2) theory, denoted $U_N^{(2,2)}$, serves as the basic building block for the $\mathcal{N} = (2,2)$ theories we consider, and its quiver is depicted in Fig. 1.

Using elliptic genera, we will study $\mathcal{N} = (2,2)$ quiver gauge theories constructed from the building block $U_N^{(2,2)}$. The minimal review of the elliptic genera is given in Appendix B. The contribution to the elliptic genus from the basic building block $U_N^{(2,2)}$, which consists of $N^2$ chiral multiplets, is

$$\mathcal{I}_{U_N}^{(2,2)}(\boldsymbol{a}, \boldsymbol{b}, c) = \prod_{i,j=1}^{N} \frac{\vartheta_1(y a_i b_j c)}{\vartheta_1(a_i b_j c)}, \tag{1}$$

where we impose the condition $\prod_{i=1}^{N} a_i = 1 = \prod_{i=1}^{N} b_i$ for the SU($N$) fugacities.

We construct $\mathcal{N} = (2,2)$ quiver gauge theories by gauging the flavor symmetries of the building blocks $U_N^{(2,2)}$. To gauge the SU($N$) flavor symmetries, we include the contribution from the SU($N$) vector multiplet in the Jeffrey-Kirwan integral of the elliptic genus

$$\mathcal{I}_{\text{SU}(N)}^{(2,2),\text{V}}(\zeta) = \frac{1}{N!} \left( \frac{\eta(q)^3}{\vartheta_1(y)} \right)^{N-1} \prod_{\substack{i,j=1 \\ i \neq j}}^{N} \frac{\vartheta_1(\zeta_i/\zeta_j)}{\vartheta_1(y \zeta_i/\zeta_j)}, \tag{2}$$

where the condition $\prod_{i=1}^{N} \zeta_i$ is imposed on the gauge fugacities. We do not introduce a superpotential in a quiver theory.

As in the case of $\mathcal{N} = (0,4)$ theories [7], the $\mathcal{N} = (2,2)$ elliptic genera for quiver types of genus greater than zero also turn out to be remarkably simple. First, consider the theory of genus one with one minimal puncture, which consists of the free chiral plus the SU($N$) gauge theory with adjoint chiral. The SU($N$) gauge theory with adjoint chiral is indeed the $\mathcal{N} = (4,4)$ vector multiplet, where the potential for the scalar fields is

$$V = \frac{1}{2e^2} \text{Tr}[\sigma, \sigma^\dagger]^2 + \frac{e^2}{2} \text{Tr}[\phi, \phi^\dagger]^2 + \frac{1}{2} \text{Tr}[\phi^\dagger, \sigma^\dagger][\sigma, \phi] + \frac{1}{2} \text{Tr}[\phi^\dagger, \sigma][\sigma^\dagger, \phi]. \tag{3}$$

Here, $\phi$ is the lowest component of the $\mathcal{N} = (2,2)$ adjoint chiral while $\sigma$ is the scalar in the $\mathcal{N} = (2,2)$ vector multiplet. Therefore, the moduli space is spanned by the mutually commuting eigenvalues of $\phi$ and $\sigma$ up to the action of the Weyl group, which leaves the maximal torus $U(1)^{N-1}$ of the gauge group unbroken. Consequently, the vacuum moduli space of the theory is

$$\mathbb{C} \times \frac{\mathfrak{t}_{\mathbb{C}} \times \mathfrak{t}_{\mathbb{C}}}{S_N}, \tag{4}$$

where $\mathfrak{t}_{\mathbb{C}}$ is the Cartan subalgebra of SL($N, \mathbb{C}$). The IR CFT is the orbifold CFT with this target, and therefore the central charge is

$$c_L = c_R = 3(N^2 - (N^2 - 1)) + 6(N - 1) = 3 + 6(N - 1). \tag{5}$$

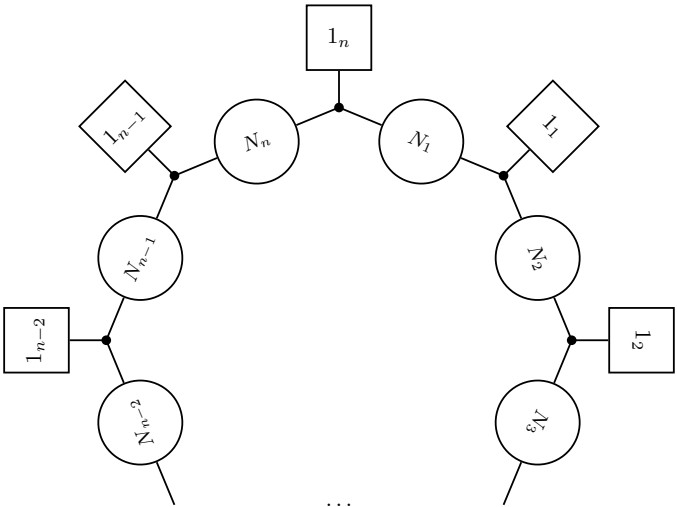

Figure 2: Quiver theory of genus one with $n$ punctures.

The first term accounts for the contributions from $N^2$ chiral multiplets and $N^2 - 1$ gauginos, while the second term arises from the unbroken gauge group $U(1)^{N-1}$. The elliptic genus of the theory is then expressed as[1]

$$
\begin{aligned}
\mathcal{I}^{(2,2),N}_{g=1,n=1} &= \oint_{\mathrm{JK}} \frac{d\boldsymbol{a}}{2\pi i \boldsymbol{a}} \mathcal{I}^{(2,2)}_{U_N}(\boldsymbol{a}, \boldsymbol{a}^{-1}, c) \mathcal{I}^{(2,2),\mathrm{V}}_{\mathrm{SU}(N)}(\boldsymbol{a}) \\
&= \frac{\vartheta_1(y^N c^N) \vartheta_1(y)}{\vartheta_1(c^N) \vartheta_1(y^N)} = \frac{\vartheta_1(yc)}{\vartheta_1(c)} \cdot \frac{\vartheta_1(y^N c^N) \vartheta_1(c) \vartheta_1(y)}{\vartheta_1(yc) \vartheta_1(c^N) \vartheta_1(y^N)}.
\end{aligned}
\tag{6}
$$

Similar to [7, Eqn. (3.37)], the first term represents the contribution from a free hypermultiplet, while the second term corresponds to the elliptic genus of an $\mathcal{N} = (4, 4)$ vector multiplet. The chiral ring of the symmetric product $(\mathfrak{t}_{\mathbb{C}} \times \mathfrak{t}_{\mathbb{C}})/S_N$ is generated by the operators $\mathrm{Tr}\!\left(\phi^i \sigma^j\right)$ for $i + j > 0$. The elliptic genus of the $\mathcal{N} = (4, 4)$ vector multiplet is expected to be expressed in terms of contributions from these chiral operators, along with Fermi fields that impose their relations. Remarkably, these contributions cancel out, resulting in the compact form of the elliptic genus given above. Currently, the authors do not have an explicit formulation of the relations among the generators or a detailed understanding of the cancellation mechanism. (For the SU(2) gauge group, such a chiral ring relation is given in [11, §2.2.3].)

Let us consider the quiver theory of genus one with $n$ punctures as illustrated in Fig. 2. The theory can be understood as the (2,2) reduction of the corresponding Lagrangian class $\mathcal{S}$ theory. At a generic point on the moduli space of chiral multiplets, the diagonal Cartan subgroup $U(1)^{N-1}$ is unbroken at the infra-red. Therefore, the corresponding twisted chiral field can take the vacuum expectation value. Therefore, the complex dimension of the vacuum moduli space is $2(N-1) + n$, and the central charge of the IR CFT is

$$
c_L = c_R = 3n(N^2 - (N^2 - 1)) + 6(N - 1) = 3n + 6(N - 1),
\tag{7}
$$

where the first term accounts for the contributions from $n$ sets of $N^2$ chiral multiplets and $n$ sets of $N^2 - 1$ gauginos, while the second term arises from the unbroken gauge group $U(1)^{N-1}$.

---

[1]Disclaimer: In this paper, we present numerous conjectural identities for elliptic genera, such as (6), expressed in terms of JK residue integrals. However, these identities are not rigorously proven. Instead, we verify them by explicitly computing JK residue integrals up to rank five and performing $q$-expansions up to $\mathcal{O}(q^2)$. Establishing formal proof of these identities remains an intriguing open problem.

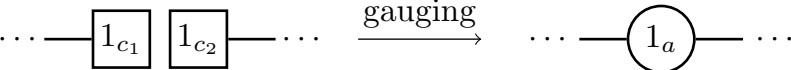

Figure 3: U(1) gauging.

Performing the JK residue integral, we verify the equality of the elliptic genus indeed takes the form of a product of (6):

$$\mathcal{I}^{(2,2),N}_{g=1,n}(c_1,\ldots,c_n) = \prod_{s=1}^{n} \frac{\vartheta_1(y^N c_s^N)\vartheta_1(y)}{\vartheta_1(c_s^N)\vartheta_1(y^N)}\,, \tag{8}$$

where $c_i$ are the U(1) flavor symmetries associated to the $n$ puncture. Thus, this indicates that the elliptic genera receive local contributions from minimal punctures for this class of $\mathcal{N}=(2,2)$ theories.

Unlike class $\mathcal{S}$ theories, we consider U(1) gauging in 2d $\mathcal{N}=(2,2)$ theories when gluing the building blocks $U_N$. However, gauging a U(1) node results in a theory that is no longer a (2,2) reduction of a class $\mathcal{S}$ theory. Nonetheless, as in Fig. 3, we can gauge the anti-diagonal part of U(1) while preserving the diagonal U(1) as a global symmetry.

At the level of the elliptic genus, the U(1) gauging procedure is given by

$$\mathcal{I}_{\mathcal{T}_1}(c_1,\ldots)\mathcal{I}_{\mathcal{T}_2}(c_2,\ldots) \quad \rightarrow \quad \frac{\eta(q)^3}{\vartheta_1(y)}\oint_{\mathrm{JK}} \frac{da}{2\pi i a}\mathcal{I}_{\mathcal{T}_1}(ad,\ldots)\mathcal{I}_{\mathcal{T}_2}(a^{-1}d,\ldots), \tag{9}$$

where $d$ is the diagonal U(1) flavor fugacity.

Let us consider the quiver theories of genus two, which involves the U(1) gauging. There are two types of quivers, as shown in Fig. 4. In the theory of genus one with one puncture, the Cartan subgroup $\mathrm{U}(1)^{N-1}$ is unbroken. For the genus-two theory, the Cartan subgroup $\mathrm{U}(1)^{2(N-1)}$ of the gauge group $\mathrm{SU}(N)\times\mathrm{SU}(N)$ is unbroken. Consequently, the corresponding twisted chiral fields can take the expectation value, and the central charge of the genus-two theory is

$$c_L = c_R = 3(2N^2 - 2(N^2-1)-1)+12(N-1) = 3+12(N-1). \tag{10}$$

Here, the first term accounts for the contributions from two sets of $N^2$ chiral multiplets and two sets of $N^2-1$ gauginos, along with an additional single U(1) gaugino. The second term originates from the unbroken gauge group $\mathrm{U}(1)^{2(N-1)}$. Since the two quiver descriptions have different Lagrangians, they have different expressions by JK residue integrals for the elliptic

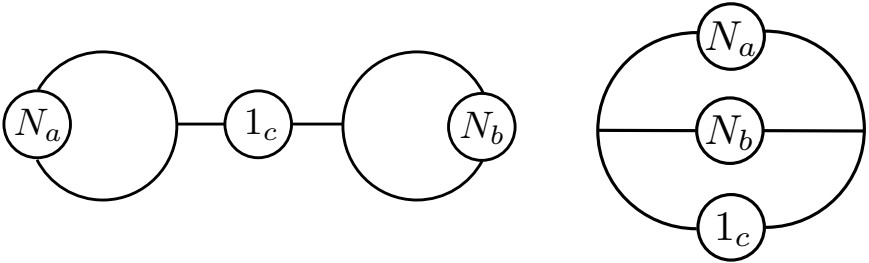

Figure 4: Genus two theories.

genera

$$\mathcal{I}^{(2,2),N}_{\bigcirc\!-\!\bigcirc} = \frac{\eta(q)^3}{\vartheta_1(y)} \int_{\mathrm{JK}} \frac{da}{2\pi i a} \frac{db}{2\pi i b} \frac{dc}{2\pi i c} \mathcal{I}^{(2,2)}_{U_N}(a,a^{-1};dc)\mathcal{I}^{(2,2)}_{U_N}(b,b^{-1};dc^{-1})\mathcal{I}^{(2,2),\mathrm{V}}_{\mathrm{SU}(N)}(a)\mathcal{I}^{(2,2),\mathrm{V}}_{\mathrm{SU}(N)}(b),$$

$$\mathcal{I}^{(2,2),N}_{\ominus} = \frac{\eta(q)^3}{\vartheta_1(y)} \int_{\mathrm{JK}} \frac{da}{2\pi i a} \frac{db}{2\pi i b} \frac{dc}{2\pi i c} \mathcal{I}^{(2,2)}_{U_N}(a,b;dc)\mathcal{I}^{(2,2)}_{U_N}(a^{-1},b^{-1};dc^{-1})\mathcal{I}^{(2,2),\mathrm{V}}_{\mathrm{SU}(N)}(a)\mathcal{I}^{(2,2),\mathrm{V}}_{\mathrm{SU}(N)}(b).$$

However, the explicit evaluation of the JK residue integrals verifies their agreement:

$$\begin{aligned}
\mathcal{I}^{(2,2),N}_{\bigcirc\!-\!\bigcirc} = \mathcal{I}^{(2,2),N}_{\ominus} &= \frac{\eta(q)^3\vartheta_1(y)}{\vartheta_1(y^N)^2} \oint_{\mathrm{JK}} \frac{da}{2\pi i a} \frac{\vartheta_1(y^N a^N d^N)}{\vartheta_1(a^N d^N)} \frac{\vartheta_1(y^N a^{-N} d^N)}{\vartheta_1(a^{-N} d^N)} \\
&= N \frac{\vartheta_1(y)\vartheta_1(y^N d^{2N})}{\vartheta_1(y^N)\vartheta_1(d^{2N})},
\end{aligned} \tag{11}$$

indicating that they are dual to each other. This suggests that the IR theory is independent of specific descriptions of genus-two theories, implying an underlying TQFT structure in this class of theories.

For a $(2,2)$ quiver theory of genus $g > 0$ constructed from the $U^{(2,2)}_2$ theory, the minimum number of U(1) gauge groups required is $g - 1$. Hence, extending the previous results, we can consider a $(2,2)$ quiver theory of genus $g > 0$ with $n$ punctures, where the numbers of SU($N$) and U(1) gauge groups are $2(g-1)+n$ and $g-1$, respectively. At a generic point in the moduli space of the chiral multiplets, we conjecture that the U(1)$^{g(N-1)}$ gauge group remains unbroken. Consequently, the central charge of the theory is:

$$c_L = 6g(N-1) + 3(n+g-1) = c_R. \tag{12}$$

For this class of theories, the elliptic genus depends only on $(g,n)$, and is independent of the quiver descriptions (or frames), which is expressed as

$$\mathcal{I}^{(2,2),N}_{g>0,n}(c_1,\ldots,c_n) = \mathcal{I}^{(2,2),N}_{g=1,n}(c_1,\ldots,c_n) \prod_{i=1}^{g-1} \frac{N\vartheta_1(y)\vartheta_1(y^N d_i^{2N})}{\vartheta_1(y^N)\vartheta_1(d_i^{2N})}, \tag{13}$$

where $\mathcal{I}^{(2,2),N}_{g=1,n}$ is given by (8). Thus, theories with different quiver descriptions are all dual to each other. Moreover, the integral formula (11) guarantees that the above form of the elliptic genera is consistent with the TQFT structure as

$$\begin{aligned}
\mathcal{I}^{(2,2),N}_{g=g_1+g_2,n_1+n_2-2} &= \frac{\eta(q)^3}{\vartheta_1(y)} \int_{\mathrm{JK}} \frac{da}{2\pi i a} \mathcal{I}^{(2,2),N}_{g_1,n_1}(\ldots,d_{g-1}a)\mathcal{I}^{(2,2),N}_{g_2,n_2}(d_{g-1}a^{-1},\ldots), \\
\mathcal{I}^{(2,2),N}_{g+1,n-2} &= \frac{\eta(q)^3}{\vartheta_1(y)} \int_{\mathrm{JK}} \frac{da}{2\pi i a} \mathcal{I}^{(2,2),N}_{g,n}(\ldots,d_g a, d_g a^{-1}).
\end{aligned} \tag{14}$$

As a result of JK residue integrals, these elliptic genera are all expressed as simple products of theta functions, suggesting that they are dual to $\mathcal{N} = (2,2)$ Landau-Ginzburg models. The challenge lies in identifying the correct superpotential. The method for this is given by using $S^2$ partition functions [12] based on the techniques of [13–15]. We leave this problem for future investigation.

Furthermore, one can introduce FI parameters for the U(1) gauge groups and discrete theta angles for SU($N$) gauge groups. It is also desirable to study the phases of the $\mathcal{N} = (2,2)$ quiver theories considered in this paper with respect to these parameters, following the approach of [1].

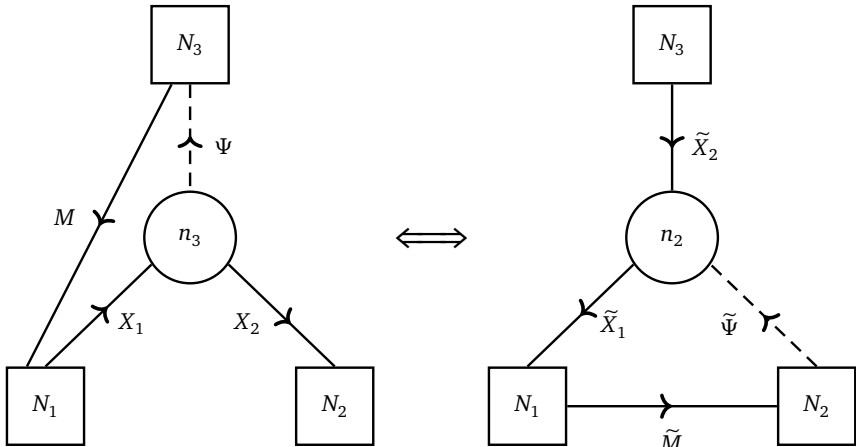

Figure 5 (SU-dual): 2d (0,2) duality for SQCDs with SU gauge groups, where $n_3 = \frac{N_1+N_2-N_3}{2}$ and $n_2 = \frac{N_1+N_3-N_2}{2}$ ($N_1 \geq N_2 + N_3$). The solid line represents a chiral multiplet while the dashed line represents a Fermi multiplet.

# 3 $\mathcal{N} = (0,2)$ dualities

In this section, we explore deeper into the rich landscape of 2d $\mathcal{N} = (0,2)$ dualities. 2d $\mathcal{N} = (0,2)$ supersymmetric theories are an important class of quantum field theories with chiral supersymmetry. They appear in contexts ranging from heterotic string worldsheet models to effective theories on branes. Despite having only two supercharges, techniques developed in recent years allow us to study their rich dynamics. In particular, understanding their infrared behavior and uncovering possible dualities remains a central topic of interest.

A foundational breakthrough in this direction was the discovery of the triality of 2d $\mathcal{N} = (0,2)$ U($N$) gauge theories [6], which demonstrated that three distinct unitary gauge theories become identical in the infra-red. This phenomenon closely parallels 4d $\mathcal{N} = 1$ Seiberg duality. Inspired by this example, more examples of 2d (0,2) dualities are found by reducing known 4d dual pairs on $S^2$ with topological twist [7,9,11,16,17]. In particular, Seiberg-like dualities for 2d $\mathcal{N} = (0,2)$ SU and Sp gauge theories were uncovered by compactifying their 4d $\mathcal{N} = 1$ counterparts on $S^2$ [9,17]. Similarly, 2d $\mathcal{N} = (0,2)$ gauge/Landau-Ginzburg (LG) dualities emerged from the twisted compactification of 4d $\mathcal{N} = 2$ SCFTs [7].

Motivated by these developments, we extend 2d $\mathcal{N} = (0,2)$ Seiberg-like dualities to novel trialities for Sp and SO gauge groups. Moreover, using these dualities (or trialities) as building blocks, we derive a broader class of 2d $\mathcal{N} = (0,2)$ dualities.

## 3.1 Comments on 2d (0,2) Seiberg-like dualities with SU gauge groups

Let us first review the duality of 2d $\mathcal{N} = (0,2)$ SU gauge theories obtained by the twisted compactification of the original 4d $\mathcal{N} = 1$ Seiberg duality [18] on $S^2$. This duality is illustrated in Fig. 5 (SU-dual).[2]

The two dual theories are described as follows:

- The first theory is an SU($n_3$) gauge theory with:

    - $N_1$ fundamental chiral multiplets $X_1$,

---

[2]This duality was originally discovered in [9], which we regrettably overlooked in the first version of our paper. In that version, we presented a special case of this duality in a manner that might have implied it was our original derivation. We would like to clarify that full credit for this discovery belongs to [9]. The purpose of this subsection is to elucidate key subtleties and provide additional insights into the duality.

- $N_2$ anti-fundamental chiral multiplets $X_2$,
- $N_3$ anti-fundamental Fermi multiplets $\Psi$,

where $\mathbb{Z} \ni n_3 = \frac{N_1+N_2-N_3}{2}$. Additionally, the theory contains a gauge-singlet meson $M$, which couples via the superpotential:

$$\mathcal{W} = \text{Tr}(\Psi M X_1). \tag{15}$$

- The second theory is an $SU(n_2)$ gauge theory with:

  - $N_1$ anti-fundamental chiral multiplets $\widetilde{X}_1$,
  - $N_2$ fundamental Fermi multiplets $\widetilde{\Psi}$,
  - $N_3$ fundamental chiral multiplets $\widetilde{X}_2$,

  where $\mathbb{Z} \ni n_2 = \frac{N_1+N_3-N_2}{2}$. This theory also contains a gauge-singlet meson $\widetilde{M}$, which couples via the superpotential:

$$\mathcal{W} = \text{Tr}\big(\widetilde{\Psi}\widetilde{X}_1\widetilde{M}\big). \tag{16}$$

For the duality to hold, we impose the condition:

$$N_1 \geq N_2 + N_3. \tag{17}$$

The charges of each multiplet under the global symmetry are summarized in the following table:

Table 2: Charges of each multiplet under global symmetries in the dual pair of theories depicted in Fig. 5 (SU-dual).

|  | $X_1$ | $X_2$ | $\Psi$ | $M$ | $\widetilde{X}_1$ | $\widetilde{X}_2$ | $\widetilde{\Psi}$ | $\widetilde{M}$ |
|---|---|---|---|---|---|---|---|---|
| $SU(N_1)$ | $\overline{\Box}$ | $\mathbf{1}$ | $\mathbf{1}$ | $\Box$ | $\Box$ | $\mathbf{1}$ | $\mathbf{1}$ | $\overline{\overline{\Box}}$ |
| $SU(N_2)$ | $\mathbf{1}$ | $\Box$ | $\mathbf{1}$ | $\mathbf{1}$ | $\mathbf{1}$ | $\mathbf{1}$ | $\overline{\Box}$ | $\mathbf{1}$ |
| $SU(N_3)$ | $\mathbf{1}$ | $\mathbf{1}$ | $\Box$ | $\overline{\Box}$ | $\mathbf{1}$ | $\overline{\Box}$ | $\mathbf{1}$ | $\Box$ |
| $U(1)_1$ | $-1$ | $0$ | $0$ | $1$ | $1-\frac{N_1}{n_2}$ | $\frac{N_1}{n_2}$ | $\frac{N_1}{n_2}$ | $-1$ |
| $U(1)_2$ | $0$ | $1$ | $0$ | $0$ | $0$ | $0$ | $-1$ | $0$ |
| $U(1)_3$ | $0$ | $0$ | $1$ | $-1$ | $0$ | $-1$ | $0$ | $1$ |

**Why the triality fails with SU gauge groups:** However, unlike in the case of the unitary gauge group, this duality does not extend to a triality. The failure of triality when transitioning from $U(N_c)$ to $SU(N_c)$ can be attributed to the absence of a Fayet-Iliopoulos (FI) term, as explained below.

The 2d $\mathcal{N} = (0, 2)$ triality [6] is an equivalence of three unitary gauge theories in the infrared. This triality can be understood through the relationship among the target geometry of the non-linear sigma models in the infrared [19, 20]. The target geometry involves the tautological bundle $S$ of rank $k$ and the quotient bundle $Q$ of rank $(n-k)$ on a Grassmannian $\text{Gr}(k, n)$, related by the short exact sequence:

$$0 \longrightarrow S \longrightarrow \mathcal{O}^n \longrightarrow Q \longrightarrow 0.$$

For the 2d $\mathcal{N} = (0, 2)$ triality [6], an essential bundle isomorphism plays a key role:

$$
\begin{array}{ccc}
S & & Q^* \\
\downarrow & \cong & \downarrow \\
\text{Gr}(k, n) & & \text{Gr}(n-k, n).
\end{array}
\tag{18}
$$

This isomorphism reflects the interchange between the tautological bundle $S$ and the dual of the quotient bundle $Q^*$ under the duality transformation $\text{Gr}(k, n) \longleftrightarrow \text{Gr}(n-k, n)$.

For unitary gauge groups, one can introduce the Fayet-Iliopoulos (FI) term $\zeta$. For a positive FI term ($\zeta > 0$), one of the triality theories flows to the non-linear sigma model with the target space

$$
\begin{array}{ccc}
S^{\oplus N_3} \oplus Q^{\oplus N_2} & & S^{*\oplus N_2} \oplus Q^{*\oplus N_3} \\
\downarrow & \cong & \downarrow \\
\text{Gr}(n_3, N_1) & & \text{Gr}(n_2, N_1) ,
\end{array}
\tag{19}
$$

where $n_i = \frac{-N_i + N_{i+1} + N_{i+2}}{2}$ (indices are considered mod 3). Conversely, for a negative FI term ($\zeta < 0$), the target space configuration is:

$$
\begin{array}{ccc}
S^{*\oplus N_3} \oplus Q^{*\oplus N_1} & & S^{\oplus N_1} \oplus Q^{\oplus N_3} \\
\downarrow & \cong & \downarrow \\
\text{Gr}(n_3, N_2) & & \text{Gr}(n_1, N_2) .
\end{array}
\tag{20}
$$

Permuting the indices $(N_1, N_2, N_3)$, we see that $(0, 2)$ triality manifests as equivalence among these target spaces. We refer to [20, Fig. 5] for a detailed illustration.

However, the triality does *not* hold, namely, the three theories are *inequivalent* in the infrared if we use a special unitary gauge group $\text{SU}(N_c)$ instead of $\text{U}(N_c)$, removing Fermi fields in the determinant representation of the unitary gauge group. In fact, elliptic genera distinguish the three theories if the gauge groups are special unitary groups $\text{SU}(N_c)$.

One key obstruction is that the condition (17) for the duality in Fig. 5 (SU-dual) is incompatible with the triality condition $N_i + N_{i+1} \geq N_{i+2}$ [6]. Furthermore, the FI term cannot be introduced for an $\text{SU}(N_c)$ gauge group. Hence, the two geometric configurations described by (19) and (20) cannot be connected through an $\text{SU}(N_c)$ gauge theory. In fact, this underpins why the triality fails when naively changing from $\text{U}(N_c)$ to $\text{SU}(N_c)$.

**Non-linear sigma model:**  With the SU gauge groups, the dual pair in Fig. 5 (SU-dual) flows to the non-linear sigma model with the same target manifold

$$
\begin{array}{ccc}
\det(S) \oplus S^{\oplus N_2} \oplus Q^{\oplus N_3} & & \det(Q^*) \oplus Q^{*\oplus N_2} \oplus S^{*\oplus N_3} \\
\downarrow & \cong & \downarrow \\
\text{Gr}(n_3, N_1) & & \text{Gr}(n_2, N_1) .
\end{array}
\tag{21}
$$

In fact, the condition (17) ensures that the base Grassmannian is defined as the moduli space of subspaces in $\mathbb{C}^{N_1}$. Note that, due to the SU gauge group, the $N_1$ (anti-)fundamental chirals parametrize

$$
\begin{array}{ccc}
\det(S) & & \det(Q^*) \\
\downarrow & \cong & \downarrow \\
\text{Gr}(n_3, N_1) & & \text{Gr}(n_2, N_1) .
\end{array}
\tag{22}
$$

where $\det(S)$ and $\det(Q^*)$ denote the determinant line bundles of $S$ and $Q^*$, respectively. Thus, the proposed duality for SU gauge groups can be understood as the equivalence of the target geometries in (21).

**Central charges:**  Since the target space is non-compact, one of the fundamental assumptions of $c$-extremization in [21, 22] is violated. Specifically, the issue arises because a non-holomorphic current may exist for the flavor symmetry associated with the non-compact direction. This current cannot mix with the $R$-symmetry current, which invalidates the naive application of $c$-extremization [17, 22].

To address this issue, it is necessary to identify the chiral operators that parametrize these non-compact directions and assign them an $R$-charge of 0. In our example, while the Grassmannian itself is compact, the total space (22) of the determinant line bundle, parametrized by $N_1$ (anti-)fundamental chirals, is non-compact. Since these $N_1$ (anti-)fundamental chirals transform with the same charge under the $U(1)_1$ flavor symmetry, their $R$-charges must all be set to zero.

Consequently, in both dual theories, the $R$-charges of all chiral fields must be uniformly set to 0. Additionally, the $R$-charge of the Fermi field is fixed at 1 to ensure that the superpotential retains an $R$-charge of 1 [17, 22].

With this assignment, we verify the agreement of the central charges for the dual theories:

$$c_L = 2(n_3^2 + N_1 N_3 + 1), \qquad c_R = 3(n_3^2 + N_1 N_3 + 1). \tag{23}$$

Note that the right-moving central charge is three times the dimension of the target space (21).

Let us compare this with the case of $\mathcal{N} = (0, 2)$ triality studied in [6]. For unitary gauge groups, the (anti-)fundamental chiral multiplets in the theory simply parameterize a compact Grassmannian manifold (without the determinant line bundle in this case). In addition, the presence of a superpotential imposes further constraints on the $R$-charges of the supermultiplets. Consequently, the $c$-extremization must be performed to determine the $R$-symmetry. Hence, although the vacuum moduli space is non-compact as seen in (19) and (20), the application of the $c$-extremization is still required.

**Other checks for the duality:** It is also straightforward to check that 't Hooft anomaly matches under the duality for each global symmetry. In addition, the elliptic genera can be computed using the JK residue integral:

$$\mathcal{I}_A = \frac{1}{n_3!} \oint_{\text{JK}} \frac{d\boldsymbol{a}}{2\pi i \boldsymbol{a}} \prod_{i=1}^{n_3} \frac{\prod_{j \neq i} \vartheta_1(a_i a_j^{-1}) \cdot \prod_{j=1}^{N_3} \vartheta_1(a_i^{-1} d_j x_3)}{\prod_{j=1}^{N_1} \vartheta_1(a_i b_j^{-1} x_1^{-1}) \cdot \prod_{j=1}^{N_2} \vartheta_1(a_i^{-1} c_j x_2)} \frac{\eta(q)^{n_3^2 + 3n_3 + N_1 N_3 - 2}}{\prod_{i=1}^{N_1} \prod_{j=1}^{N_3} \vartheta_1(x_1 x_3^{-1} b_i d_j^{-1})},$$

$$\mathcal{I}_B = \frac{1}{n_2!} \oint_{\text{JK}} \frac{d\boldsymbol{a}}{2\pi i \boldsymbol{a}} \prod_{i=1}^{n_2} \frac{\prod_{j \neq i} \vartheta_1(a_i a_j^{-1}) \cdot \prod_{j=1}^{N_2} \vartheta_1(a_i c_j^{-1} x_2^{-1} x_1^{\frac{N_1}{n_2}})}{\prod_{j=1}^{N_1} \vartheta_1(a_i^{-1} b_j x_1^{-\frac{n_3}{n_2}}) \cdot \prod_{j=1}^{N_3} \vartheta_1(a_i d_j^{-1} x_3^{-1} x_1^{\frac{N_1}{n_2}})} \frac{\eta(q)^{n_2^2 + 3n_2 + N_1 N_2 - 2}}{\prod_{i=1}^{N_1} \prod_{j=1}^{N_2} \vartheta_1(x_2 x_1^{-1} c_j b_i^{-1})}.$$

We verify the identity $\mathcal{I}_A = \mathcal{I}_B$ up to the rank-five integral. Notably, when $N_1 = N_2 + N_3$, an additional condition is required:

$$x_1^{N_1} = x_2^{N_2} x_3^{N_3}. \tag{24}$$

## 3.2  From 2d (0,2) Seiberg-like dualities to new trialities

In addition to SU gauge groups, the twisted compactification of 4d $\mathcal{N} = 1$ dualities with Sp gauge groups [23] on $S^2$ was first explored in [9], and later refined in [17]. This leads to the 2d (0,2) duality between Sp SQCD and a Landau-Ginzburg theory. Following their approach, we will find a new 2d (0,2) duality from the twisted compactification of the Intriligator-Seiberg duality [24] for the SO gauge group on $S^2$. Furthermore, in this subsection, we extend these 2d (0,2) Seiberg-like dualities to new trialities.

**Sp gauge group:** The twisted compactification on $S^2$ is applied to 4d $\mathcal{N} = 1$ dualities with Sp$(2N)$ gauge groups [23], leading to the identification of a 2d $\mathcal{N} = (0, 2)$ duality between Sp$(2N)$ SQCD (AS-1) and an LG model (AS-3) [9, 17]. In this work, we further generalize this result and propose a new triality, which we will outline below.

Before proceeding, let us remark on an important distinction between the 2d $\mathcal{N} = (0, 2)$ Seiberg-like dualities involving the SU and Sp gauge groups. A key difference is that Sp gauge

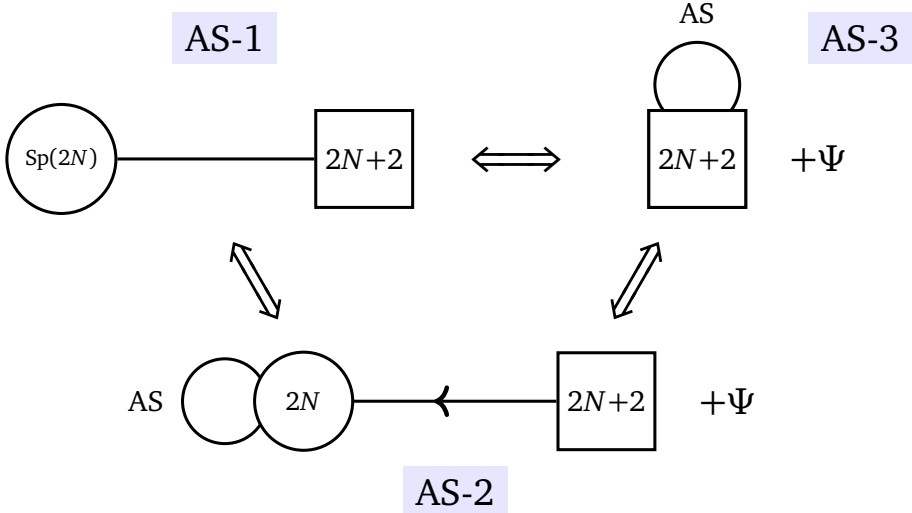

Figure 6 (AS-dual): A triality among three distinct theories: (AS-1) an Sp($2N$) gauge theory with $2N + 2$ fundamental chirals, (AS-2) an SU($2N$) gauge theory with one anti-symmetric and $2N + 2$ fundamental chirals, coupled to a neutral Fermi multiplet $\Psi$, and (AS-3) a Landau-Ginzburg model consisting of $(N + 1)(2N + 1)$ chiral multiplets coupled to a Fermi multiplet $\Psi$.

theories only have chiral multiplets in the fundamental representation, whereas SU gauge theories involve both fundamental and anti-fundamental chirals in their duality. Consequently, upon the compactification on $S^2$, we obtain a duality between a 2d $\mathcal{N} = (0, 2)$ Sp($2N$) gauge theory and a Landau-Ginzburg (LG) model. However, to our knowledge, a duality between two 2d $\mathcal{N} = (0, 2)$ Sp($2N$) gauge theories has not yet been discovered.

AS-1. Sp($2N$) gauge theory with $2N + 2$ fundamental chirals $Z$ and no superpotential.

AS-2. SU($2N$) gauge theory with one anti-symmetric chiral $X$ and $2N + 2$ fundamental chirals $Y$. Additionally, there is a neutral Fermi multiplet $\Psi$, forming a superpotential

$$\mathcal{W} = \Psi \operatorname{Pf} X. \tag{25}$$

AS-3. LG model of one Fermi $\Psi$ and $(N + 1)(2N + 1)$ chirals, forming an anti-symmetric $(2N + 2) \times (2N + 2)$ matrix $A$ with a superpotential

$$\mathcal{W} = \Psi \operatorname{Pf} A. \tag{26}$$

The U(1)$_x$ charges of the fields are given as follows:

$$
\begin{array}{c|ccccc}
 & \Psi & A & Z & X & Y \\
\hline
\text{U}(1)_x & -(2N+2) & 2 & 1 & \frac{2N+2}{N} & \frac{1}{N}
\end{array}
. \tag{27}
$$

As a simple check, their central charges agree as

$$c_L = 2N(2N + 3), \qquad c_R = 3N(2N + 3). \tag{28}$$

The elliptic genera also agree

$$
\begin{aligned}
\mathcal{I} &= -\frac{\eta(q)^{2N(N+3)}}{2^N N!} \oint_{\mathrm{JK}} \frac{d\boldsymbol{a}}{2\pi i \boldsymbol{a}} \prod_{i=1}^{N} \frac{\vartheta_1(a_i^{\pm 2}) \prod_{j>i} \vartheta_1(a_j^{\pm} a_i^{\pm})}{\prod_{j=1}^{2N+2} \vartheta_1(x a_i^{\pm} b_j)} \\
&= \frac{\eta(q)^{(2N^2+9N-3)} \vartheta_1(x^{-(2N+2)})}{(2N)!} \oint_{\mathrm{JK}} \frac{d\boldsymbol{a}}{2\pi i \boldsymbol{a}} \prod_{i=1}^{2N} \frac{\prod_{i\neq j} \vartheta_1(a_i/a_j)}{\prod_{i<j} \vartheta_1(x^{\frac{2N+2}{N}} a_i a_j) \prod_{j=1}^{2N+2} \vartheta_1(x^{\frac{1}{N}} a_i b_j)} \\
&= \frac{\eta(q)^{N(2N+3)} \vartheta_1(x^{-(2N+2)})}{\prod_{i<j} \vartheta_1(x^2 b_i b_j)} .
\end{aligned}
\tag{29}
$$

We find this triality by considering the (0,2) reduction of 4d $\mathcal{N}=2$ SCFTs [7,9] on $S^2$. For a 4d $\mathcal{N}=2$ theory with gauge group $G$, the $\beta$-function is given by

$$
\beta \propto 2T(\mathbf{adj}) - \sum_i T(R_i),
\tag{30}
$$

where $T(R)$ denotes the Dynkin indices of a representation $R$ for a half-hypermultiplet. On the other hand, the gauge anomaly of the corresponding (0,2) theory is expressed as

$$
\mathrm{Tr}(\gamma_3 G^2) = \sum_i T(R_i) - T(\mathbf{adj}).
\tag{31}
$$

Since only half of the 4d half-hypermultiplets become (0,2) chiral multiplets upon reduction (see Tab. 1), the condition for the vanishing $\beta$-function in 4d $\mathcal{N}=2$ theory is equivalent to the condition for the vanishing 2d gauge anomaly. Therefore, the (0,2) theory obtained from the Lagrangian 4d $\mathcal{N}=2$ SCFT is well-defined. In [7], the (0,2) reduction of 4d $\mathcal{N}=2$ Lagrangian SCFTs of type $A$ class $\mathcal{S}$ is considered. The elliptic genera of these theories can be expressed in a surprisingly simple form, involving products of theta functions and eta functions. In some cases, the LG dual theories are identified.

Here, we consider the 4d $\mathcal{N}=2$ SCFTs: $\mathrm{Sp}(2N)$ gauge theory with $2N+2$ fundamental half-hypermultiplets, and $\mathrm{SU}(2N)$ gauge theory with one anti-symmetric and $2N+2$ fundamental hypermultiplets. These theories arise from D4-NS5-O6$^-$ brane system (See [25, §3.1]). Their (0,2) reductions yield the AS-1 and AS-2 theories shown in Fig. 6 (AS-dual), and they turn out to be dual to a (0,2) LG model (AS-3). Notably, the integrand of the elliptic genus (29) coincides with that of the corresponding 4d $\mathcal{N}=2$ Schur indices (up to the modification of the U(1)$_x$-charges due to the superpotential (25)) [7,9].

Additionally, we find a similar triality with the flavor group $\mathrm{SU}(N)$ with odd $N$, which is given in Appendix C.2. As another generalization, we also propose a duality between the following theories (Fig. 7).

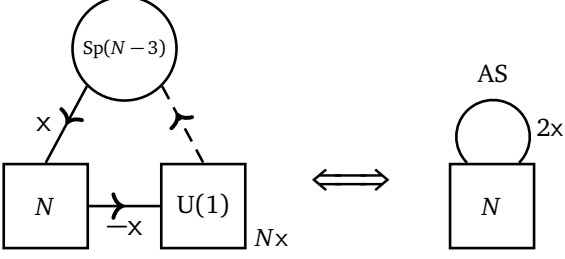

Figure 7: A duality between a free theory with $N(N-1)/2$ chirals and $\mathrm{Sp}(N-3)$ gauge theory with $N$ fundamental chirals, one fundamental Fermi multiplet and $N$ neutral mesons.

- $\mathrm{Sp}(N-3)$ gauge theory with $N$ fundamental chirals $X$, one fundamental Fermi multiplet $\Psi$ and $N$ neutral mesons $M$ with a superpotential

$$\mathcal{W} = \omega^{\alpha\beta} \Psi_\alpha X_{\beta,i} M^i, \tag{32}$$

where $\omega^{\alpha\beta}$ is the symplectic form that projects the tensor product $\square \otimes \square$ to the trivial representation for representations of the gauge group $\mathrm{Sp}(N-3)$.

- A free theory with $N(N-1)/2$ chirals.

This duality is used frequently in what follows. The elliptic genera are given by

$$
\mathcal{I} = \frac{\eta(q)^{(N^2+2N-9)/2}}{2^{(N-3)/2}\frac{N-3}{2}!} \oint_{\mathrm{JK}} \frac{d\boldsymbol{a}}{2\pi i \boldsymbol{a}} \prod_{i=1}^{\frac{N-3}{2}} \frac{\vartheta_1(a_i^\pm x^{-N})\vartheta_1(a_i^{\pm 2})\prod_{j<i}\vartheta_1(a_i^\pm a_j^\pm)}{\prod_{j=1}^N \vartheta_1(x a_i^\pm b_j)} \cdot \frac{1}{\prod_{i=1}^N \vartheta_1(x^{N-1} b_i^{-1})}
$$
$$
= \prod_{i<j} \frac{\eta(q)}{\vartheta_1(x^2 b_i b_j)}. \tag{33}
$$

This duality is used to derive other dualities in what follows.

**SO gauge group:** We also consider the twisted compactification of the 4d $\mathcal{N} = 1$ Intriligator-Seiberg duality [24] on $S^2$, which results in a duality between a 2d $\mathcal{N} = (0,2)$ SO SQCD (Sym-1) and an LG model (Sym-3). This procedure is detailed in Appendix C.1. Like in the case of the Sp gauge group, the compactification of the 4d $\mathcal{N} = 1$ duality on $S^2$ does not produce a duality between two SO gauge theories, as it involves only fundamental chiral multiplets. Moreover, we extend this duality to a triality, as illustrated in Fig. 8 (Sym-dual).

Sym-1. $\mathrm{SO}(N)$ gauge theory with $N-2$ fundamental chirals $Y$. Additionally, there is a gauge neutral chiral $X$ and Fermi $\Psi$, forming a superpotential

$$\mathcal{W} = \Psi(X^2 + \det A), \tag{34}$$

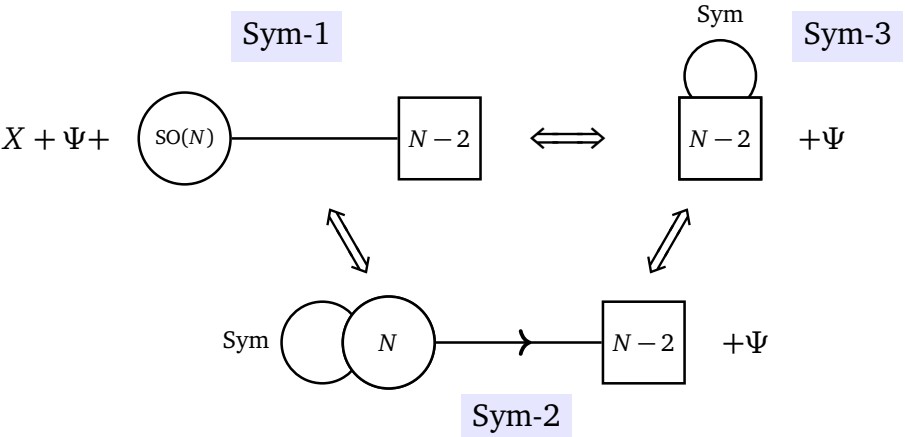

Figure 8 (Sym-dual): A triality among three distinct theories: (Sym-1) an $\mathrm{SO}(N)$ gauge theory of SQCD type with $N-2$ fundamental chirals coupled to a neutral chiral $X$ and Fermi $\Psi$, (Sym-2) an $\mathrm{SU}(N)$ gauge theory with one symmetric and $N-2$ fundamental chirals, coupled to a neutral Fermi multiplet $\Psi$, and (Sym-3) a Landau-Ginzburg model consisting of $(N-2)(N-1)/2$ chiral multiplets coupled to a Fermi multiplet $\Psi$.

where $A$ is defined by the projector $\eta^{\alpha\beta}$ of the tensor product $\square \otimes \square$ to the trivial representation for SO($N$) gauge group as

$$A_{ij} := \eta^{\alpha\beta} Y_{\alpha,i} Y_{\beta,j}. \tag{35}$$

Sym-2. SU($N$) gauge theory with one symmetric chiral $Z$ and $N-2$ fundamental chirals $W$. Furthermore, there is a neutral Fermi multiplet $\Psi$, which forms a superpotential

$$\mathcal{W} = \Psi \det A, \tag{36}$$

where

$$A_{ij} = Z_{\alpha\beta} W_i^\alpha W_j^\beta. \tag{37}$$

Sym-3. LG model of one Fermi $\Psi$ and $(N-2)(N-1)/2$ chirals, forming a symmetric $(N-2) \times (N-2)$ matrix $A$ with a superpotential

$$\mathcal{W} = \Psi \det A. \tag{38}$$

The U(1)$_x$ charges of the fields are given as follows:

$$
\begin{array}{c|cccccc}
 & \Psi & A & X & Y & Z & W \\
\hline
\text{U(1)}_x & -2(N-2) & 2 & N-2 & 1 & \frac{2(N-2)}{N} & \frac{2}{N}
\end{array}. \tag{39}
$$

These theories have the same central charges

$$c_L = N(N-3), \qquad c_R = \frac{3N(N-3)}{2}. \tag{40}$$

Furthermore, the elliptic genera agree

$$
\begin{aligned}
\mathcal{I} &= \frac{2\eta(q)^{(N^2-3\chi)/2} \vartheta_1(x^{-2(N-2)})}{2^{\lfloor (N-1)/2 \rfloor} \lfloor N/2 \rfloor! \, \vartheta_1(x^{N-2})} \oint_{\text{JK}} \frac{d\boldsymbol{a}}{2\pi i \boldsymbol{a}} \frac{\prod_{i<j}^{\lfloor N/2 \rfloor} \vartheta_1(a_i^\pm a_j^\pm)}{\prod_{i=1}^{\lfloor N/2 \rfloor} \prod_{j=1}^{N-2} \vartheta_1(x a_i^\pm b_j)} \left( \frac{\prod_{i=1}^{\lfloor N/2 \rfloor} \vartheta_1(a_i^\pm)}{\prod_{j=1}^{N-2} \vartheta_1(x b_j)} \right)^\chi \\
&= -\frac{\eta(q)^{\frac{1}{2}(N^2+3N-6)} \vartheta_1(x^{-2(N-2)})}{N!} \oint_{\text{JK}} \frac{d\boldsymbol{a}}{2\pi i \boldsymbol{a}} \prod_{i=1}^{N} \frac{\prod_{i\neq j} \vartheta_1(a_i/a_j)}{\prod_{i\leq j} \vartheta_1(x^{\frac{2(N-2)}{N}} a_i a_j) \prod_{j=1}^{N-2} \vartheta_1(x^{\frac{2}{N}} b_j/a_i)} \\
&= \frac{\eta(q)^{N(N-3)/2} \vartheta_1(x^{-2(N-2)})}{\prod_{i\leq j} \vartheta_1(x^2 b_i b_j)},
\end{aligned} \tag{41}
$$

where $\chi \equiv N \mod 2$.

In fact, we arrive at this triality by considering the (0,2) reduction of the 4d $\mathcal{N} = 2$ SCFTs: SO($2N$) gauge theory with $2N-2$ fundamental half-hypermultiplets, and SU($2N$) gauge theory with one symmetric and $2N-2$ fundamental hypermultiplets. These theories arise from D4-NS5-O6$^+$ brane system (See [25, §3.1]). Their (0,2) reductions yield the Sym-1 and Sym-2 theories shown in Fig. 8 (Sym-dual), and they turn out to be dual to a (0,2) LG model (Sym-3).

## 3.3 Duality between SU and Sp gauge theory

It is natural to ask whether the duality between the Sp and SU gauge theories (i.e. AS-1 and AS-2) in Fig. 6 (AS-dual) can be generalized to include the fundamental Fermi multiplets. To explore this, adding anti-fundamental Fermi multiplets to the AS-2 theory, we consider the following theory:

- SU($N_1 - N_2 - 2$) gauge theory with one anti-symmetric chiral $X$ (U(1)$_x$-charge 2), $N_1$ fundamental chirals $Y$ and $N_2$ anti-fundamental Fermi's $\Gamma$ where $N_1 - N_2$ is even. In addition, there is one free Fermi multiplet $\Psi$ (U(1)$_x$-charge $N = N_1 - N_2 - 2$), forming a superpotential

$$\mathcal{W} = \Psi \ \mathrm{Pf} X \, . \tag{42}$$

The elliptic genus of the theory is

$$\mathcal{I}_{\mathrm{SU}} = \frac{\vartheta_1(x^N)\eta(q)^{(N^2+9N-6)/2}}{N!} \oint_{\mathrm{JK}} \frac{d\boldsymbol{a}}{2\pi i \boldsymbol{a}} \prod_{i=1}^{N} \frac{\prod_{j\neq i} \vartheta_1(a_i/a_j) \cdot \prod_{j=1}^{N_2} \vartheta_1(a_i^{-1} c_j y_2)}{\prod_{j<i} \vartheta_1(x^2 a_i a_j) \prod_{j=1}^{N_1} \vartheta_1(a_i b_j y_1^{-1})} \, . \tag{43}$$

We aim to establish a duality with an Sp gauge theory by adding anti-fundamental Fermi multiplets to the AS-1 theory. (See the right panel of Fig. 10.) However, as we shall see, the resulting Sp gauge theory lacks the U(1)$_x$ symmetry. Therefore, we perform a sequence of duality transformations starting from the theory described above (illustrated in Fig. 9) and then apply a Higgsing procedure to eliminate the U(1)$_x$ symmetry. Let us explain this process step by step.

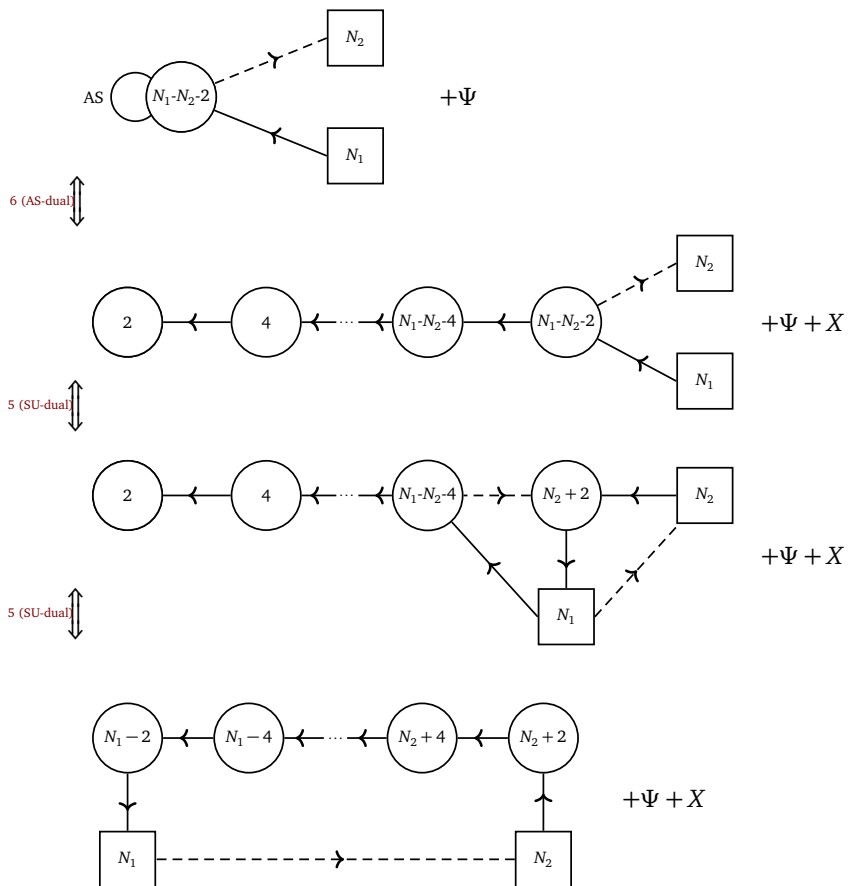

Figure 9: A chain of dualities. Starting with the first theory (top), we apply AS-dual between AS-2 and AS-3 repeatedly to obtain the quiver gauge theory shown in the second line. Next, we apply SU-dual to the SU($N_1 - N_2 - 2$) gauge group in the second theory, resulting in the third theory. Finally, successive applications of SU-dual to the remaining SU gauge groups, proceeding from right to left, yield the fourth theory at the bottom. The additional Fermi and chiral fields $\Psi$ and $X$ are included as indicated throughout the chain, and they form a superpotential term $\mathcal{W} = \Psi X$.

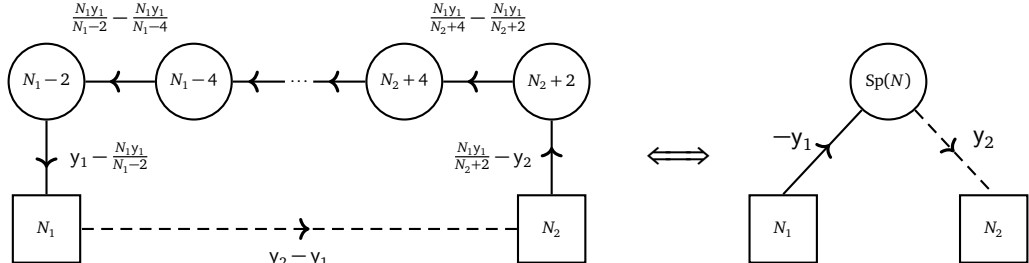

Figure 10: A duality between the quiver gauge theory with SU gauge groups and $Sp(N_1 - N_2 - 2)$ gauge theory. The labels with a linear combination of $y_1$ and $y_2$ represent U(1) flavor chemical potentials for the corresponding bifundamental fields.

We begin by recursively applying the duality between the AS-2 and AS-3 theories in Fig. 6 (AS-dual). Specifically, we apply this duality to the SU gauge node that contains a single anti-symmetric chiral multiplet, as demonstrated in [17, 26]. This transformation yields a linear quiver theory with SU gauge groups, as shown in the second line of Fig. 9. It is important to note that for SU(2), the anti-symmetric representation is trivial. Consequently, the dual description includes an additional chiral multiplet $X$ and a superpotential term $\mathcal{W} = \Psi X$, where $\Psi$ is a Fermi multiplet. In this intermediate theory, the bifundamental chiral multiplets between gauge groups, as well as $\Psi$ and $X$, are charged under the $U(1)_x$ symmetry.

Next, we recursively apply the SU Seiberg-like duality from Fig. 5 (SU-dual) to each gauge node, proceeding from right to left. This sequence of dualities ultimately produces the theory shown at the bottom of Fig. 9, where all the theories in Fig. 9 are dual to each other.

The final theory is characterized by the superpotential

$$\mathcal{W} = \Psi X + \mathrm{Tr}\left[ \Gamma\, Y^{(1)} \cdots Y^{\left(\frac{N_1 - N_2}{2}\right)} \right], \tag{44}$$

where $\Gamma$ denotes the Fermi meson, and each $Y^{(i)}$ is a bifundamental chiral multiplet transforming under the $SU(N_2 + 2i) \times SU(N_2 + 2i - 2)$ gauge or flavor groups for $i = 1, \ldots, \frac{N_1 - N_2}{2}$. These fields are charged under a linear combination of the $U(1)_{y_1}$ and $U(1)_{y_2}$ global symmetries. At this point, the $U(1)_x$ symmetry can be removed by giving the vacuum expectation value $\langle X \rangle$, effectively decoupling the chiral multiplet $X$ and the Fermi multiplet $\Psi$. This procedure completes the elimination of the $U(1)_x$ symmetry and ensures that the global symmetry structure of the resulting theory matches that of the target Sp gauge theory.

As a result, we propose a duality between the Sp gauge theory and the quiver gauge theory with SU gauge nodes obtained through this sequence of dualities and Higgsing, as illustrated in Fig. 10. The specifics of these dual theories are detailed below:

- The quiver theory with SU gauge groups features a superpotential:

$$\mathcal{W} = \mathrm{Tr}\, \Gamma Y^{(1)} \cdots Y^{\left(\frac{N_1 - N_2}{2}\right)}, \tag{45}$$

  where $\Gamma$ is the Fermi meson, and $Y^{(i)}$ represents the bifundamental chiral fields between the $SU(N_2 + 2i)$ and $SU(N_2 + 2i - 2)$ gauge/flavor groups ($i = 1, \ldots, \frac{N_1 - N_2}{2}$). The elliptic

genus of the theory is

$$\mathcal{I}_{\text{quiver}} = \prod_{k=1}^{\frac{N}{2}} \frac{1}{(N_1 - 2k)!} \int_{\text{JK}} \frac{d\boldsymbol{a}}{2\pi i \boldsymbol{a}} \prod_{k=1}^{\frac{N}{2}} \frac{\prod_{i \neq j}^{N_1 - 2k} \vartheta_1\left(a_i^{(k)}/a_j^{(k)}\right)}{\prod_{i=1}^{N_1 - 2k + 2} \prod_{j=1}^{N_1 - 2k} \vartheta_1\left(y_1^{\frac{-2N_1}{(N_1 - 2k + 2)(N_1 - 2k)}} a_i^{(k-1)}/a_j^{(k)}\right)}$$

$$\times \frac{\eta(q)^{\frac{1}{4}(3N_1 - N_2)N} \prod_{i=1}^{N_1} \prod_{j=1}^{N_2} \vartheta_1\left(y_2 y_1^{-1} b_i^{-1} c_j\right)}{\prod_{i=1}^{N_2 + 2} \prod_{j=1}^{N_2} \vartheta_1\left(a_i^{\left(\frac{N}{2}\right)} c_j^{-1} y_2^{-1} y_1^{\frac{N_1}{N_2 + 2}}\right)}, \tag{46}$$

where the fugacity $a^{(0)} = b$.

- $\text{Sp}(N_1 - N_2 - 2)$ gauge theory with $N_1$ fundamental chirals and $N_2$ fundamental Fermi's. The elliptic genus of the theory is

$$\mathcal{I}_{\text{Sp}} = \frac{\eta(q)^{\frac{N^2}{2} + 3N}}{2^{\frac{N}{2}} \frac{N}{2}!} \oint_{\text{JK}} \frac{d\boldsymbol{a}}{2\pi i \boldsymbol{a}} \prod_{i=1}^{\frac{N}{2}} \frac{\vartheta_1(a_i^{\pm 2}) \prod_{j<i} \vartheta_1(a_i^{\pm} a_j^{\pm}) \cdot \prod_{j=1}^{N_2} \vartheta_1(a_i^{\pm} c_j y_2)}{\prod_{j=1}^{N_1} \vartheta_1(a_i^{\pm} b_j^{-1} y_1^{-1})}. \tag{47}$$

Note that, throughout this subsection, we assume that $N \equiv N_1 - N_2 - 2$ is even.

In fact, the central charges of both theories match:

$$c_L = (N_1 - N_2 - 2)(N_1 - N_2 + 1), \qquad c_R = \frac{3}{2}(N_1 - N_2 - 2)(N_1 - N_2 + 1). \tag{48}$$

Furthermore, we have verified that the elliptic genera agree, $\mathcal{I}_{\text{quiver}} = \mathcal{I}_{\text{Sp}}$, up to the rank-six JK integrals ($N_1 = 7$ and $N_2 = 1$ case).

At the level of elliptic genera, the Higgsing procedure can be understood in the following way. The elliptic genus (43) reduces precisely to (47) when the $U(1)_x$ flavor fugacity is turned off:

$$\lim_{x \to 1} \mathcal{I}_{\text{SU}} = \mathcal{I}_{\text{Sp}}. \tag{49}$$

Since all theories shown in Fig. 9 are dual to each other, their elliptic genera must coincide with (43). Consequently, the identity (49) holds equally between the second theory in Fig. 9 and the Sp gauge theory. Furthermore, reversing the direction of arrows in the quiver diagrams on

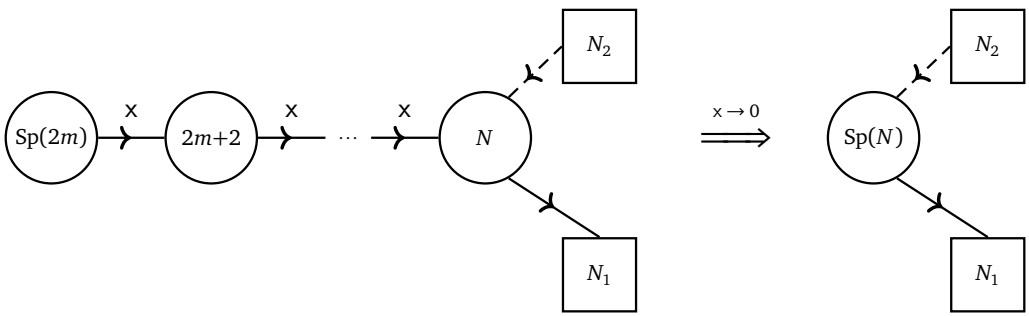

Figure 11: The quiver gauge theory on the left features an $\text{Sp}(2m)$ gauge group at the most left, followed by multiple SU gauge nodes in the middle, where the ranks increase by two at each step from left to right, with $2m < N$. The quiver theory on the right represents an $\text{Sp}(N)$ gauge theory. The two theories are related by Higgsing at the chemical potential $x$ is set to be zero. The elliptic genus of the left quiver gauge theory matches that of the right $\text{Sp}(N)$ gauge theory when the $U(1)_x$ fugacities associated with the bifundamental chiral fields are turned off.

both sides establishes a similar Higgsing relation between the two theories depicted in Fig. 11. Specifically, once the $U(1)_x$ fugacity associated to the bifundamental chials is turned off, the elliptic genus of the quiver theory on the left-hand side in Fig. 11 coincides with that of the Sp gauge theory on the right-hand side. This relation will play a role in deriving additional dualities in subsequent subsections. (See Fig. 14 and Fig. 16.)

## 3.4 SU gauge theories with anti-symmetric chiral

Using the dualities above, we can derive another duality. In this subsection, we will show a duality between SU gauge theories with one anti-symmetric chiral, (anti-)fundamental chirals, and potentially fundamental Fermi multiplets.

We consider an $SU(N_A)$ gauge theory with the following matter content:

- One anti-symmetric chirals $Y$.

- $n_1$ anti-fundamental chirals $X_1$.

- $N$ fundamental chirals $X_2$.

- $m_1$ anti-fundamental Fermi $\Psi$ where $m_1$ is either zero or one.

- A Fermi meson $\Gamma$, transforming as $(\overline{\square}, \square)$ under the flavor symmetries $SU(n_1) \times SU(N)$.

- A chiral meson $M_1$, transforming as $(-1, \square)$ under the flavor symmetries $U(m_1) \times SU(N)$.

- A chiral meson $M_2$, transforming as $(1, \square)$ under the flavor symmetries $U(m_1) \times SU(n_1)$.

- A Fermi meson $\Lambda$, transforming as $\begin{smallmatrix}\overline{\overline{\square}}\\\square\end{smallmatrix} = \wedge^{n_1-2}\square$ under the flavor symmetry $SU(n_1)$.

- $1-m_1$ neutral chiral $P$ under the flavor symmetries $U(m_1)$.

The gauge group is determined by anomaly-free condition, which is given by

$$N_A = N + n_1 - m_1 - 2 \,. \tag{50}$$

The superpotential takes the form

$$\mathcal{W} = \Gamma X_2 X_1 + \Psi M_1 X_2 + \Lambda^{ij} X_{1,i}^{\alpha} X_{1,j}^{\beta} Y_{\alpha\beta} \,. \tag{51}$$

The dual theory has a similar structure: it consists of an $SU(N_B)$ gauge theory, where the matter content is modified by replacing $n_1$ and $m_1$ in the original theory with $n_2$ and $m_2$, respectively, where $m_2 \in \{0,1\}$. The gauge anomaly-free condition in the dual theory is given by

$$N_B = N + n_2 - m_2 - 2 \,. \tag{52}$$

Note that the duality holds only under the condition

$$N \geq m_1 + m_2 + 2 \,. \tag{53}$$

As illustrated in Fig. 12, this duality can be derived through a sequence of duality transformations. Roughly speaking, an SU anti-symmetric chiral multiplet is replaced by an Sp gauge group with a bifundamental chiral multiplet via the duality in either Fig. 6 (AS-dual) or Fig. 7. Thus, the process begins by applying one of these dualities—selecting Fig. 6 (AS-dual) or Fig. 7 depending on whether $N_A$ is even or odd. Following this initial step, the duality shown in Fig. 5 (SU-dual) is applied. To obtain the final dual theory, a second application of either Fig. 6 (AS-dual) or Fig. 7 is required, depending on whether the intermediate gauge group rank is even or odd. This systematic procedure ultimately leads to the dual theory.

Table 3: Summary of symmetries for Theory A (upper one) and Theory B (lower one) in Fig. 12.

| | $X_1$ | $X_2$ | $\Psi$ | $M_1$ | $M_2$ | $\Gamma$ | $Y$ | $\Lambda$ | $P$ |
|---|---|---|---|---|---|---|---|---|---|
| SU($N_A$) | $\overline{\square}$ | $\square$ | $\overline{\square}$ | $\mathbf{1}$ | $\mathbf{1}$ | $\mathbf{1}$ | $\begin{smallmatrix}\square\\\square\end{smallmatrix}$ | $\mathbf{1}$ | $\mathbf{1}$ |
| SU($N$) | $\mathbf{1}$ | $\overline{\square}$ | $\mathbf{1}$ | $\square$ | $\mathbf{1}$ | $\square$ | $\mathbf{1}$ | $\mathbf{1}$ | $\mathbf{1}$ |
| SU($n_1$) | $\square$ | $\mathbf{1}$ | $\mathbf{1}$ | $\mathbf{1}$ | $\square$ | $\overline{\square}$ | $\mathbf{1}$ | $\begin{smallmatrix}\overline{\square}\\\overline{\square}\end{smallmatrix}$ | $\mathbf{1}$ |
| U(1)$_0$ | $-1$ | $1$ | $-N_A-1$ | $N_A$ | $-N_A$ | $0$ | $2$ | $0$ | $-N_A$ |
| U(1)$_1$ | $0$ | $-1$ | $N$ | $1-N$ | $N$ | $1$ | $0$ | $0$ | $N$ |
| U(1)$_2$ | $1$ | $0$ | $-n_1$ | $n_1$ | $1-n_1$ | $-1$ | $0$ | $-2$ | $-n_1$ |
| | $\widetilde{X}_1$ | $\widetilde{X}_2$ | $\widetilde{\Psi}$ | $\widetilde{M}_1$ | $\widetilde{M}_2$ | $\widetilde{\Gamma}$ | $\widetilde{Y}$ | $\widetilde{\Lambda}$ | $\widetilde{P}$ |
| SU($N_B$) | $\overline{\square}$ | $\square$ | $\overline{\square}$ | $\mathbf{1}$ | $\mathbf{1}$ | $\mathbf{1}$ | $\begin{smallmatrix}\square\\\square\end{smallmatrix}$ | $\mathbf{1}$ | $\mathbf{1}$ |
| SU($N$) | $\mathbf{1}$ | $\overline{\square}$ | $\mathbf{1}$ | $\square$ | $\mathbf{1}$ | $\square$ | $\mathbf{1}$ | $\mathbf{1}$ | $\mathbf{1}$ |
| SU($n_2$) | $\square$ | $\mathbf{1}$ | $\mathbf{1}$ | $\mathbf{1}$ | $\square$ | $\overline{\square}$ | $\mathbf{1}$ | $\begin{smallmatrix}\overline{\square}\\\overline{\square}\end{smallmatrix}$ | $\mathbf{1}$ |
| U(1)$_0$ | $\frac{N_A}{N_B}$ | $-\frac{N_A}{N_B}$ | $N_A+\frac{N_A}{N_B}$ | $-N_A$ | $N_A$ | $0$ | $-2\frac{N_A}{N_B}$ | $0$ | $N_A$ |
| U(1)$_1$ | $-\frac{N}{N_B}$ | $\frac{N}{N_B}-1$ | $-\frac{N}{N_B}$ | $1$ | $0$ | $1$ | $2\frac{N}{N_B}$ | $0$ | $0$ |
| U(1)$_3$ | $1$ | $0$ | $-n_2$ | $n_2$ | $1-n_2$ | $-1$ | $0$ | $-2$ | $-n_2$ |

The symmetries of the theory are summarized in Tab. 3. It is straightforward to verify that the 't Hooft anomalies of each flavor symmetry match under the duality, providing a consistency check for this duality.

Moreover, the elliptic genera for these theories are expressed as

$$\mathcal{I}_A = \frac{\eta(q)^{\frac{N_A^2+9N_A-4}{2}}}{N_A!}\oint_{\mathrm{JK}}\frac{d\boldsymbol{a}}{2\pi i\boldsymbol{a}}\prod_{i=1}^{N_A}\frac{\prod_{j\neq i}\vartheta_1(a_i a_j^{-1})\cdot\prod_{j=1}^{m_1}\vartheta_1(a_i^{-1}e_j x_0^{-1})}{\prod_{j<i}\vartheta_1(x_0^2 a_i a_j)\cdot\prod_{j=1}^{N}\vartheta_1(a_i b_j^{-1}x_0 x_1^{-1})\cdot\prod_{j=1}^{n_1}\vartheta_1(a_i^{-1}c_j x_0^{-1}x_2)}$$

$$\times\frac{\prod_{i=1}^{n_1}\prod_{j<i}\vartheta_1(x_2^{-2}c_i^{-1}c_j^{-1})}{\eta(q)^{\frac{n_1(n_1-1)}{2}-N(m_1-n_1)-m_1 n_1-1+m_1}}\prod_{i=1}^{N}\frac{\prod_{j=1}^{n_1}\vartheta_1(b_i c_j^{-1}x_1 x_2^{-1})}{\prod_{j=1}^{m_1}\vartheta_1(b_i e_j^{-1}x_1)}\cdot\frac{\vartheta_1(e_1)^{m_1-1}}{\prod_{i=1}^{m_1}\prod_{j=1}^{n_1}\vartheta_1(e_i c_j x_2)},$$

$$\mathcal{I}_B = \frac{\eta(q)^{\frac{N_B^2+9N_B-4}{2}}}{N_B!}\oint_{\mathrm{JK}}\frac{d\boldsymbol{a}}{2\pi i\boldsymbol{a}}\prod_{i=1}^{N_B}\frac{\prod_{j\neq i}\vartheta_1(a_i a_j^{-1})\cdot\prod_{j=1}^{m_2}\vartheta_1(a_i^{-1}f_j x_0^{-1}y^{-1})}{\prod_{j<i}\vartheta_1(x_0^2 y^2 a_i a_j)\cdot\prod_{j=1}^{N}\vartheta_1(a_i b_j^{-1}x_0 y x_1^{-1})\cdot\prod_{j=1}^{n_2}\vartheta_1(a_i^{-1}d_j x_0^{-1}y^{-1}x_3)}$$

$$\times\frac{\prod_{i=1}^{n_2}\prod_{j<i}\vartheta_1(x_3^{-2}d_i^{-1}d_j^{-1})}{\eta(q)^{\frac{n_2(n_2-1)}{2}-N(m_2-n_2)-m_2 n_2-1+m_2}}\prod_{i=1}^{N}\frac{\prod_{j=1}^{n_2}\vartheta_1(b_i d_j^{-1}x_1 x_3^{-1})}{\prod_{j=1}^{m_2}\vartheta_1(b_i f_j^{-1}x_1)}\cdot\frac{\vartheta_1(f_1)^{m_2-1}}{\prod_{i=1}^{m_2}\prod_{j=1}^{n_2}\vartheta_1(f_i c_j x_3)}, \tag{54}$$

where $n_1+n_2+m_1+m_2$ is even. The fugacities $f_1$, $e_1$ and $y$ are defined as

$$f_1 = x_0^{N_A}x_3^{-n_2}, \quad e_1 = x_0^{-N_A}x_1^N x_2^{-n_1}, \quad y^{N_B} = x_0^{-N_A-N_B}x_1^N. \tag{55}$$

For the special case where $N = m_1 + m_2 + 2$, an additional condition must be satisfied:

$$x_1^{(1-m_1)N}x_2^{-(1-m_1)n_1}x_3^{-(1-m_2)n_2} = x_0^{(m_2-m_1)(n_1+m_2)}. \tag{56}$$

This condition originates from (24), which is based on the relation $N_1 = N_2 + N_3$ in Fig. 5 (SU-dual).

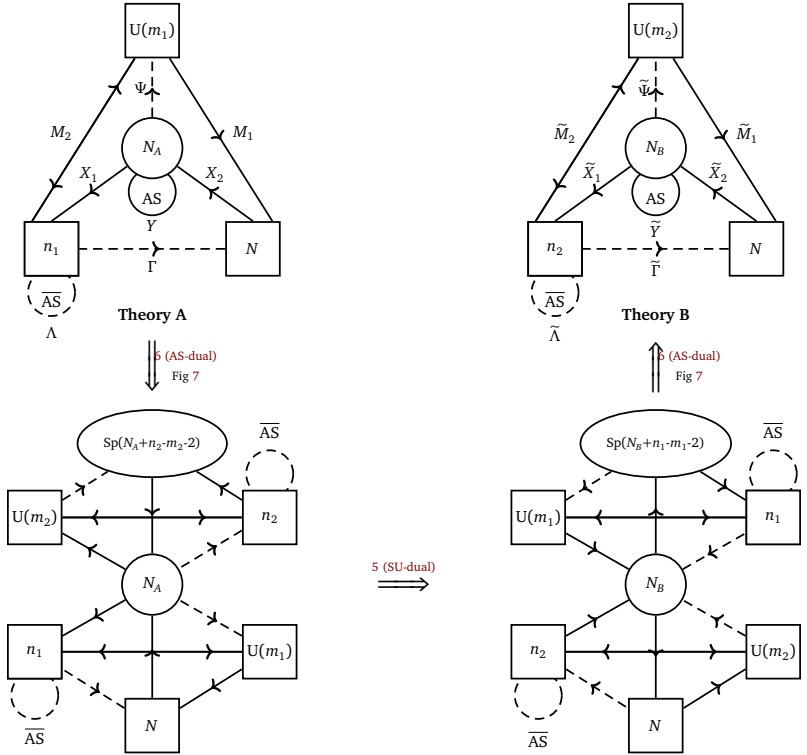

Figure 12: Duality between $SU(N_A)$ and $SU(N_B)$ gauge theories with 1AS where $N_A = N + n_1 - m_1 - 2$ and $N_B = N + n_2 - m_2 - 2$. Note that $m_1$ and $m_2$ are either zero or one.

The elliptic genus $\mathcal{I}_A$ is independent of the $SU(n_1)$ fugacity and the $U(1)_2$ fugacity $x_2$. Likewise, $\mathcal{I}_B$ is independent of the $SU(n_2)$ fugacity and the $U(1)_3$ fugacity $x_3$. Furthermore, the agreement between $\mathcal{I}_A$ and $\mathcal{I}_B$ has been verified up to the rank-five JK integrals through an expansion in $q$.

When $m_1 = 0$ and $N = 2, 3, 4$ in Theory A, Theory B becomes an LG model, as shown in [26]. To obtain the LG dual found in [26], we first transfer the Fermi mesons $\Gamma$ and $\Lambda$ from Theory A to Theory B, where they are reinterpreted as chiral mesons. By appropriately choosing $n_2$ and $m_2$, as described below, the LG duality can be established.

- $N = 2$ **and even** $n_1$: Set $n_2 = 2$ and $m_2 = 0$ in Theory B. This results in a gauge group rank $N_B = N + n_2 - m_2 - 2 = 2$. Using the duality in Fig. 6 (AS-dual), where $SU(2)$ is identified with $Sp(2)$, the corresponding LG dual theory [26, §3.1] is obtained.

- $N = 2$ **and odd** $n_1$: Assign $n_2 = 1$ and $m_2 = 0$, reducing the gauge group rank to $N_B = 1$. This corresponds to an LG theory [26, §3.2].

- $N = 3$ **and even** $n_1$: Choose $n_2 = m_2 = 0$, yielding $N_B = 1$, which directly describes an LG theory [26, §3.3].

- $N = 3$ **and odd** $n_1$: Set $n_2 = 1$ and $m_2 = 0$, giving $N_B = 2$. Applying the duality in Fig. 6 (AS-dual), we obtain the LG dual [26, §3.4].

- $N = 4$ **and even** $n_1$: Select $n_2 = m_2 = 0$, resulting in $N_B = 2$. The LG dual [26, §3.5] follows from the duality in Fig. 6 (AS-dual).

- $N = 4$ **and odd** $n_1$: Assign $n_2 = 0$ and $m_2 = 1$, leading to $N_B = 1$. This implies Theory B is an LG theory [26, §3.6].

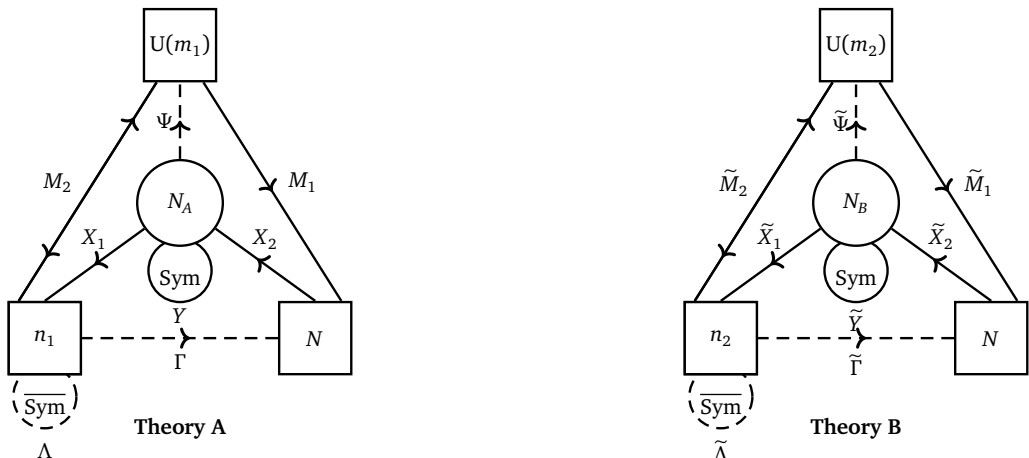

Figure 13: Duality for $SU(N_A)$ and $SU(N_B)$ gauge theories with 1Sym where $N_A = N + n_1 - m_1 + 2$ and $N_B = N + n_2 - m_2 + 2$. Note that $m_1$ and $m_2$ are either zero or one.

The duality holds under the condition in (53), which explicitly excludes the cases where the number of fundamental chirals is $N = 1$ or $N = 0$. Alternative dualities for these excluded cases are discussed in Appendices C.3 and C.4.

## 3.5 SU gauge theories with symmetric chiral

In a similar manner, we establish a duality between SU gauge theories containing one symmetric chiral multiplet, as depicted in Fig. 13. The gauge groups of the dual theories are given by

$$N_A = N + n_1 - m_1 + 2, \qquad N_B = N + n_2 - m_2 + 2. \tag{57}$$

The matter content and superpotential closely resemble those of the previous duality, with the key difference being the replacement of anti-symmetric representations by symmetric representations. To keep our discussion concise, we do not repeat the detailed discussion of matter content and symmetries.

Essentially, through the duality depicted in Fig. 8 (Sym-dual), an SU symmetric chiral can be replaced by an SO gauge group with a bifundamental chiral. Consequently, as in Fig. 12, this duality is derived by sequentially applying the known dualities illustrated in Fig. 8 (Sym-dual) and 5 (SU-dual).

The expressions for the elliptic genera of these theories are provided below:

$$\mathcal{I}_A = \frac{\eta(q)^{\frac{N_A^2+3N_A-4}{2}}}{N_A!} \oint_{JK} \frac{d\boldsymbol{a}}{2\pi i \boldsymbol{a}} \prod_{i=1}^{N_A} \frac{\prod_{j\neq i} \vartheta_1(a_i a_j^{-1}) \cdot \prod_{j=1}^{m_1} \vartheta_1(a_i^{-1} e_j x_0^{-1})}{\prod_{j\leq i} \vartheta_1(x_0^2 a_i a_j) \cdot \prod_{j=1}^{N} \vartheta_1(a_i b_j^{-1} x_0 x_1^{-1}) \cdot \prod_{j=1}^{n_1} \vartheta_1(a_i^{-1} c_j x_0^{-1} x_2)}$$

$$\times \frac{\prod_{i=1}^{n_1} \prod_{j\leq i} \vartheta_1(x_2^2 c_i c_j)}{\eta(q)^{\frac{n_1(n_1+1)}{2}-N(m_1-n_1)-m_1 n_1-1}} \cdot \prod_{i=1}^{N} \frac{\prod_{j=1}^{n_1} \vartheta_1(b_i c_j^{-1} x_1 x_2^{-1})}{\prod_{j=1}^{m_1} \vartheta_1(b_i e_j^{-1} x_1)} \cdot \frac{\vartheta_1(e_1^2)^{-m_1} \vartheta_1(e_1)^{m_1-1}}{\prod_{i=1}^{m_1} \prod_{j=1}^{n_1} \vartheta_1(e_i c_j x_2)},$$

$$\mathcal{I}_B = \frac{\eta(q)^{\frac{N_B^2+3N_B-4}{2}}}{N_B!} \oint_{JK} \frac{d\boldsymbol{a}}{2\pi i \boldsymbol{a}} \prod_{i=1}^{N_B} \frac{\prod_{j\neq i} \vartheta_1(a_i a_j^{-1}) \cdot \prod_{j=1}^{m_2} \vartheta_1(a_i^{-1} f_j x_0^{-1} y^{-1})}{\prod_{j\leq i} \vartheta_1(x_0^2 y^2 a_i a_j) \cdot \prod_{j=1}^{N} \vartheta_1(a_i b_j^{-1} x_0 y x_1^{-1}) \cdot \prod_{j=1}^{n_2} \vartheta_1(a_i^{-1} d_j x_0^{-1} y^{-1} x_3)}$$

$$\times \frac{\prod_{i=1}^{n_2} \prod_{j\leq i} \vartheta_1(x_3^2 d_i d_j)}{\eta(q)^{\frac{n_2(n_2+1)}{2}-N(m_2-n_2)-m_2 n_2-1}} \cdot \prod_{i=1}^{N} \frac{\prod_{j=1}^{n_2} \vartheta_1(b_i d_j^{-1} x_1 x_3^{-1})}{\prod_{j=1}^{m_2} \vartheta_1(b_i f_j^{-1} x_1)} \cdot \frac{\vartheta_1(f_1^2)^{-m_2} \vartheta_1(f_1)^{m_2-1}}{\prod_{i=1}^{m_2} \prod_{j=1}^{n_2} \vartheta_1(f_i d_j x_3)}, \tag{58}$$

where the fugacities $f_1$, $e_1$ and $y$ are defined as

$$f_1 = x_0^{N_A} x_3^{-n_2}, \quad e_1 = x_0^{-N_A} x_1^{N} x_2^{-n_1}, \quad y^{N_B} = x_0^{-N_A-N_B} x_1^{N}. \tag{59}$$

The elliptic genus $\mathcal{I}_A$ does not depend on the SU($n_1$) fugacity or the U(1)$_2$ fugacity $x_2$, while $\mathcal{I}_B$ is independent of the SU($n_2$) fugacity and the U(1)$_3$ fugacity $x_3$. Moreover, the agreement between $\mathcal{I}_A$ and $\mathcal{I}_B$ has been confirmed, via an expansion in $q$, up to the rank-five JK integrals.

### 3.6 Dualities with adjoint chiral

Consider the 4d $\mathcal{N} = 4$ SCFT. Its (0,2) reduction results in the (0,2) gauge theory only with a (0,2) adjoint chiral multiplet $\phi$, which is indeed a (2,2) vector multiplet. The IR central charge of the $\mathcal{N} = (2,2)$ pure Yang-Mills theory with gauge group $G$ is given by

$$c_L = c_R = 3 \operatorname{rank} G. \tag{60}$$

As we will demonstrate, the theory is indeed dual to rank $G$ free $\mathcal{N} = (2,2)$ chiral multiplets.

**Sp($2N$)+1Adj:** Since the duality for the SU($N$) gauge theory with a single adjoint matter field is well-studied in [7,27], we now extend our analysis to other classical gauge groups. We begin by considering the Sp($2N$) gauge theory with an adjoint chiral field.

A useful observation is that Adj $\cong$ Sym for the Sp($2N$) gauge group. This allows us to apply the duality depicted in Fig. 8 (Sym-dual). As illustrated in Fig. 14, a sequence of duality transformations relates Sp($2N$) + 1Adj to Sp($2N-2$) + 1Adj and SU(2) + 1Adj, with an extra chiral multiplet $X$ and a Fermi multiplet $\Psi$.

Furthermore, it is known [7,27] that the SU(2) + 1Adj theory is dual to an $\mathcal{N} = (2,2)$ twisted chiral multiplet consisting of a free $\mathcal{N} = (0,2)$ chiral multiplet $\operatorname{Tr} Y^2$ and a Fermi multiplet $\Gamma$. We claim that they form a superpotential

$$\mathcal{W} = \Gamma X, \tag{61}$$

so that eliminates both the chiral multiplet $X$ and the Fermi multiplet $\Gamma$. As a result, the duality transformation establishes an equivalence between Sp($2N$)+1Adj and Sp($2N-2$)+1Adj with the additional free chiral multiplet $\operatorname{Tr} Y^2$ and the Fermi multiplet $\Psi$.

This duality recursively reduces the rank of the Sp($2N$) gauge group, and at the end, we obtain the following duality:

$$\text{Sp}(2N) + 1\text{Adj} \iff N \text{ free chirals} + N \text{ free Fermi's.} \tag{62}$$

Note that $N$ free chirals correspond to the gauge-invariant operators made out of the adjoint chiral $Z$:

$$\operatorname{Tr}(Z^{2k}), \qquad k = 1, \ldots, N, \tag{63}$$

and $N$ free Fermi's are necessary to form $\mathcal{N} = (2,2)$ twisted chiral multiplets. The duality can be verified by the JK residue computation

$$
\begin{aligned}
\mathcal{I}_{\text{Sp}(2N)}^{\text{Adj}} &= \frac{\eta(q)^{3N} \vartheta_1(y)^N}{2^N N!} \oint_{\text{JK}} \frac{d\boldsymbol{a}}{2\pi i \boldsymbol{a}} \frac{\prod_{i<j}^N \vartheta_1(a_i^\pm a_j^\pm) \cdot \prod_{i=1}^N \vartheta_1(a_i^{2\pm})}{\prod_{i\leq j}^N \vartheta_1(y a_i^\pm a_j^\pm)} \\
&= (-1)^N \prod_{k=1}^N \frac{\vartheta_1(y^{2k-1})}{\vartheta_1(y^{2k})}.
\end{aligned} \tag{64}
$$

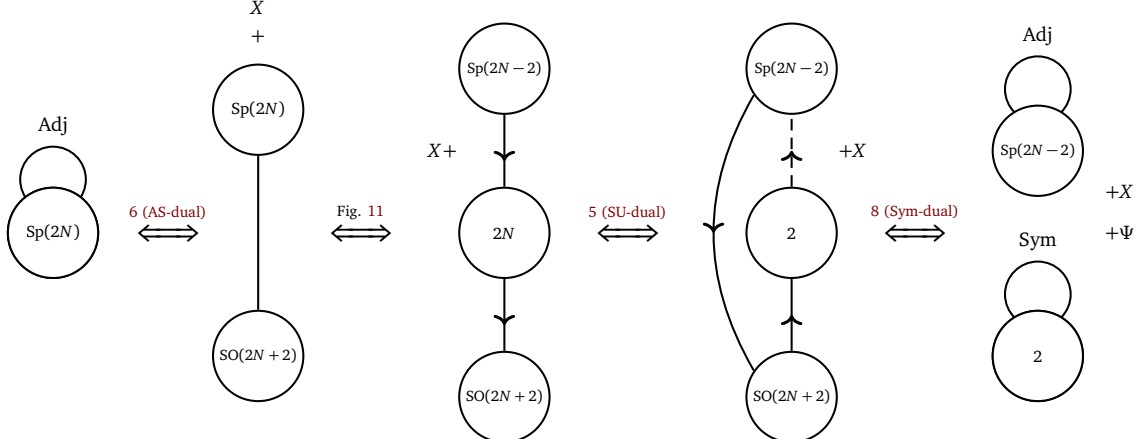

Figure 14: The recursion relation of $\text{Sp}(2N) + 1\text{Adj}$. The leftmost diagram represents the original $\text{Sp}(2N)$ gauge theory with one adjoint chiral. Using a sequence of duality transformations, this theory is mapped to a different description. In particular, from the third to the fourth diagram, we first apply the duality in Fig. 5 (SU-dual) to the $\text{SU}(2N)$ gauge node, followed by the duality in Fig. 8 (Sym-dual) applied to the $\text{Sp}(2N-2)$ gauge node. As a result, the theory is ultimately mapped to $\text{Sp}(2N-2) + 1\text{Adj}$ and $\text{SU}(2) + 1\text{Adj}$, along with additional chiral $X$ and Fermi $\Psi$, forming the superpotential (61). This establishes the recursion pattern, illustrating how $\text{Sp}(2N)$ gauge group transforms into $\text{Sp}(2N-2)$ gauge group, and iterative applications of this procedure lead to lower-rank cases.

**SO($n$)+1Adj:** Now, let us consider the SO($n$) gauge group. Note that Adj $\cong$ AS for the SO($n$) gauge group. Depending on whether $n$ is even or odd, we can apply the dualities presented in Fig. 6 (AS-dual) and Fig. 7.

As illustrated in Fig. 15, these dualities, when followed by the duality in Fig. 8 (Sym-dual), establish an equivalence between the SO($n$) + 1Adj theory and the Sp gauge theory with an adjoint matter field, supplemented by additional chiral and Fermi multiplets. This connects the SO($n$) case to the previously discussed duality for $\text{Sp}(2N) + 1\text{Adj}$ in Eq. (62).

In the upper part of Fig. 15, the SO($2N$) + 1Adj theory is dual to the $\text{Sp}(2N-2) + 1\text{Adj}$ theory, along with an additional free chiral multiplet $X$ and a Fermi multiplet $\Psi$. Examining the U(1) charge, we identify that the chiral multiplet $X$ corresponds to the gauge-invariant operator $\text{Pf}\,Z$, where $Z$ represents the adjoint chiral superfield of the SO($2N$) gauge theory. Consequently, this duality leads to the relation:

$$\text{SO}(2N) + 1\text{Adj} \iff N \text{ free chirals} + N \text{ free Fermi's}, \tag{65}$$

where the $N$ free chiral multiplets correspond to gauge-invariant operators constructed from the adjoint chiral superfield $Z$:

$$\text{Pf}(Z), \quad \text{Tr}(Z^{2k}), \qquad k = 1, \ldots, N-1. \tag{66}$$

In the lower part of Fig. 15, a similar sequence of dualities establishes that the SO($2N + 1$) + 1Adj theory is dual to the $\text{Sp}(2N-2) + 1\text{Adj}$ theory, again with an additional free chiral multiplet $X$ and Fermi multiplet $\Psi$. By examining the U(1) charge, we recognize that the chiral multiplet $X$ corresponds to the gauge-invariant operator $\text{Tr}\,Z^{2N}$, where $Z$ is the adjoint chiral multiplet of the SO($2N + 1$) gauge theory. Incorporating Eq. (62), we arrive at

the following triality:

$$\text{SO}(2N+1)+1\text{Adj} \quad \Longleftrightarrow \quad N \text{ free chirals} + N \text{ free Fermi's}$$

$$\text{Sp}(2N)+1\text{Adj} \tag{67}$$

where the $N$ free chiral multiplets correspond to gauge-invariant operators constructed from the adjoint chiral multiplet $Z$:

$$\text{Tr}(Z^{2k}), \qquad k=1,\dots,N. \tag{68}$$

In both even and odd cases, $N$ free Fermi multiplets are necessary to form $N$ (2,2) free twisted chiral multiplets.

In fact, the explicit computations of the elliptic genera verify the results

$$
\begin{aligned}
\mathcal{I}_{\text{SO}(2N)}^{\textbf{Adj}} &= \frac{2\eta(q)^{3N}}{2^{N-1}N!\cdot\vartheta_1(y)^N}\oint_{\text{JK}}\frac{d\boldsymbol{a}}{2\pi i\boldsymbol{a}}\prod_{i<j}^{N}\frac{\vartheta_1(a_i^\pm a_j^\pm)}{\vartheta_1(ya_i^\pm a_j^\pm)}\\
&= (-1)^N\frac{\vartheta_1(y^{N-1})}{\vartheta_1(y^N)}\cdot\prod_{k=1}^{N-1}\frac{\vartheta_1(y^{2k-1})}{\vartheta_1(y^{2k})},
\end{aligned}
\tag{69}
$$

$$
\begin{aligned}
\mathcal{I}_{\text{SO}(2N+1)}^{\textbf{Adj}} &= \frac{2\eta(q)^{3N}}{2^{N}N!\cdot\vartheta_1(y)^N}\oint_{\text{JK}}\frac{d\boldsymbol{a}}{2\pi i\boldsymbol{a}}\prod_{i<j}^{N}\frac{\vartheta_1(a_i^\pm a_j^\pm)}{\vartheta_1(ya_i^\pm a_j^\pm)}\cdot\prod_{i=1}^{N}\frac{\vartheta_1(a_i^\pm)}{\vartheta_1(ya_i^\pm)}\\
&= (-1)^N\prod_{k=1}^{N}\frac{\vartheta_1(y^{2k-1})}{\vartheta_1(y^{2k})}.
\end{aligned}
\tag{70}
$$

It is easy to verify the identities of the elliptic genera coming from the isomorphisms between gauge algebras:

$$\mathcal{I}_{\text{SO}(4)}^{\textbf{Adj}} = (\mathcal{I}_{\text{SU}(2)}^{\textbf{Adj}})^2, \qquad \mathcal{I}_{\text{SO}(5)}^{\textbf{Adj}} = \mathcal{I}_{\text{Sp}(2)}^{\textbf{Adj}}, \qquad \mathcal{I}_{\text{SO}(6)}^{\textbf{Adj}} = \mathcal{I}_{\text{SU}(4)}^{\textbf{Adj}}. \tag{71}$$

The chiral algebra of the 4d $\mathcal{N}=4$ theory with gauge group $G$ admits the free-field realization by the $bc\beta\gamma$ system [28]. The elliptic genus (69, 70, 64) is the vacuum character of the corresponding $bc\beta\gamma$ system up to a factor [29]. Consequently, the elliptic genera are reducible module characters of the chiral algebra of the 4d $\mathcal{N}=4$ theory with the corresponding gauge groups.

## 3.7 SU gauge theories with symmetric and anti-symmetric chirals

We now consider the SU($N$) gauge theory that includes both symmetric and anti-symmetric chiral multiplets. As discussed earlier, an SU symmetric chiral multiplet can be replaced by an SO gauge group with bifundamental fields, while an SU anti-symmetric chiral multiplet can be replaced by an Sp gauge group with bifundamental fields. We then apply a sequence of duality transformations to obtain an LG dual model. This procedure closely parallels the approach in the previous subsection, where we derived a recursion relation to systematically reduce the ranks of the gauge groups.

The replacement of the SU anti-symmetric chiral multiplet with an Sp gauge field depends on whether $N$ is even or odd—specifically, via Fig. 6 (AS-dual) for even $N$ and Fig. 7 for odd $N$. Therefore, we divide the discussion into two cases: even and odd $N$.

Consider the case that $N$ is even. As shown in Fig. 16, a sequence of the known dualities provides a "recursion relation" of SU+1Sym+1AS theories in terms of the ranks of gauge

groups. Starting with SU($N$)+1Sym+1AS, supplemented by additional 1 chiral multiplets $Z^{(0)}$ and 3 Fermi multiplets $\Psi^{(0)}_{1,2,3}$, with a superpotential

$$\mathcal{W} = \Psi^{(0)}_1 \det X^{(0)} + \Psi^{(0)}_2 \mathrm{Pf}\, Y^{(0)} + \Psi^{(0)}_3 \left( (Z^{(0)})^2 + B^{(0)}(2) \right). \tag{72}$$

As shown in Fig. 16, a series of duality transformations systematically map the theory to a similar theory: SU($N-2$)+1Sym+1AS, along with additional chiral multiplets $Z^{(1)}, \widetilde{Z}^{(1)}$ and Fermi multiplets $\Psi^{(1)}_{1,2,3,4}$. (Here, $\Psi^{(0)}_2$ forms a superpotential with a newly generated chiral field $P^{(1)}$, which then decouples from the remaining theory. Since $\Psi^{(0)}_2$ and $P^{(1)}$ do not contribute to either the central charge or the elliptic genus, we will ignore it in the subsequent discussion for convenience.) The superpotential of the dual side takes a similar form:

$$\mathcal{W} = \Psi^{(1)}_1 (\det X^{(1)} + \widetilde{Z}^{(1)}) + \Psi^{(1)}_2 \mathrm{Pf}\, Y^{(1)} + \Psi^{(1)}_3 \left( (Z^{(1)})^2 + B^{(1)}(2) \right) + \Psi^{(0)}_3 \left( (Z^{(0)})^2 + \mathrm{Tr}\left[ (A^{(0)})^2 \right] \right), \tag{73}$$

where $B^{(i)}(n)$ is a neutral baryon with U(1)$_{x^{(i)}}$ × U(1)$_{y^{(i)}}$ charge $(2n, 2(N-2i-n))$ that defined as:

$$B^{(i)}(n) \equiv \epsilon^{\alpha_1 \dots \alpha_{N-2i}} \epsilon^{\beta_1 \dots \beta_{N-2i}} Y^{(i)}_{\alpha_1 \beta_1} \dots Y^{(i)}_{\alpha_{N-n-2i} \beta_{N-n-2i}} X^{(i)}_{\alpha_{N-n-2i+1} \beta_{N-n-2i+1}} \dots X^{(i)}_{\alpha_{N-2i} \beta_{N-2i}}. \tag{74}$$

Moreover, the dual side also has an SU(2)+1Sym theory with the symmetric tensor $A^{(0)}$, and couples, via (73), with the chiral $Z^{(0)}$ and Fermi $\Psi^{(0)}_3$ from the original theory, as shown

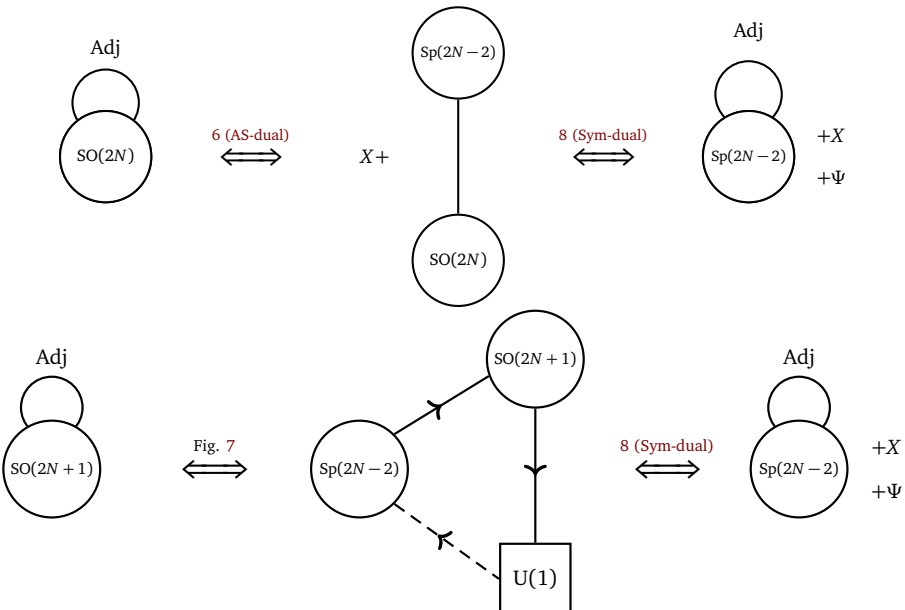

Figure 15: A chain of dualities provides the relation between SO+1Adj and Sp+1Adj. In the upper part of the diagram, the SO($2N$) + 1Adj theory is dual to the Sp($2N-2$) + 1Adj theory, with additional free chiral $X$ and Fermi $\Psi$. The chiral $X$ corresponds to the gauge-invariant operator $\mathrm{Pf}\, Z$, where $Z$ is the adjoint chiral of the SO($2N$) gauge theory.

In the lower part of the diagram, a similar chain of dualities relates the SO($2N+1$) + 1Adj theory to Sp($2N-2$) + 1Adj, again with additional free chiral $X$ and Fermi $\Psi$. Here, the free chiral $X$ corresponds to the gauge-invariant operator $\mathrm{Tr}\, Z^{2N}$, where $Z$ is the adjoint chiral of the SO($2N+1$) gauge theory. Consequently, SO($2N+1$) + 1Adj is dual to Sp($2N$) + 1Adj.

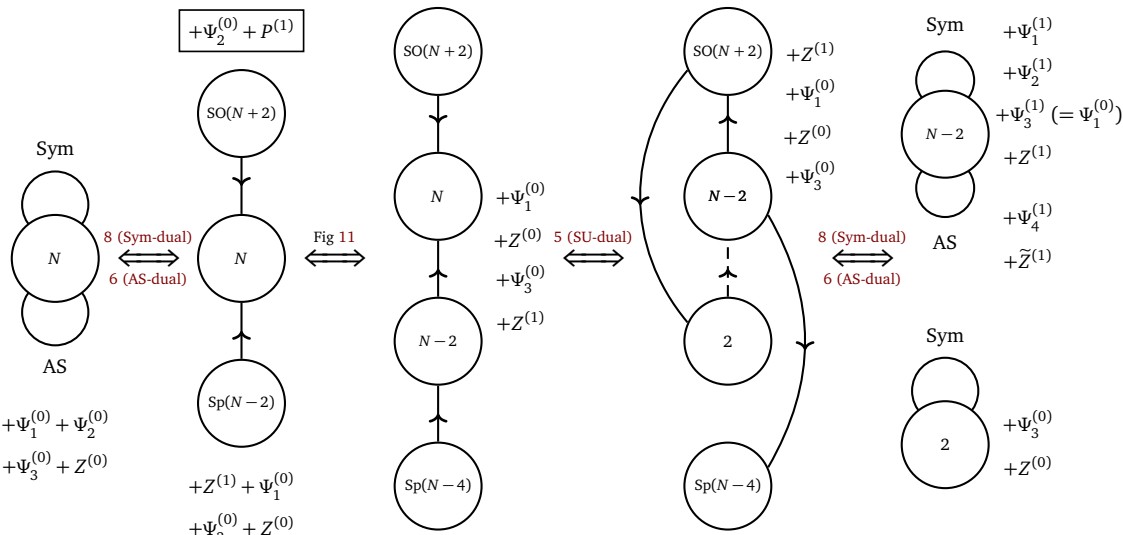

Figure 16: The first quiver on the left corresponds to the theory SU($N$)+1AS+1Sym, along with three Fermi's $\Psi^{(0)}_{1,2,3}$ and one chiral field $Z^{(0)}$. By applying the Fig. 8 (Sym-dual) and Fig. 6 (AS-dual) transformations, we obtain the second quiver, which describes a new theory. This new theory generates two neutral chirals $P^{(1)}$, $Z^{(1)}$: $P^{(1)}$ couples with $\Psi^{(0)}_2$ and decouples from the other fields in the theory. From the third to the fourth diagram, we first apply the duality in Fig. 5 (SU-dual) to the SU($N-2$) gauge node, followed by the same duality applied to the SU($N$) gauge node. As a result, the theory is ultimately mapped to SU($N-2$)+1Sym+1AS and SU(2)+1Sym, along with additional chirals $Z^{(0)}, Z^{(1)}, \widetilde{Z}^{(1)}$ and Fermi's $\Psi^{(0)}_3, \Psi^{(1)}_{1,2,3,4}$. This establishes the recursion pattern, illustrating how SU($N$) gauge group transforms into SU($N-2$) gauge group and iterative applications of this procedure lead to lower-rank cases.

in the last quiver of Fig. 16. Again, the SU(2)+1Sym theory is dual to one chiral $\mathrm{Tr}\big(A^{(0)}\big)^2$ and Fermi with opposite charges with the chiral $Z^{(0)}$ and Fermi $\Psi^{(0)}_3$. As a result, their combined contribution to the elliptic genus cancels out:

$$\underbrace{\frac{\vartheta_1(x^{-2}y^{-N+2})}{\vartheta_1(x^4 y^{2N-4})}}_{\text{SU(2)+1Sym}} \cdot \overbrace{\underbrace{\frac{\vartheta_1(x^{-4}y^{-2N+4})}{\vartheta_1(x^2 y^{N-2})}}_{Z^{(0)}}}^{\Psi^{(0)}_3} = 1\,. \tag{75}$$

Although their contributions to the elliptic genus cancel out, these fields are not entirely negligible, as their combined central charge contribution from SU(2)+1Sym, $\Psi^{(0)}_3$, and $Z^{(0)}$ is 3. This establishes a recursive pattern of SU+1Sym+1AS demonstrating how an SU($N$) gauge group systematically reduces to SU($N-2$) through a series of duality transformations, as emphasized in Fig. 16. We summarize the charges of the fields in the SU($N$)+1AS+1Sym theory in Tab. 4, as well as the charges of the fields in the SU($N-2$)+1AS+1Sym theory obtained after performing this series of duality transformations.

By repeatedly applying this process, the gauge group rank is systematically reduced, allowing the theory to be expressed in terms of lower-rank cases. The duality recursion reaches its lowest order in the case of SU(2)+1Sym+1AS along with additional Fermi and chiral fields.

With this definition of $B^{(i)}(n)$, it is straightforward to observe that

$$B^{(i)}(0) \propto \det Y^{(i)}\,, \quad B^{(i)}(N-2i) \propto \det X^{(i)}\,. \tag{76}$$

Table 4: We start from the theory in the left table, which includes the Sym chiral $X^{(0)}$ and the AS chiral $Y^{(0)}$ under SU($N$), as well as three Fermi fields $\Psi^{(0)}_{1,2,3}$ and a chiral field $Z^{(0)}$. After performing the duality depicted in Fig. 16, we obtain the fields in the right table, which include the Sym chiral $X^{(1)}$ and the AS chiral $Y^{(1)}$ under SU($N-2$), four Fermi fields $\Psi^{(1)}_{1,2,3,4}$, and two chiral fields $Z^{(1)}$ and $\widetilde{Z}^{(1)}$. The U(1)$_{x(i)}$ and U(1)$_{y(i)}$ charges are introduced for convenience, but they can also be rewritten in terms of U(1)$_{x(0)}$ and U(1)$_{y(0)}$ using the recursion relations: $x^{(i+1)} = \frac{4}{N-2i-2}x^{(i)} + \frac{N-2i-4}{N-2i-2}y^{(i)}$ and $y^{(i+1)} = \frac{N-2i+2}{N-2i-2}x^{(i)} - \frac{2}{N-2i-2}y^{(i)}$.

| | SU($N$) | U(1)$_{x(0)}$ | U(1)$_{y(0)}$ |
|---|---|---|---|
| $X^{(0)}_{\alpha\beta}$ | ⊞ | 2 | 0 |
| $Y^{(0)}_{\alpha\beta}$ | ⊟ | 0 | 2 |
| $\Psi^{(0)}_1$ | 1 | $-2N$ | 0 |
| $\Psi^{(0)}_2$ | 1 | 0 | $-N$ |
| $\Psi^{(0)}_3$ | 1 | $-4$ | $-2N+4$ |
| $Z^{(0)}$ | 1 | 2 | $N-2$ |

| | SU($N-2$) | U(1)$_{x(1)}$ | U(1)$_{y(1)}$ |
|---|---|---|---|
| $X^{(1)}_{\alpha\beta}$ | ⊞ | 2 | 0 |
| $Y^{(1)}_{\alpha\beta}$ | ⊟ | 0 | 2 |
| $\Psi^{(1)}_1$ | 1 | $-2(N-2)$ | 0 |
| $\Psi^{(1)}_2$ | 1 | 0 | $-(N-2)$ |
| $\Psi^{(1)}_3$ | 1 | $-4$ | $-2(N-2)+4$ |
| $\Psi^{(1)}_4$ | 1 | $-2N+2$ | 2 |
| $Z^{(1)}$ | 1 | 2 | $(N-2)-2$ |
| $\widetilde{Z}^{(1)}$ | 1 | $2N-4$ | 0 |

This allows us to identify part of the free field relations between the two dual theories:

$$\Psi^{(i)}_3 = \Psi^{(i-1)}_1, \qquad Z^{(i)} = \text{Tr}\big[(X^{(i-1)})^{N/2}\big], \qquad \widetilde{Z}^{(i)} = B^{(i-1)}(4). \tag{77}$$

At the final stage of the duality sequence, we obtain $\frac{N-2}{2}$ copies of SU(2) + 1Sym, along with $N+1$ Fermi multiplets, $N-1$ chiral multiplets, and an SU(2) + 1AS + 1Sym theory, as illustrated in Fig. 17. The corresponding superpotential takes the form:

$$\mathcal{W} = \Psi^{(\frac{N-2}{2})}_1\Big(\det X^{(\frac{N-2}{2})} + \widetilde{Z}^{(\frac{N-2}{2})}\Big) + \Psi^{(\frac{N-2}{2})}_2 \text{Pf} Y^{(\frac{N-2}{2})} + \Psi^{(\frac{N-2}{2})}_3 \det X^{(\frac{N-2}{2})}$$

$$+ \sum_{i=0}^{1} \Psi^{(i)}_3\big((Z^{(i)})^2 + \text{Tr}[(A^{(i)})^2]\big) + \sum_{i=2}^{\frac{N-2}{2}} \Psi^{(i)}_3\big(\widetilde{Z}^{(i-1)} + (Z^{(i)})^2 + \text{Tr}[(A^{(i)})^2]\big). \tag{78}$$

For SU(2), the antisymmetric representation is trivial. As discussed earlier, SU(2) + 1Sym is dual to a pair of chiral and Fermi fields. Therefore, the recursive duality takes the following form

$$
\begin{array}{ccc}
\begin{matrix}\text{SU}(N) + 1\text{Sym} + 1\text{AS} + 3\text{ Fermi's} + 1\text{ chiral} \\ \text{with superpotential (72)}\end{matrix} & & \begin{matrix}\left(\frac{3N}{2}\right)\text{ chirals } + \left(\frac{3N}{2}+1\right)\text{ Fermi's} \\ \text{with superpotential (78)}\end{matrix} \\
\searrow & & \nearrow \\
& \begin{matrix}\text{SU}(N-2) + 1\text{Sym} + 1\text{AS} + 6\text{ Fermi's} + 4\text{ chirals} \\ \text{with superpotential (73)}.\end{matrix} &
\end{array}
\tag{79}
$$

At each step of this iterative procedure, we neglect the fields $\Psi^{(i)}_2$ and $P^{(i+1)}$, as they decouple from the theory due to the superpotential term $\mathcal{W} = \Psi^{(i)}_2 P^{(i+1)}$. The matter content on the right side of the first row is obtained by applying this duality iteration to the final stage, as illustrated in Fig. 17.



Figure 17: In the final stage of the iterative procedure, we have $\frac{N-2}{2}$ copies of SU(2)+1Sym, where $A^{(i)}$ is the corresponding symmetric tensor with charges $(2, N-2i-2)$ under $U(1)_{x^{(i)}} \times U(1)_{y^{(i)}}$. Since each SU(2)+1Sym is dual to a pair of Fermi and chiral fields, so these SU(2)+1Sym theories are dual to $\frac{N-2}{2}$ pairs of chiral and Fermi fields. Additionally, we have a total of $N+1$ Fermi fields, denoted as $\Psi_{3,4}^{(i)}$, and a total of $N-1$ chiral fields, denoted as $Z^{(i)}$ and $\widetilde{Z}^{(i)}$. Finally, there is the last-stage SU(2)+1AS+1Sym, where $Y^{(\frac{N-2}{2})}$ is the AS tensor and $X^{(\frac{N-2}{2})}$ is the Sym tensor. As previously mentioned, the AS representation of SU(2) is trivial, so this SU(2)+1AS+1Sym theory is dual to two chiral fields and one Fermi field. The corresponding final superpotential is given by (78).

For free Fermi and free chiral fields, we assign an R-charge of 0. For interacting Fermi and chiral fields, we assign R-charges of 1 and 0, respectively. With these assignments, we compute the central charge of the theory as:

$$c_L = 2N - 4, \qquad c_R = 3N - 6. \tag{80}$$

The elliptic genus for even $N$ is given by:

$$\mathcal{I} = \frac{\eta(q)^{3N-4}}{N!} \frac{\vartheta_1(x^{-2N})\vartheta_1(y^{-N})\vartheta_1(x^{-4}y^{4-2N})}{\vartheta_1(x^2 y^{N-2})} \oint_{\text{JK}} \frac{d\boldsymbol{a}}{2\pi i \boldsymbol{a}} \prod_{i=1}^{N} \frac{\prod_{j \neq i} \vartheta_1(a_i/a_j)}{\prod_{j \leq i} \vartheta_1(x^2 a_i a_j) \cdot \prod_{j < i} \vartheta_1(y^2 a_i a_j)}$$

$$= (-1)^{\frac{N}{2}+1} \frac{\vartheta_1(x^{2N})}{\eta(q)} \prod_{i=1}^{\frac{N-2}{2}} \frac{\vartheta_1(x^{4i+2} y^{2N-4i-2})}{\vartheta_1(x^{4i+4} y^{2N-4i-4})}. \tag{81}$$

When $N$ is odd, we can start with the theory SU($N$)+1AS+1Sym, along with three Fermi's $\Psi_{1,2,3}$ and a free chiral $Z$. By applying a series of dualities as illustrated in Fig. 18, we can transform it into the even case ($N-1$), which corresponds to the theory SU($N-1$)+1AS+1Sym, along with three Fermi's $\Psi_{1,2,3}^{(0)}$ and one chiral $Z^{(0)}$. This allows us to use the aforementioned formulas and dualities for iterative derivation. The corresponding charges for these two theories are provided in Tab 5. Through this duality, we can apply the iterative dualities illustrated in Fig. 16, ultimately resulting in $\frac{N-1}{2}+3$ free Fermi's and $\frac{N-1}{2}+2$ free chirals.

The superpotential for the odd $N$ case is given by:

$$\mathcal{W} = \Psi_1 \det X + \Psi_2 B(2) + \Psi_3 B(3). \tag{82}$$

We identify some of the relations of the two sides as:

$$\Psi_1^{(0)} = \Psi_3, \quad \Psi_3^{(0)} = \Psi_1, \quad Z^{(0)} = \text{Tr}\big[X^{N/2}\big]. \tag{83}$$

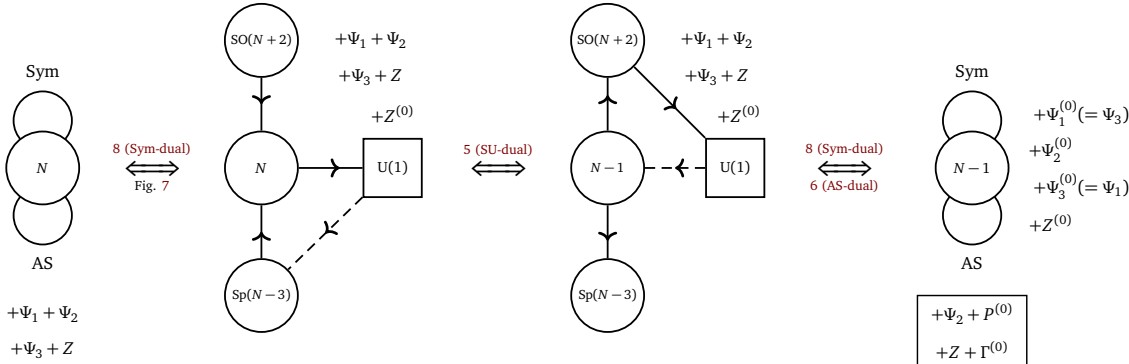

Figure 18: A chain of duality transformations relate SU($N$)+1Sym+1AS to SU($N-1$)+1Sym+1AS where $N$ is odd. In the final quiver, $Z^{(0)}$ and $P^{(0)}$ are chiral fields generated during the duality process, while $\Psi_2^{(0)}$ and $\Gamma^{(0)}$ are newly generated Fermions. The boxed components do not contribute to the elliptic genus. Furthermore, $\Psi_2$ couples to $P^{(0)}$, $Z$ is a free chiral field, and $\Gamma^{(0)}$ is a free Fermion.

Table 5: The table on the left corresponds to the charge table of the theory SU($N$)+1AS+1Sym with three Fermi's $\Psi_{1,2,3}$ and one free chiral $Z$ field when $N$ is odd. The table on the right represents the theory obtained after applying a series of transformations as depicted in Fig. 18, namely SU($N-1$)+1AS+1Sym with three Fermi's $\Psi_{1,2,3}^{(0)}$ and one chiral $Z^{(0)}$. Here, U(1)$_{x^{(0)}}$ and U(1)$_{y^{(0)}}$ are introduced for convenience, but they can be rewritten in terms of U(1)$_x$ and U(1)$_y$, with the charge relations given by $x^{(0)} = \frac{3}{N-1}x + \frac{N-3}{N-1}y$ and $y^{(0)} = \frac{N+2}{N-1}x - \frac{2}{N-1}y$.

| | SU($N$) | U(1)$_x$ | U(1)$_y$ |
|---|---|---|---|
| $X_{\alpha\beta}$ | ⊞ | 2 | 0 |
| $Y_{\alpha\beta}$ | 目 | 0 | 2 |
| $\Psi_1$ | 1 | $-2N$ | 0 |
| $\Psi_2$ | 1 | $-2$ | $-2N+2$ |
| $\Psi_3$ | 1 | $-6$ | $-2N+6$ |
| $Z$ | 1 | 4 | $2N-4$ |

| | SU($N-1$) | U(1)$_{x^{(0)}}$ | U(1)$_{y^{(0)}}$ |
|---|---|---|---|
| $X_{\alpha\beta}^{(0)}$ | ⊞ | 2 | 0 |
| $Y_{\alpha\beta}^{(0)}$ | 目 | 0 | 2 |
| $\Psi_1^{(0)}$ | 1 | $-2(N-1)$ | 0 |
| $\Psi_2^{(0)}$ | 1 | 0 | $-(N-1)$ |
| $\Psi_3^{(0)}$ | 1 | $-4$ | $-2(N-1)+4$ |
| $Z^{(0)}$ | 1 | 2 | $(N-1)-2$ |

The corresponding elliptic genus when $N$ is odd is:

$$
\mathcal{I} = \frac{\eta(q)^{3N-4}}{N!} \frac{\vartheta(x^{-2N})\vartheta(x^{-2}y^{-2N+2})\vartheta(x^{-6}y^{-2N+6})}{\vartheta(x^4 y^{2N-4})} \oint_{\mathrm{JK}} \frac{d\boldsymbol{a}}{2\pi i \boldsymbol{a}} \prod_{i=1}^{N} \frac{\prod_{j \neq i}\vartheta_1(a_i/a_j)}{\prod_{j \leq i}\vartheta_1(x^2 a_i a_j) \cdot \prod_{j < i}\vartheta_1(y^2 a_i a_j)}
$$

$$
= (-1)^{\frac{N+1}{2}} \frac{\vartheta(x^6 y^{2N-6})}{\eta(q)} \prod_{i=1}^{\frac{N-3}{2}} \frac{\vartheta(x^{4i+4}y^{2N-4i-4})}{\vartheta(x^{4i+2}y^{2N-4i-2})}. \tag{84}
$$

## 4 Conclusions

This paper has uncovered new dualities in 2d $\mathcal{N} = (2,2)$ and $\mathcal{N} = (0,2)$ supersymmetric theories, expanding the known landscape of 2d dualities and showing their connections to 4d physics. Table 6 summarizes the 4d $\mathcal{N} = 1$ and $\mathcal{N} = 2$ theories considered in this work, along with the corresponding 2d dualities obtained via twisted compactification on $S^2$. Furthermore, we derive additional 2d dualities from these constructions.

Table 6: Summary of 4d $\mathcal{N} = 1$ and $\mathcal{N} = 2$ theories and their corresponding 2d $\mathcal{N} = (2,2)$ and $\mathcal{N} = (0,2)$ dualities obtained via twisted compactification on $S^2$. Each entry in the right column points to the relevant section or figure where the associated 2d duality is discussed.

| 4d theories | 2d dualities |
|:---:|:---:|
| $\mathcal{N} = 2$ Lagrangian class $\mathcal{S}$ theories | $\mathcal{N} = (2,2)$ dualities in §2 |
| $\mathcal{N} = 1$ Seiberg dual theories | $\mathcal{N} = (0,2)$ duality in Fig. 5 (SU-dual) |
| $\mathcal{N} = 1$ Intriligator–Pouliot dual theories<br>$\mathcal{N} = 2$ Sp$(2N) + (2N+2)\mathbf{F}$<br>$\mathcal{N} = 2$ SU$(2N) + 1\mathbf{AS} + (2N+2)\mathbf{F}$ | $\mathcal{N} = (0,2)$ triality in Fig. 6 (AS-dual) |
| $\mathcal{N} = 1$ Intriligator–Seiberg dual theories<br>$\mathcal{N} = 2$ SO$(N) + (N-2)\mathbf{F}$<br>$\mathcal{N} = 2$ SU$(N) + 1\mathbf{Sym} + (N-2)\mathbf{F}$ | $\mathcal{N} = (0,2)$ triality in Fig. 8 (Sym-dual) |
| $\mathcal{N} = 4$ SYM theory | $\mathcal{N} = (2,2)$ pure-YM/LG duality in §3.6 |

Specifically, we derive new 2d $\mathcal{N} = (0,2)$ dualities by the twisted compactification of 4d $\mathcal{N} = 1$ dualities on $S^2$. We also show that the $(0,2)$ reduction of a 4d $\mathcal{N} = 2$ SCFT yields 2d $\mathcal{N} = (0,2)$ gauge theory that is dual to a Landau-Ginzburg model—even when the parent theory does not belong to class $\mathcal{S}$. This indicates that Gauge/LG dualities with $\mathcal{N} = (0,2)$ supersymmetry arising from $\mathcal{N} = 2$ SCFTs in 4d are more general than previously understood. As shown in [30], every 4d $\mathcal{N} = 2$ SCFT admits a chiral algebra structure derived from its Schur BPS sector. Building on the results of [16, 31], it was proposed in [7] that the elliptic genus of the $(0,2)$ theory obtained from a Lagrangian class $\mathcal{S}$ theory of type $A$ can be expressed as a linear combination of characters of the corresponding chiral algebra. It would be intriguing to investigate whether such a correspondence between elliptic genera and chiral algebra characters extends more broadly to the $(0,2)$ reductions studied in this work.

While we have found various novel 2d dualities, this study certainly represents only a modest step into what is undoubtedly a vast and intricate domain, with many open questions remaining to be explored. One natural avenue for future investigation is to seek brane constructions that realize the new dualities and trialities presented in this work. Brane setups have proven to be a powerful tool in understanding 2d $(0,2)$ gauge theories, as demonstrated in [32–35]. Identifying brane realizations of these newly discovered dualities could provide a more systematic framework for understanding their structure and potential generalizations.

Another promising direction is to explore connections between these new dualities and integrable structures. The interplay between 2d supersymmetric gauge theories and integrable systems has been studied in [36–38], and it would be interesting to investigate whether the dualities we have found correspond to known integrable models or lead to new integrable structures.

2d supersymmetric theories represent exceptionally fertile ground in theoretical physics. Although intensive study has been devoted to 2d supersymmetric theories since the foundational work [1], our understanding of both $\mathcal{N} = (2,2)$ and $\mathcal{N} = (0,2)$ theories remains limited. Despite their rich theoretical landscape, a vast "*Terra incognita*" remains largely unexplored. We believe that the following topics need further study:

- Further discovery and analysis of $\mathcal{N} = (2,2)$ and $\mathcal{N} = (0,2)$ non-Abelian dualities.

- Supersymmetric enhancement from $\mathcal{N} = (0,2)$ to $\mathcal{N} = (2,2)$ theories.

- Construction and investigation of $\mathcal{N} = (2,2)$ and $\mathcal{N} = (0,2)$ non-Lagrangian theories.

- Supersymmetric boundary conditions in $\mathcal{N} = (2,2)$ and $\mathcal{N} = (0,2)$ theories.

We conclude this short paper with the hope that motivated readers will further develop this field.

## Acknowledgments

First and foremost, we would like to express our sincere gratitude to SciPost referees for their detailed and insightful comments, as well as for their patience in carefully reviewing our manuscript. Their invaluable feedback has greatly improved the quality of our paper.

We are grateful to Yiwen Pan for his collaboration on our previous work [7], which laid the foundation for this paper. Our thanks extend to Runkai Tao for his contributions in the early stages of this project, and to Yiwen Pan and Yuji Tachikawa for their valuable comments on the draft. In addition, we would like to thank Wei Cui, Junkang Huang, Zixiao Huang, Du Pei, and Shutong Zhuang for their insightful discussions. In particular, we would like to extend special thanks to Mauricio Romo for his substantial contributions, illuminating discussions, and generous assistance throughout this work.

**Funding information** This work is supported by the Shanghai Municipal Science and Technology Major Project (No. 24ZR1403900).

## A Conventions

For symplectic groups, we use the notation $\mathrm{Sp}(N)$, where $N$ is twice the rank, ensuring that $N$ is always even. For various SU, Sp, and SO gauge theories, we frequently adopt the following concise notations for a representation of a chiral multiplet under a gauge group:

- Fundamental representation: **F**.

- Anti-fundamental representation: **AF**.

- Adjoint representation: **Adj**.

- anti-symmetric representation: **AS**.

- Symmetric representation: **Sym**.

Additionally, all quiver diagrams for $\mathcal{N} = (0,2)$ theories follow the conventions outlined below,

- Gauge node for a gauge group $G$: $\left(\; G \;\right)$

- Flavor node corresponding to a flavor group $F$ with chemical potential f: $\boxed{F}_{\,\mathrm{f}}$

- Chiral multiplet: ───────

- Fermi multiplet: ─ ─ ─ ─ ─ ─ ─ ─

- In the case of the $\mathrm{SU}(N)$ group, we simply write $N$ in both flavor and gauge nodes.

- For representations of SU($N$) groups, an inward-pointing arrow indicates a multiplet in the fundamental representation **F**, while an outward-pointing arrow denotes a multiplet in the anti-fundamental representation **AF**. For example, $N_1 \times N_2$ chiral mesons in the representation $\overline{\mathbf{N_1}} \otimes \mathbf{N_2}$ under the flavor group SU($N_1$) × SU($N_2$) are represented as:

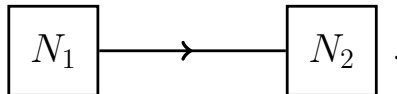

- For clarity, the charges and chemical potentials of the U(1) flavor symmetries associated with various multiplets are sometimes indicated in the quiver diagram. For instance, the chiral multiplet carries a charge of 2 under the U(1)$_x$ flavor symmetry is represented as follows:

$$\underline{\qquad\qquad}^{2\mathsf{x}}\underline{\qquad\qquad} \quad .$$

Here, the fugacity $x$ is related to the chemical potential $\mathsf{x}$ by $x = e^{2\pi i \mathsf{x}}$.

## B  Fundamentals of 2d $\mathcal{N} = 2$ theories

In this appendix, we provide a concise review of 2d $\mathcal{N} = 2$ supersymmetric theories relevant to this paper. To ensure that our discussion is self-contained, we present the necessary background and clarify the notation used in the main content and analyses of this paper. Specifically, we examine the anomaly coefficients in 2d $\mathcal{N} = (0,2)$ supersymmetric theories and explain the computation of elliptic genera for both $\mathcal{N} = (0,2)$ and $\mathcal{N} = (2,2)$ supersymmetric theories.

Given that $(0,2)$ theories are chiral, it is crucial to pay attention to anomalies. Consider a $(0,2)$ theory with a global symmetry $F$ described by a simple Lie algebra. The 't Hooft anomaly coefficient $k_F$ associated with this symmetry can be determined by

$$\mathrm{Tr}\, \gamma_3 f^a f^b = k_F \delta^{ab} \,, \tag{B.1}$$

where $f^a$ are the generators of $F$, $\gamma_3$ is the gamma matrix that quantifies chirality, and the trace is taken over the Weyl Fermions in the theory.

In particular, the anomaly associated with the U(1)$_R$-symmetry is related to the right-moving central charge $c_R$ as

$$c_R = 3 \,\mathrm{Tr}\big(\gamma_3 R^2\big). \tag{B.2}$$

To determine the U(1)$_R$-charges of various fields, $c$-extremization is performed [21, 22] if the theory meets the following two assumptions:

1. The theory is bounded, and the energy spectrum is bounded from below.

2. The vacuum moduli space is compact, and the ground state wavefunction is normalizable.

If chiral multiplets parametrize compact directions, we need to apply the $c$-extremization on them. However, if there is no restriction on $R$-charges by a superpotential, the chiral fields spanning non-compact directions must have zero $R$-charge.

In fact, the vacuum moduli space in every $\mathcal{N} = (0,2)$ theory considered in this paper is non-compact, and all the chiral multiplets span some non-compact directions, making $c$-extremization inapplicable. In such cases, the right-moving central charge is determined as three times the complex dimension of the moduli space, which serves as the target space of the non-linear sigma model in the infra-red limit [7].

Once $c_R$ is determined, the left-moving central charge can be obtained from the gravitational anomaly, which is the difference between the number of chiral and Fermi multiplets

$$c_R - c_L = \text{Tr}(\gamma_3). \tag{B.3}$$

To study 2d supersymmetric theories, the elliptic genus is a powerful observable that can be evaluated from the UV description [39, 40]. In this paper, we consider the elliptic genus in the Ramond sector defined by

$$\mathcal{I}^{(0,2)}(q, z) = \text{Tr}_R(-1)^F q^{H_L} \bar{q}^{H_R} \prod_a z_a^{f_a}, \tag{B.4}$$

where the left- and right-moving Hamiltonians are $2H_L = H + iP$ and $2H_R = H - iP$, respectively, in the Euclidean signature. In a superconformal theory, these operators correspond to the zero-mode generators $L_0, \bar{L}_0$ of the superconformal algebra. Due to supersymmetry, only the right-moving ground states ($H_R = 0$) contribute to the elliptic genus in the Ramond sector. Consequently, the elliptic genus is a holomorphic function of $q$.

Given a Lagrangian of $\mathcal{N} = (0, 2)$ theory, the computation of the elliptic genus is straightforward [39, 40]. Let us review the contributions from different types of multiplets. The contribution of an $\mathcal{N} = (0, 2)$ chiral multiplet in a representation $\lambda$ of the gauge and flavor group is

$$\mathcal{I}_{\text{chi}}^{(0,2)}(q, z) = \prod_{w \in \lambda} i \frac{\eta(q)}{\vartheta_1(z^w)}, \tag{B.5}$$

where $w$ is a weight of the representation $\lambda$. The contribution of an $\mathcal{N} = (0, 2)$ Fermi multiplet in a representation $\lambda$ of the gauge and flavor group is given by

$$\mathcal{I}_{\text{fer}}^{(0,2)}(q, z) = \prod_{w \in \lambda} i \frac{\vartheta_1(z^w)}{\eta(q)}, \tag{B.6}$$

The contribution of an $\mathcal{N} = (0, 2)$ vector multiplet with gauge group $G$ is

$$\mathcal{I}_{\text{vec}}^{(0,2)}(q, z) = \frac{(-i\eta(q))^{2\,\text{rk}G}}{|W_G|} \prod_{\alpha \in \Delta} i \frac{\vartheta_1(z^\alpha)}{\eta(q)}. \tag{B.7}$$

where $\Delta$ is the set of roots of the gauge group $G$. Then, the elliptic genus of an $\mathcal{N} = (0, 2)$ quiver gauge theory can be schematically expressed as the Jeffrey-Kirwan (JK) residue integral [40–43]

$$\mathcal{I}^{(0,2)} = \oint_{\text{JK}} \prod_{\text{gauge}} \frac{dz}{2\pi i z} \mathcal{I}_{\text{vec}}^{(0,2)}(q, z) \prod_{\text{matter}} \mathcal{I}_{\text{chi}}^{(0,2)}(q, z) \mathcal{I}_{\text{fer}}^{(0,2)}(q, z). \tag{B.8}$$

Note that the notations and conventions are the same as in [7]. The Dedekind eta function is defined as

$$\eta(q) = q^{\frac{1}{24}} \prod_{n=1}^{\infty} (1 - q^n), \tag{B.9}$$

where $q = e^{2\pi i \tau}$ and $\text{Im}\,\tau > 0$. The Jacobi theta functions are defined by

$$\vartheta_1(z|q) := -i \sum_{r \in \mathbb{Z} + \frac{1}{2}} (-1)^{r - \frac{1}{2}} z^r q^{\frac{r^2}{2}}$$

$$= i q^{1/8} z^{1/2} \prod_{k=1}^{\infty} (1 - q^k)(1 - z^{-1}q^k)(1 - zq^{k-1}). \tag{B.10}$$

For the sake of brevity, the $q$ is often omitted, and we simply write $\vartheta_1(z)$.

For $\mathcal{N} = (2,2)$ theories, the elliptic genus in the Ramond-Ramond sector is given by

$$\mathcal{I}^{(2,2)}(q, y, z) = \text{Tr}_{RR}(-1)^F q^{H_L} \bar{q}^{H_R} y^{-J} \prod_a z_a^{f_a} , \tag{B.11}$$

where $J$ represents a left-moving $U(1)_R$ symmetry.

To understand the contributions of $\mathcal{N} = (2,2)$ multiplets, note that a $\mathcal{N} = (2,2)$ chiral multiplet can be decomposed into $\mathcal{N} = (0,2)$ chiral and Fermi multiplets. Consequently, the contribution of an $\mathcal{N} = (2,2)$ chiral multiplet with $R$-charge $r$ in a representation $\lambda$ of the gauge and flavor group is

$$\mathcal{I}^{(2,2)}_{\text{chi}}(q, y, z) = \prod_{w \in \lambda} \frac{\vartheta_1(y^{1-r/2}z^w)}{\vartheta_1(y^{-r/2}z^w)} . \tag{B.12}$$

In addition, an $\mathcal{N} = (2,2)$ vector multiplet consists of $\mathcal{N} = (0,2)$ vector and adjoint chiral multiplets. Thus, the contribution of an $\mathcal{N} = (2,2)$ vector multiplet with gauge group $G$ is

$$\mathcal{I}^{(2,2)}_{\text{vec}}(q, y, y, z) = \frac{1}{|W_G|} \left( \frac{\eta(q)^3}{\vartheta_1(y)} \right)^{\text{rk}G} \prod_{\alpha \in \Delta} \frac{\vartheta_1(z^\alpha)}{\vartheta_1(yz^\alpha)} , \tag{B.13}$$

where $|W_G|$ is the order of the Weyl group associated with the gauge group $G$.

Similarly, performing the JK residue integral, we obtain the elliptic genus of an $\mathcal{N} = (2,2)$ quiver gauge theory as

$$\mathcal{I}^{(2,2)} = \oint_{\text{JK gauge}} \prod \frac{dz}{2\pi i z} \mathcal{I}^{(2,2)}_{\text{vec}}(q, y, z) \prod_{\text{matter}} \mathcal{I}^{(2,2)}_{\text{chi}}(q, y, z) . \tag{B.14}$$

In this paper, we extensively use elliptic genera to check new dualities and trialities. However, the identities of elliptic genera presented here remain conjectural and are not rigorously proven. Instead, we verify these conjectures by explicitly evaluating elliptic genera using JK integrals up to rank five and performing $q$-expansions up to $\mathcal{O}(q^2)$.

# C   Some details on 2d $\mathcal{N} = (0,2)$ dualities

This appendix presents the detailed and technical aspects of the 2d $\mathcal{N} = (0,2)$ dualities discussed in §3. While the main text focuses on key results, some derivations, additional dualities, and further supporting evidence are presented here. Although these details are too technical for the main discussion, they are essential for an understanding of the underlying structure of $\mathcal{N} = (0,2)$ dualities.

## C.1   Twisted compactification of 4d Intriligator-Seiberg duality on $S^2$

We consider the twisted compactification of 4d $\mathcal{N} = 1$ Intriligator-Seiberg duality [24] on $S^2$, resulting in a 2d $(0,2)$ SO gauge theory dual to Landau-Ginzburg model. Our approach follows a method similar to that used in [9]. Tab. 7 summarizes the matter contents of the 4d $\mathcal{N} = 1$ Intriligator-Seiberg duality where the theory in the right has a superpotential

$$\mathcal{W} = \hat{Q}^i \hat{Q}^j M_{ij} . \tag{C.1}$$

Sym

Sym

$SO(N_c)$ ⟶ $N_c - 2$  ⟺  $SO(N_f - N_c + 4)$ ⟶ $N_f - N_c + 2$

Figure 19: A trivial 2d $\mathcal{N} = (0,2)$ duality obtained from the compactification of 4d $\mathcal{N} = 1$ Intriligator-Seiberg duality on $S^2$.

Upon compactification on $S^2$, we reassign non-negative integer $R$-charges $r_a$ to each fundamental chiral multiplet in the left theory and $\hat{r}_a$ to those in the right theory [10]. In 4d, the $U(1)_R \, SO(N_c) \, SO(N_c)$ mixed gauge anomaly can be calculated as

$$\mathrm{Tr}\, RGG = T_{SO}(\square) \sum_{a=1}^{N_f} (r_a - 1) + T_{SO}(\mathrm{adj}) = 0 \,. \tag{C.2}$$

This anomaly cancellation imposes the following constraint on the $R$-charges of the fundamental chirals:

$$\sum_{a=1}^{N_f} r_a = N_f - N_c + 2 \,. \tag{C.3}$$

Similarly, in the dual theory, the anomaly condition leads to:

$$\sum_{a=1}^{N_f} \hat{r}_a = N_c - 2 \,. \tag{C.4}$$

To satisfy these conditions, we assign $r_a = 1$ for $a = 1, \ldots, N_f - N_c + 2$ and $r_a = 0$ for the remaining indices. The dual $R$-charges are then given by $\hat{r}_a = 1 - r_a$. Consequently, the $R$-charges of the mesons in the symmetric representation of the dual theory are determined as:

$$r(M_{ab}) = 2 - \hat{r}_a - \hat{r}_b \,, \qquad a \leq b \,. \tag{C.5}$$

Following [10], upon compactification on $S^2$, the 2d $(0,2)$ chiral multiplets arise from 4d fields with integral $R$-charges less than 1, while 2d $(0,2)$ Fermi multiplets originate from 4d fields with integral $R$-charges greater than 1. 4d fields with $R$-charge exactly equal to 1 do not contribute to the 2d theory.

Thus, we arrive at the conclusion that the 2d $(0,2)$ $SO(N_c)$ gauge theory with $N_c - 2$ fundamental chirals is dual to the $SO(N_f - N_c + 4)$ gauge theory, which includes $N_f - N_c + 2$ fundamental chirals, $(N_c - 2)(N_c - 1)/2$ chiral mesons, and $(N_f - N_c + 2)(N_f - N_c + 3)/2$ Fermi mesons. After transferring the chiral mesons to the other side, we obtain the duality depicted in Fig. 19.

Table 7: Matter contents of 4d $\mathcal{N} = 1$ SO Intriligator-Seiberg duality.

|  | $SO(N_c)$ | $SU(N_f)$ | $U(1)_R$ |
|---|---|---|---|
| $Q$ | $\square$ | $\square$ | $\frac{N_f - N_c + 2}{N_f}$ |

|  | $SO(N_f - N_c + 4)$ | $SU(N_f)$ | $U(1)_R$ |
|---|---|---|---|
| $\hat{Q}$ | $\square$ | $\overline{\square}$ | $\frac{N_c - 2}{N_f}$ |
| $M$ | $\mathbf{1}$ | $\mathbf{Sym}$ | $\frac{2(N_f - N_c + 2)}{N_f}$ |

It is evident from Fig. 19 that the dual theories are independent of the rank of the gauge groups. By evaluating the elliptic genus

$$
\mathcal{I} = \frac{2\eta(q)^{\frac{3N-3\chi-2}{2}} \prod_{i \leq j} \vartheta_1(x^2 b_i b_j)}{2^{\lfloor (N-1)/2 \rfloor} \lfloor N/2 \rfloor!} \oint_{\mathrm{JK}} \frac{d\boldsymbol{a}}{2\pi i \boldsymbol{a}} \frac{\prod_{i<j}^{\lfloor N/2 \rfloor} \vartheta_1(a_i^{\pm} a_j^{\pm})}{\prod_{i=1}^{\lfloor N/2 \rfloor} \prod_{j=1}^{N-2} \vartheta_1(x a_i^{\pm} b_j)} \left( \frac{\prod_{i=1}^{\lfloor N/2 \rfloor} \vartheta_1(a_i^{\pm})}{\prod_{j=1}^{N-2} \vartheta_1(x b_j)} \right)^{\chi}
$$

$$
= \frac{\vartheta_1(x^{N-2})}{\eta(q)}, \tag{C.6}
$$

we conclude that the theory is dual to one free Fermi field.

By reorganizing the field content, we obtain the 2d $(0,2)$ version of the Intriligator-Seiberg duality between Sym-1 and Sym-3 in Fig. 8 (Sym-dual).

## C.2 Triality with SU($N$) flavor symmetry for odd $N$

We have identified a new triality, as illustrated in Fig. 6 (AS-dual), in which the flavor symmetry is SU($N$) with $N$ being even. Building on this discovery, we now present a closely related triality that arises when the flavor symmetry is SU($N$) with $N$ being odd. This new case exhibits analogous structures while introducing subtle differences in superpotentials due to the parity of $N$. We propose that the following three theories become equivalent in the infra-red.

AS-1'. Sp($N-1$) gauge theory with $N$ fundamental chirals $P$, one fundamental chiral $Q$, and no superpotential.

AS-2'. SU($N-2$) gauge theory with one anti-symmetric chiral $X$, $N$ fundamental chirals $Y$ and $N$ chiral mesons $Z$. Additionally, there is a neutral Fermi multiplet $\Psi$, forming a superpotential

$$
\mathcal{W} = \Psi(\epsilon^{\alpha_1 \cdots \alpha_{N-2}} X_{\alpha_1 \alpha_2} \cdots X_{\alpha_{N-4} \alpha_{N-3}} Y^i_{\alpha_{N-2}} Z_i). \tag{C.7}
$$

AS-3'. LG model of one Fermi $\Psi$, $N$ chiral mesons $Z$ and $\frac{1}{2}N(N-1)$ chirals, forming an anti-symmetric $N \times N$ matrix $A$ with a superpotential

$$
\mathcal{W} = \Psi(\epsilon^{i_1 \cdots i_N} A_{i_1 i_2} \cdots A_{i_{N-2} i_{N-1}} Z_{i_N}). \tag{C.8}
$$

The U(1)$_x$ and the U(1)$_y$ charges of the fields are given as follows:

|          | $P$ | $Q$ | $\Psi$ | $A$ | $X$ | $Y$ | $Z$ |
|----------|-----|-----|--------|-----|-----|-----|-----|
| U(1)$_x$ | 1   | 0   | $-N$   | 2   | $\frac{2N}{N-2}$ | $\frac{2}{N-2}$ | 1   |
| U(1)$_y$ | 0   | $-1$ | 1     | 0   | 0   | 0   | $-1$ |

$$\tag{C.9}$$

A straightforward verification shows that the central charges of these theories agree, which are given by

$$
c_L = (N+2)(N-1), \qquad c_R = \frac{3}{2}(N+2)(N-1). \tag{C.10}
$$

Furthermore, we verify that their elliptic genera agree

$$
\mathcal{I} = -\frac{\eta(q)^{(N^2+4N-5)/2}}{2^{\frac{N-1}{2}} \frac{N-1}{2}!} \oint_{\mathrm{JK}} \frac{d\boldsymbol{a}}{2\pi i \boldsymbol{a}} \prod_{i=1}^{\frac{N-1}{2}} \frac{\vartheta_1(a_i^{\pm 2}) \prod_{j<i} \vartheta_1(a_i^{\pm} a_j^{\pm})}{\prod_{j=1}^{N} \vartheta_1(a_i^{\pm} b_j x) \cdot \vartheta_1(a_i^{\pm} y^{-1})}
$$

$$
= \frac{\eta(q)^{\frac{N^2+7N-20}{2}} \vartheta_1(x^{-N} y)}{(N-2)! \prod_{i=1}^{N} \vartheta_1(b_i y^{-1} x)} \oint_{\mathrm{JK}} \frac{d\boldsymbol{a}}{2\pi i \boldsymbol{a}} \prod_{i=1}^{N-2} \frac{\prod_{j \neq i} \vartheta_1(a_i/a_j)}{\prod_{j<i} \vartheta_1(a_i a_j x^{\frac{2N}{N-2}}) \cdot \prod_{j=1}^{N} \vartheta_1(a_i b_j^{-1} x^{\frac{2}{N-2}})}
$$

$$
= \frac{\eta(q)^{(N^2+N-2)/2}}{\prod_{i<j} \vartheta_1(x^2 b_i b_j) \prod_{i=1}^{N} \vartheta_1(b_i y^{-1} x)} \frac{\vartheta_1(x^{-N} y)}{}. \tag{C.11}
$$

## C.3 SU gauge theory with one anti-symmetric and one fundamental chiral

In §3.4, we studied the dualities of SU($n$) gauge theories with one anti-symmetric chiral multiplet. For the duality in Fig. 12 to hold, the number of fundamental chirals must satisfy $N \geq 2$, as required by condition (53). Here, we consider the special case where the number of fundamental chirals is reduced to one, i.e., $N = 1$.

As illustrated in Fig. 21, a sequence of dualities leads to an equivalence between an SU($n$) gauge theory and an Sp($n + m - 3$) gauge theory, where $m$ is either 0 or 1 such that $n + m$ is odd. The matter content and superpotentials for both theories are detailed below.

- **Theory A:** an SU($n-1$) gauge theory with the following matters:

    – One anti-symmetric chiral $Y$.

    – One fundamental chiral $X_2$.

    – $n$ anti-fundamental chirals $X_1$.

    – One free Fermi $\Psi$ that transforms as $(1-n, m)$ under the symmetries U(1).$_0 \times$ U(1)$_1$.

    – A Fermi meson $\Gamma$.

The superpotential of the theory takes the following form:

$$\mathcal{W} = \begin{cases} \Psi \operatorname{Pf} Y + \operatorname{Tr} \Gamma X_1 X_2, & \text{when } n \text{ is odd,} \\ \Psi \epsilon^{\alpha_1 \ldots \alpha_{n-1}} Y_{\alpha_1 \alpha_2} \ldots Y_{\alpha_{n-3} \alpha_{n-2}} X_{2, \alpha_{n-1}} + \operatorname{Tr} \Gamma X_1 X_2, & \text{when } n \text{ is even.} \end{cases} \tag{C.12}$$

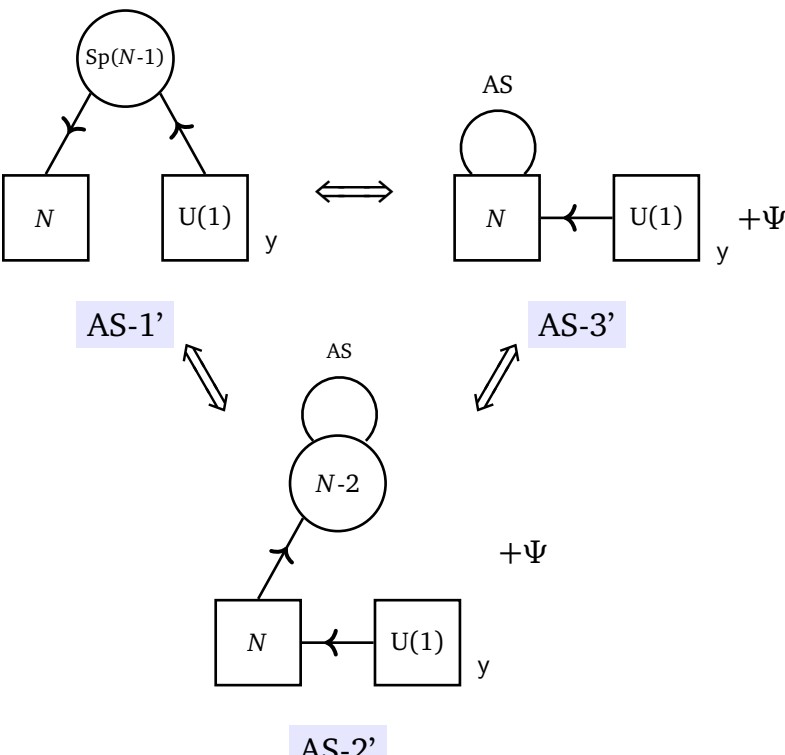

Figure 20: A triality among three distinct theories: (AS-1') an Sp($N-1$) gauge theory of SQCD type with $N+1$ fundamental chirals, (AS-2') an SU($N-2$) gauge theory with one anti-symmetric, $N$ fundamental chirals and $N$ chiral mesons, coupled to a neutral Fermi multiplet $\Psi$, and (AS-3') a Landau-Ginzburg model consisting of $\frac{1}{2}N(N-1)$ chiral multiplets and $N$ chiral mesons coupled to a Fermi multiplet $\Psi$.

- **Theory B:** an $\mathrm{Sp}(n+m-3)$ gauge theory, where $m$ is either zero or one such that $n+m$ is odd, with the following matters:

  - One fundamental chiral $\widetilde{X}_1$ with charge $(\square, \square, 1)$ under the flavor symmetries $\mathrm{Sp}(n+m-3) \times \mathrm{SU}(n) \times \mathrm{U}(1)_2$.
  - One fundamental chiral $\widetilde{X}_2$ with charge $(\square, -m)$ under the symmetries $\mathrm{Sp}(n+m-3) \times \mathrm{U}(m)_1$.
  - One chiral meson $Z$ that charges $(\overline{\square}, 1-n, n-1)$ under $\mathrm{SU}(n) \times \mathrm{U}(1)_0 \times \mathrm{U}(1)_2$.
  - One fundamental Fermi $\Lambda$ with charge $(\square, n-1, -n)$ under $\mathrm{Sp}(n+m-3) \times \mathrm{U}(1)_0 \times \mathrm{U}(1)_2$.
  - One Fermi meson $\widetilde{\Gamma}$ with charge $(\overline{\square}, 1, -1)$ under $\mathrm{SU}(n) \times \mathrm{U}(m)_1 \times \mathrm{U}(1)_2$.
  - One single free Fermi $\widetilde{\Psi}$ with charge $(n-1, 1, -n)$ under $\mathrm{U}(1)_0 \times \mathrm{U}(1-m)_1 \times \mathrm{U}(1)_2$.

The superpotential of theory B is given by:

$$\mathcal{W} = \mathrm{Tr}\, \widetilde{\Gamma} \widetilde{X}_1 \widetilde{X}_2 + \mathrm{Tr}\, \Lambda \widetilde{X}_1 Z \,. \tag{C.13}$$

The central charges agree as

$$c_L = n^2 - n - 2\,, \qquad c_R = \frac{3}{2}(n^2 - n - 2)\,. \tag{C.14}$$

Moreover, the elliptic genera agree and are expressed as follows:

$$\mathcal{I}_A = \frac{\eta(q)^{\frac{n^2+5n-14}{2}}}{(n-1)!} \oint_{\mathrm{JK}} \frac{d\boldsymbol{a}}{2\pi i \boldsymbol{a}} \prod_{i=1}^{n-1} \frac{\prod_{j\neq i} \vartheta_1(a_i/a_j)}{\prod_{j<i} \vartheta_1(x_0^2 a_i a_j) \cdot \prod_{j=1}^{n} \vartheta_1(a_i^{-1} c_j x_2 x_0^{-1}) \cdot \vartheta_1(a_i x_1^{-1} x_0)}$$

$$\times \vartheta_1(x_0^{1-n} x_1^m) \prod_{i=1}^{n} \vartheta_1(c_i^{-1} x_1 x_2^{-1})\,, \tag{C.15}$$

$$\mathcal{I}_B = \frac{\eta(q)^{N_B(N_B+6)/2}}{2^{\frac{N_B}{2}} \frac{N_B}{2}!} \oint_{\mathrm{JK}} \frac{d\boldsymbol{g}}{2\pi i \boldsymbol{g}} \prod_{i=1}^{\frac{N_B}{2}} \frac{\vartheta_1(g_i^{\pm 2}) \prod_{j<i} \vartheta_1(g_i^{\pm} g_j^{\pm}) \cdot \vartheta_1(g_i^{\pm} y^{-1})}{\prod_{j=1}^{n} \vartheta_1(x_2 g_i^{\pm} c_j) \cdot \vartheta_1(g_i^{\pm} x_1^{-1})^m}$$

$$\times \frac{\vartheta_1(y^{-1} x_1)^{1-m} \prod_{i=1}^{n} \vartheta_1(x_1 x_2^{-1} c_i^{-1})^m}{\eta(q)^{(m-1)(n-1)} \prod_{i=1}^{n} \vartheta_1(y c_i^{-1} x_2^{-1})}\,, \tag{C.16}$$

where $N_B = n+m-3$ and $y = x_1^{-(n-1)} x_2^n$.

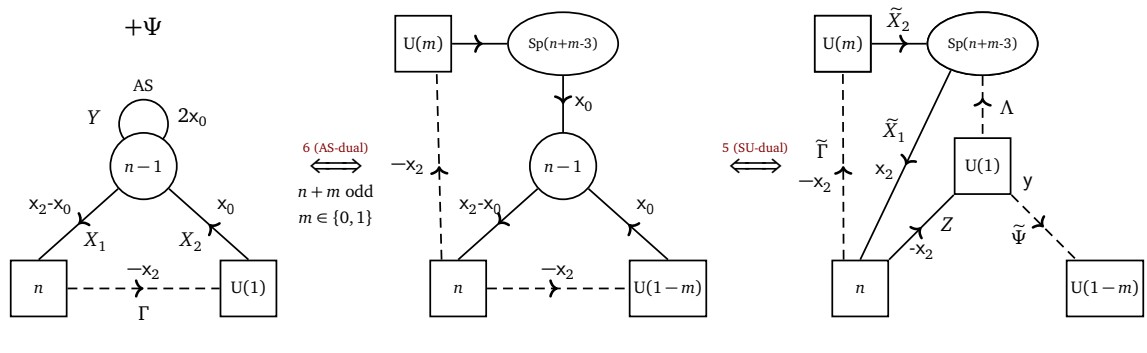

Figure 21: A duality between $\mathrm{SU}(n)$ and $\mathrm{Sp}(n+m-3)$ gauge theories, where $m$ is either 0 or 1 such that $n+m$ is odd. The transformation between Theory A and Theory B follows AS-dual and Sym-dual, as indicated. The chemical potential is given by $y = nx_2 - (n-1)x_1$.

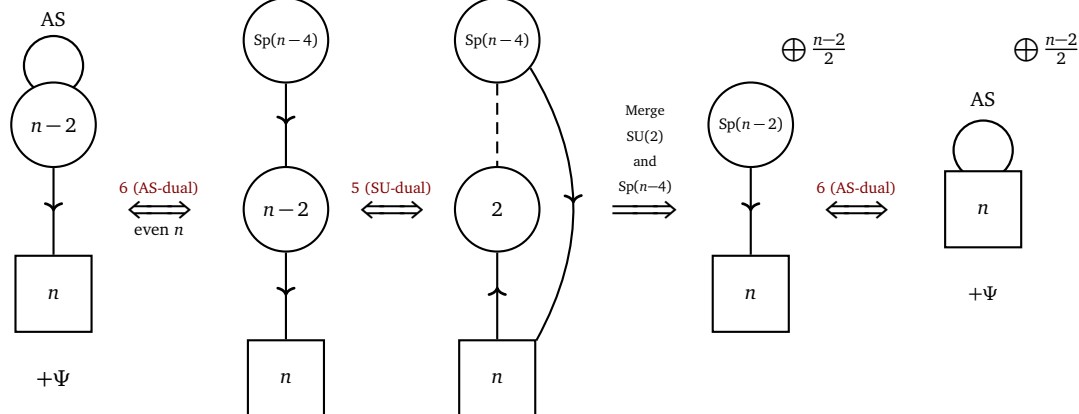

Figure 22: A sequence of transformations applied to the SU($n-2$) gauge theory with one anti-symmetric chiral and $n$ anti-fundamental chirals where $n$ is even. In the transition from the third to the fourth diagram, the Sp($n-4$) and SU(2) gauge nodes are merged into a single Sp($n-2$) node. This merging introduces an additional factor of $(n-2)/2 = |W(\text{Sp}(n-2))|/|W(\text{Sp}(n-4)) \times W(\text{SU}(2))|$, reflecting the ratio of Weyl group orders in the elliptic genus. Additionally, at this step, the direction of the arrow between nodes 2 and $n$ is reversed from $(2 \leftarrow n)$ to $(2 \rightarrow n)$, utilizing the identity $\vartheta_1(z) = -\vartheta_1(1/z)$. For the 4th and 5th quiver diagrams, we consider a disjoint union of $(n-2)/2$ copies of the theory. Through these transformations, we establish an identity between the elliptic genera at each step of the sequence.

## C.4  SU gauge theory with one anti-symmetric and no fundamental chiral

Finally, we examine the case where there is no fundamental chiral multiplet, i.e., $N = 0$. This corresponds to an SU($n-2$) gauge theory with one anti-symmetric chiral multiplet and $n$ anti-fundamental chirals. One approach to studying this case is to directly compute the elliptic genus using the JK residue integrals. Alternatively, we can derive the result through a sequence of known dualities and operations.

In this case, we consider transformations that provide the identities of the elliptic genus, focusing only on the field contents while ignoring detailed aspects of the theory, such as the superpotential. As before, this depends on whether $n$ is even or odd: we apply the duality in Fig. 6 (AS-dual) or Fig. 20 to the anti-symmetric chiral multiplet, followed by a sequence of additional dualities and transformations.

Let us consider the case of even $n$, and Fig. 22 illustrates a series of transformations that provides the identity of elliptic genera. Based on this identity, we propose the duality between the following two theories:

- An SU($n-2$) gauge theory with one anti-symmetric chiral $X$, $n$ anti-fundamental chirals $Y$, and a neutral Fermi multiplet $\Psi$. The superpotential is given by:

$$\mathcal{W} = \Psi \,\text{Pf}\,X \,. \tag{C.17}$$

- A disjoint union of $\frac{n-2}{2}$ copies of a theory with $\frac{1}{2}n(n-1)$ chirals, forming an anti-symmetric $n \times n$ matrix $A$, and a Fermi multiplet $\Psi$, with the superpotential:

$$\mathcal{W} = \Psi \,\text{Pf}\,A \,. \tag{C.18}$$

The U(1)$_x$ charges of the fields are summarized as follows,

$$
\begin{array}{c|cccc}
 & X & Y & \Psi & A \\
\hline
\text{U(1)}_x & \frac{2n}{n-2} & -\frac{2}{n-2} & -n & 2
\end{array} \,. \tag{C.19}
$$

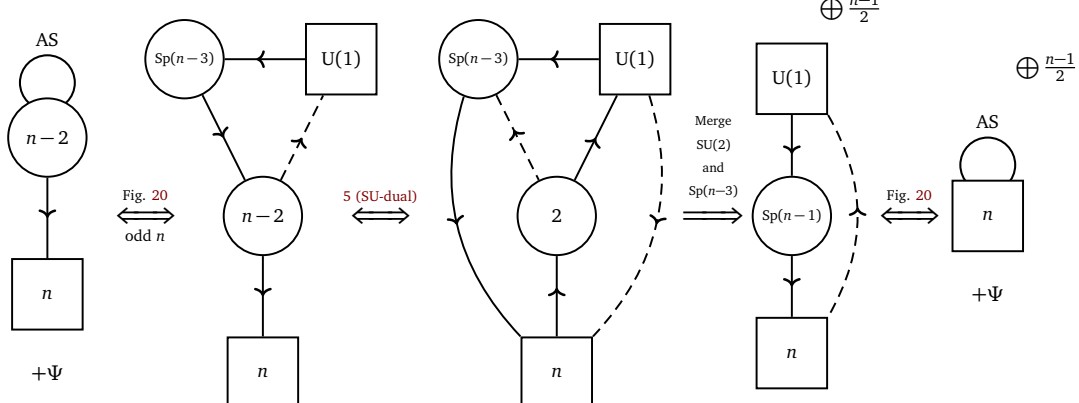

Figure 23: A sequence of transformations applied to the SU($n-2$) gauge theory with one anti-symmetric chiral and $n$ anti-fundamental chirals where $n$ is odd. In the transition from the third to the fourth diagram, the Sp($n-3$) and SU(2) gauge nodes are merged into a single Sp($n-1$) node. This merging introduces an additional factor of $(n-1)/2 = |W(\text{Sp}(n-1))|/|W(\text{Sp}(n-3)) \times W(\text{SU}(2))|$, reflecting the ratio of Weyl group orders in the elliptic genus. Additionally, at this step, the direction of the arrow between nodes 2 and $n$ is reversed from $(2 \leftarrow n)$ to $(2 \rightarrow n)$, utilizing the identity $\vartheta_1(z) = -\vartheta_1(1/z)$. For the 4th and 5th quiver diagrams, we consider a disjoint union of $(n-1)/2$ copies of the theory. Through these transformations, we establish an identity between the elliptic genera at each step of the sequence.

The elliptic genera are given by

$$
\begin{aligned}
\mathcal{I} &= -\frac{\eta(q)^{\frac{n^2+5n-20}{2}}\vartheta_1(x^{-n})}{(n-2)!} \oint_{\text{JK}} \frac{d\boldsymbol{a}}{2\pi i \boldsymbol{a}} \prod_{i=1}^{n-2} \frac{\prod_{j \neq i} \vartheta_1(a_i/a_j)}{\prod_{j<i}\vartheta_1(x^{\frac{2n}{n-2}}a_i a_j) \cdot \prod_{j=1}^{n}\vartheta_1(a_i^{-1}b_j x^{-\frac{2}{n-2}})} \\
&= \frac{n-2}{2}\frac{\vartheta_1(x^{-n})}{\eta(q)}\prod_{i<j}\frac{\eta(q)}{\vartheta_1(x^2 b_i b_j)}.
\end{aligned}
\tag{C.20}
$$

Now we turn to the case of odd $n$, and Fig. 23 illustrates a sequence of transformations that provide the identities of the elliptic genus. Based on this, we propose a duality between the following two theories

- An SU($n-2$) gauge theory with one anti-symmetric chiral $X$, $n$ anti-fundamental chirals $Y$, and one Fermi multiplet $\Psi$. The superpotential is given by:

$$
\mathcal{W} = \Psi \det X. \tag{C.21}
$$

- A disjoint union of $\frac{n-1}{2}$ copies of a theory with $\frac{n(n-1)}{2}$ chiral multiplets $A$, and one Fermi multiplet $\Psi$. The superpotential is given by:

$$
\mathcal{W} = \Psi \det A. \tag{C.22}
$$

The U(1)$_x$ and U(1)$_d$ charges of the fields are summarized as follows:

$$
\begin{array}{c|cccc}
 & \Psi & X & Y & A \\
\hline
\text{U}(1)_x & -2n & \frac{2n}{n-2} & -\frac{2}{n-2} & 2
\end{array}. \tag{C.23}
$$

Therefore, under the duality, there is a correspondence between the operator $\Psi(\det B)^2$ and $\Psi$. The elliptic genera are given by:

$$
\begin{aligned}
\mathcal{I} &= -\frac{\eta(q)^{\frac{n^2+5n-20}{2}}\vartheta_1(x^{-2n})}{(n-2)!} \oint_{\text{JK}} \frac{d\boldsymbol{a}}{2\pi i \boldsymbol{a}} \prod_{i=1}^{n-2} \frac{\prod_{j\neq i}\vartheta_1(a_i/a_j)}{\prod_{j<i}\vartheta_1(a_i a_j x^{\frac{2n}{n-2}}) \cdot \prod_{j=1}^{n}\vartheta_1(a_i^{-1} b_j x^{-\frac{2}{n-2}})} \\
&= \frac{n-1}{2}\frac{\vartheta_1(x^{-2n})}{\eta(q)} \prod_{i<j}\frac{\eta(q)}{\vartheta_1(x^2 b_i b_j)}.
\end{aligned}
\tag{C.24}
$$

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
