# Peer review of "d dualities from 4d"

_SciPost Physics, doi:SciPost Phys. 18, 180 (2025)_

## Round 1 · Referee Report · Anonymous (Referee 1) · 2024-9-13

Report

The manuscript investigates new dualities for two-dimensional supersymmetric gauge theories, with either $\mathcal{N}=(2,2)$ or $\mathcal{N}=(0,2)$ supersymmetry. This is achieved by studying the compactification of mainly 4d N=2 SCFTs, but also the 4d N=1 Seiberg duality, on a two-sphere with a suitable topological twist. The results of this manuscript enrich our current understanding of the dynamics of 2d SQFTs and their connection with four-dimensional physics. Notably, the perspective of the 4d to 2d compactification allowed the authors to discover a new family of 2d N=(2,2) theories labelled by Riemann surfaces with punctures and to show that these enjoy non-trivial infrared dualities, which can be nicely understood from the 4d perspective as different degeneration limits of the two-dimensional surface. Although the results of this manuscript are already interesting on their own, the topic demands for further study and I encourage the authors to pursue this line of research. For these reasons, I recommend this paper for publication after the next more specific comments are addressed.

  • At the beginning of section 2 the authors review the field content of the 2d theories obtained by compactifying a 4d N=1 theory on $S^2$ with a twist by a specific choice of R-symmetry. However, there is a subtlety discussed in Ref. [26] that the authors did not mention. In general the result of a 4d gauge theory will lead to a direct sum of distinct 2d theories labelled by the value of the gauge flux through the $S^2$. However, as discussed in [26], one can obtain a single 2d theory if the R-symmetry chosen for the twist is such that all chiral fields have non-negative R-charge, and not just integral. Since this is a crucial assumption for the construction used in the present manuscript, I think the authors should state it.

  • I think the $U(1)_c$ symmetry of the minimal puncture of the 4d trinion theory considered on page 3 corresponds to the $U(1)_f$ involved in the topological twist from the general discussion on the previous page. If that is the case, I would suggest the authors to state this more clearly.

  • I would also suggest to add a reminder of how the 2d N=(2,2) left and right moving R-symmetries (or equivalently the vector and axial R-symmetries) are embedded inside the 4d R-symmetry and how the fields transform under them. This would make more clear the expressions for the 2d central charges and the elliptic genus. In particular, it is not clear in (2.1) and (2.2) which of these symmetries do the fugacities q and y correspond to.

  • I would suggest the authors to add a reference to the Appendix A when calculating the central charges, especially in the first few examples considered in (2.5) and (2.7). Moreover, some of the theories considered in the manuscript possess abelian flavor symmetries (the $U(1)_{c_j}$) that in principle might mix with the R-symmetry in the IR, so I would add a brief explanation as to why this does not happen. In the case of the torus with one puncture for example this is due to the extended supersymmetry. What about the case of a torus with multiple punctures?

  • Above (2.3) I would specify that the puncture of this model is minimal. Moreover, in the theory (B) of the duality stated at the beginning of Section 3 I would specify that M is a chiral field.

  • I would suggest to expand more on why in the computation of the central charges around (3.2)-(3.3)-(3.4) one should set the R-charges of the Fermi fields to 1 and those of the chirals to 0. The authors correctly say that the issue is the non-compactness of the target space, but I would expand a bit more on this since it might not be so immediate to most readers. The point is that the R-symmetry cannot mix with the flavor symmetries associated to the non-compact directions. This is done by identifying which chiral operators parametrize such directions and setting their R-charges to zero. Since such operators are constructed from all the chirals in the two theories, we then need to set their R-charges to zero. The R-charge of the Fermi field $\Psi$ is consequently set to one by requiring that the superpotential has R-charge 1.

  • An additional check of the duality of Section 3 is the matching the anomalies for possible finite symmetries in the theory. In fact, there is a symmetry acting with charge 1 on all the chirals of theory (A) which is broken by anomalies to a finite group. It should be possible to identify such symmetry also in theory (B) and to match the anomalies for it. See Ref. [21] for a similar discussion.

  • Below (3.7) the authors state that the agreement of the elliptic genera has been checked through a power expansion in $q$. It would be good to specify for which values of the parameters $N_+$, $N_-$, $N_1$, $N_2$ this has been done and also to which order in $q$. Similarly for similar checks that have been done in the rest of the manuscript. Moreover, what does it mean that they have been matched “up to sign”? Do the authors have an understanding of this sign?

  • Is the statement below (3.11) that the $SU(N_c)$ theories do not enjoy the triality completely correct? Indeed, in Ref. [26] it has been shown that the duality corresponding to $N_2\leftrightarrow N_3$ is still valid and that it can be obtained from the reduction of the Seiberg duality. The point is that in the $SU(N_c)$ case we only get a duality and not a triality.

  • In Section 4, the authors identify many $N=(0,2)$ GLSM’s that are IR dual to LG models. The latter consist of chiral fields with some superpotential interaction, however the authors do not identify the superpotentials for all of the proposed dualities, but only for some. I think the authors should add the missing ones.

  • I suspect some of the dualities in Section 4 can be obtained from $S^2$ reduction of the 4d $N=1$ S-confining dualities classified in arXiv:hep-th/9612207. For example, the duality for $Sp(N)+(2N+2)F$ was obtained in Ref. [26] as the dimensional reduction of a duality due to Intriligator and Pouliot arXiv:hep-th/9505006. Similarly, I suspect that the duality for $SO(N)+(N-2)F$ can be obtained from dimensional reduction of the Intriligator—Seiberg duality arXiv:hep-th/9503179, while those for the theories with antisymmetric matter as reduction of the S-confining dualities that can be found in the paper mentioned above. It would be really interesting if the authors could recover the dualities of this section in this way, however I do not consider it as necessary for publication.

  • At the end of page 17 the authors state that when the target space is non-compact, then central charge can be computed as three times the complex dimension of the moduli space. It would be good if the authors could expand more on this point, specifying under which assumptions this is true. A reference for the stated fact would also be good.

I have also found a couple of typos:

  • Above (2.2): “form” -> “from”.

  • At the end of second to last paragraph on page 11: “…obtained from of the Lagrangian…”, remove “of”.

Recommendation

Ask for minor revision

  • validity: top
  • significance: top
  • originality: high
  • clarity: high
  • formatting: perfect
  • grammar: excellent

Author:  Satoshi Nawata  on 2025-02-17  [id 5225]

(in reply to Report 1 on 2024-09-13)

Thank you very much for your detailed and thoughtful comments on our manuscript. We greatly appreciate your insights and have incorporated your suggestions, as well as those from the other referees, into a major revision of the draft. Below, we summarize the key changes and clarifications made in response to your comments:

  • At the beginning of Section 2, we have added the following sentence to clarify the role of gauge magnetic flux:
    “If these charges are non-negative integers, we can focus on the vanishing sector of the gauge magnetic flux.”

  • Below Table 1, we have included an additional sentence to clarify the connection between ( U(1)_c ) and ( U(1)_f ) in the (\mathcal{N}=(2,2)) reduction:
    “In the (2,2) reduction, ( U(1)_c ) plays the role of the ( U(1)_f ) introduced earlier, used in the topological twist.”

  • We acknowledge that we inadvertently missed addressing one of the referee’s comments regarding the identification of symmetries. To clarify, in the 4d theory, ( U(1)_{-R+f/2} ) corresponds to the ( U(1)_V ) vector R-symmetry in 2d, while ( U(1)_r ) in 4d is identified with ( U(1)_A ) in 2d. We sincerely apologize for the oversight and will explicitly incorporate this clarification in the next version.

  • Below equations (2.5), (2.7), and (2.11), we have added a brief explanation of how the corresponding central charges are computed. We also clarify that these central charges do not mix with abelian flavor symmetries, following the same reasoning as in (0,2) theories, due to the presence of non-compact moduli spaces.

  • Above equation (2.3), we now explicitly state:
    “Consider the theory of genus one with one minimal puncture.”

  • The remaining questions regarding 2d (0,2) dualities have been addressed through a major revision of the manuscript, in which we have significantly expanded our discussion.

  • Regarding the reduction of Seiberg duality with SU gauge groups, we are grateful to the referee for pointing out the previously discovered duality in Gadde-Razamat-Willet [arXiv:1506.08795]. The duality presented in Section 3.1 was originally identified in their work, which we regrettably overlooked in the first version of our paper. To properly acknowledge their contribution, we have added Footnote 2 on page 8 to clarify the attribution. The purpose of Section 3.1 is to highlight key subtleties and provide additional insights into this duality.

  • In Section 3.1, we further elaborate on the target geometry of the non-linear sigma model in the infrared, as given in equations (3.7) and (3.8), emphasizing its non-compact nature. Following this, we explain the reason why the ( U(1)_R )-charge cannot mix with other abelian flavor symmetries is due to the non-compactness. Consequently, we set the R-charges of chiral multiplets to zero while assigning those of Fermi multiplets to one. We mention below (3.9) that the right-moving central charge is three times the complex dimension of the targe space. We hope that this revised explanation makes the argument clearer for readers.

  • As requested by the referee, we have included an explanation of the twisted compactification of 4d (\mathcal{N}=1) Intriligator-Seiberg duality in Appendix C.1, which leads to a duality between (0,2) SO gauge theory and a Landau-Ginzburg model. Furthermore, we extend this duality to a new triality in Section 3.2. We also specify the superpotential in all the Landau-Ginzburg models to ensure clarity. We hope this addresses the referee’s question satisfactorily.

Once again, we sincerely appreciate the referee’s valuable feedback, which has helped improve the manuscript significantly. We hope that our revisions adequately address all concerns.

---

## Round 1 · Referee Report · Bruno Le Floch (Referee 2) · 2024-9-21

Strengths

1- New two-dimensional dualities between non-abelian gauge theories with low (0,2) supersymmetry: SU(N) analogues of known U(N) trialities.

2- Derivation of these dualities from 4d N=1 Seiberg duality.

3- Proposals for new dualities between (2,2) quiver theories based on twisted reductions of class S theories.

4- The strategy is sound and well-explained.

Weaknesses

1- When discussing dualities between 2d theories with non-compact vacuum moduli space, there is a well-known order of limits issue regarding whether the IR limit is taken at a fixed place in the vacuum moduli space, or going to infinity on it (see https://arxiv.org/abs/1611.02763 by Aharony-Razamat-Seiberg-Willett). This order of issue is not considered in this paper, making it difficult to know the precise meaning of the proposed dualities in the non-compact cases.

2- In section 2 the absence of superpotential seems wrong, especially given that in the explicit theory with $g=n=1$, there is a non-trivial superpotential (2.3).

3- The (2,2) dualities in section 2 only have quite weak evidence. These dualities are simply consequences of abelian gauging in a hypothetical duality between the $g=1$ theory and $n$ copies of the $g=n=1$ theory, and the authors do not explain why they propose the dualities with abelian gauging as dualities, but not the ungauged version of the duality.

4- Throughout the paper, equalities of elliptic genera are checked up to some unspecified order in the $q$-expansion, and for unspecified gauge group ranks, rather than making attempts at obtaining them analytically, or at least specifying how much they have been tested.

5- The method used in section 3, together with more elaborate R-charge assignments, should give a triality symmetry, not just a duality.

6- In section 4 the central charges etc are given without explanations, so that it is not clear how rigorously they are obtained.

Report

The authors approach of twisted reduction of 4d dualities down to 2d is sound, and I find particularly interesting their 2d (0,2) duality derived from Seiberg duality. The paper constitutes good work.

That said, given how selective SciPost physics is, I would suggest seeking another venue for publishing this article, such as JHEP, EPJC, etc.

Requested changes

1- Introduction. 'Through the computation of elliptic genera, we explicitly demonstrate that these theories are independent of the frame or complex structure of the Riemann surface.' The elliptic genus being an index, it cannot be sensitive to the complex structure, so the calculation does not show independence of the theory itself. It might be possible to argue that deformations of the complex structure are irrelevant hence do not affect the IR limit of the 2d theories, or at least check that the deformations are suitably Q-exact in the 2d theory so that they do not affect observables of interest (beyond just the elliptic genus). Maybe this Q-exactness is established in https://arxiv.org/abs/1703.08201 by Amariti-Cassia-Penati, but I didn't read that paper.

2- Eq (2.3). Equation (2.3) includes a ${\cal N}=(2,2)$ superpotential term $Tr[\phi,\phi^\dagger]^2$, and this contradicts the comment below (2.2) stating that the authors do not include any ${\cal N}=(2,2)$ superpotential. In these kinds of 2d ${\cal N}=(2,2)$ quiver theories I would expect a cubic superpotential coupling bifundamental chiral multiplets that are charged under the same gauge group. But it is not clear whether such a superpotential indeed arises in the twisted dimensional reduction. Cf comment about eq (2.8) below.

3- Eq (2.6). In equation (2.6), the second line is obtained by picking up poles in the integral; this could usefully be mentioned (not necessarily detailed). It is not clear why the ratio of four theta function gets rewritten as a larger product of factors, because it is not clear what is special about the list of scalars $Tr(\phi^i)$, $Tr(\sigma^i)$, $Tr((\phi\sigma)^{i-1})$. What about a more general $Tr(\phi^i\sigma^j)$? Is it eliminated by an F-term condition?

4- Eq (2.8). In analogy to the last expression in (2.6) it seems important to figure out which chiral ring generators contribute which factor in the product. To understand why other (gauge-invariant) polynomials in the fields $\phi_i$ and $\sigma_i$ do not contribute, the F-term relations are probably essential, which forms a good test of what the superpotential could be.

5- Eq (2.11). I disagree with the logic of saying that the equality of elliptic genera suggests the two theories are dual. Following the same logic, (2.8) would suggest that the quiver of Figure 2 is dual to $n$ decoupled theories. Is that the case? I'm especially uncertain about these 2d dualities because the target space metric is important as explained in https://arxiv.org/abs/1611.02763 by Aharony-Razamat-Seiberg-Willett.

6- Eq (3.7). If the sign is an overall sign, write $I_A=-I_B$ I suppose; if it is an undetermined sign, maybe say ``up to an undetermined overall sign''. It would be good to specify to which order in $q$ the identity has been checked. Same comment for (2.8), and which values of $N$ and $n$ as well. Same comment for footnote 1.

7- Section 3. Triality usually involves theories with three flavour symmetry groups $SU(N_i)$, corresponding to Fermi, fundamental chiral, and antifundamental chiral multiplets (plus determinant matter, but that is irrelevant in the $SU$ case). The gauge group is $U((N_1+N_2+N_3)/2 - N_a)$ in the $a$-th theory of the triality, see https://arxiv.org/abs/1310.0818 (by Gadde-Gukov-Putrov) figure 2. The natural guess for an $SU$ version is to throw away the determinant matter but keep the three kinds of matter fields. Here the authors only consider two kinds of matter, which amounts to taking $N_3=0$ in the triality. Then it is clear that one of the three gauge group ranks becomes negative, which is enough to explain why there is no third theory in the triality. An obvious question then is: is there an $SU$ triality with all three kinds of matter? The obvious attempt is to change (3.8) to a more general charge assignment $r=(0,...,0,1,...,1,2,...,2)$ for $Q$ and $\tilde{Q}$, but maybe anomaly cancellation forbids this?

8- Equation (4.5) is not consistent with the fact that $X$ is antisymmetric and $Y$ fundamental. Presumably the twisted compactification should give anti-fundamental (or anti-antisymmetric, by which I mean $\Lambda^2\overline{\Box}$) chiral multiplets so that $YXY$ can be gauge-invariant. This problem can also be seen when trying to show the identity (4.4) as a result of picking up certain residues: as written, the $1/\vartheta_1(a_kb_l)$ factor should have poles at $a_k=1/b_l$ so that $1/\vartheta_1(a_ia_j)$ would become $1/\vartheta_1(1/(b_ib_j))$, but if we replace fundamentals by antifundamentals, then the contribution of $Y$ becomes $1/\vartheta_1(b_l/a_k)$ so that poles are at $a_k=b_l$ which correctly leads to $1/\vartheta_1(b_ib_j)$ factors. Also, (4.5) is antisymmetric in $i,j$ so I think there are only $(N+1)(N+2)/2$ chirals, not twice that number. The same counting issue shows up for the next model, where there should be $(N-2)(N-1)/2$ instead of $(N-2)(N-1)$.

9- Below (4.8), it is proposed that the $\vartheta_1$ factor in the numerator is due to a dynamically-generated Fermi multiplet. This requires more justification. My guess would be that it is given by $(\Lambda^{n-1}X){\alpha\beta}(\Lambda^{n-2}Y)^{\beta\gamma}\Upsilon^\alpha{}\gamma$ where $\Lambda$ denotes antisymmetrization, so that $\Lambda^{n-1}X$ lives in the representation with Young diagram consisting of two columns of $n-1$ boxes each, while $\Lambda^{n-2}Y=\bigwedge_{i=1}^{n-2}Y^i$ lives in the conjugate of the antisymmetric representation, and where $\Upsilon$ is the gauge field strength Fermi multiplet, in the adjoint representation of $SU(N)$. The guess could be checked by tracking down where the numerator $\vartheta_1$ comes from in a vector multiplet one-loop determinant, and by checking which JK residue does give the final numerator $\vartheta_1(x^{N-1}\prod_{i=1}^{N-2}b_i)$.

10- From the dual LG point of view, the fact that fugacities $b_ib_j$ appear, and not more general fugacities $c_{ij}$, deserves explanations. Usually, such fine-tuning of meson/hadron flavour charges comes from a superpotential coupling, so I would expect a superpotential coupling like (4.11), (4.14), (4.17) also in the models discussed around (4.5) and (4.8). I don't immediately guess the correct superpotential for these two models. Relatedly, (4.17) does not fix all fugacities, namely does not break enough symmetries, so that the LG model has too much flavour symmetry: indeed, the superpotential does not involve the $N-2$ fields $\Phi_{ii}$, which are thus free. Maybe the explanation in all these cases is that some of the chiral multiplets become free in the IR?

11- Appendix A. Concerning c-extremization, I think (without proof) that in the case of a non-compact vacuum moduli space, the chiral fields spanning non-compact directions must have zero R-charge, but some other directions could be compact, in which case we can probably perform c-extremization on them. Correspondingly, the central charge is a sum of the non-compact dimensions and a compact contribution obtained by extremization. This might be discussed in the original c-extremization papers.

12- Misc comments. It would be interesting to consider also quiver tails that arise with certain patterns of puncture data in class S theories. It might make sense to cite https://arxiv.org/abs/1609.07144 by Franco-Lee-Seong, but I am not sure: it seems to exhibit trialities of 2d (0,2) quiver theories as well, but only Abelian ones I think.

Recommendation

Ask for major revision

  • validity: high
  • significance: good
  • originality: ok
  • clarity: top
  • formatting: good
  • grammar: perfect

Author:  Satoshi Nawata  on 2025-02-17  [id 5226]

(in reply to Report 2 by Bruno Le Floch on 2024-09-21)

Thank you very much for your detailed and thoughtful comments on our manuscript. We sincerely appreciate your insights and have carefully incorporated your suggestions, as well as those from the other referees, into a major revision of the draft. Below, we summarize the key changes and clarifications made in response to your comments:

  1. The phrase "complex structure of the Riemann surface" was misleading, as we do not identify the gauge coupling with the complex structure of the Riemann surface. Therefore, we have removed this phrasing. As explained below Figure 4, the two theories in the figure have different Lagrangian descriptions. Consequently, the agreement of their elliptic genera serves as a non-trivial check of our proposal.

  2. Thank you for pointing this out. Indeed, Equation (2.3) does not describe a superpotential but rather a scalar potential, which exists in any $\mathcal{N}=(2,2)$ Lagrangian theory, even in the absence of a superpotential. To avoid confusion, we have revised the text to refer to it as “the potential for scalar fields.”

  3. Below Equation (2.6) for the elliptic genus, we mention that the coordinate ring of the target space $(\mathfrak{t} \times \mathfrak{t}) / S_N$ is generated by $\text{Tr}(\phi^i \sigma^j)$, as noted by the referee. However, we are unclear about the precise relation between these generators and, therefore, do not fully understand the cancellation mechanism between the numerator and denominator that results in such a simple expression for the elliptic genus.

  4. Since we do not yet fully understand the case of genus one with a single puncture, we are currently unable to extend our understanding to more general cases, such as genus one with multiple punctures. However, explicit computations of the elliptic genus suggest a TQFT-like structure, where contributions arise only from punctures. This remarkably simple structure of elliptic genera is quite intriguing, but the underlying mechanism remains unclear.

  5. As mentioned in Point 1, the agreement of the elliptic genera serves as a non-trivial check, given that the two theories in Figure 4 have different Lagrangian descriptions. Regarding the work of Aharony-Razamat-Seiberg-Willett [arXiv:1611.02763], they studied the simplest example of an $\mathcal{N}=(0,2)$ theory with an adjoint chiral field—specifically, an $\mathcal{N}=(2,2)$ vector multiplet. They also used the elliptic genus to conjecture that the theory flows to a non-linear sigma model on $\mathbb{C}^{N-1}$. This is consistent with our claim in Section 3.6 of our paper, as well as with Equation (2.77) of [arXiv:2310.07965]. Given the structure of these 2d supersymmetric theories, we believe that the agreement of elliptic genera provides strong evidence supporting duality.

  6. In Footnote 1 on p.5, we clarify that all elliptic genus identities in the paper have not been rigorously proven. Instead, we have verified them up to rank-five JK residue integrals and $\mathcal{O}(q^2)$ terms via $q$-expansion. A similar statement is also made at the end of Appendix B. Since our paper contains numerous such identities, reiterating this point throughout the text would be redundant. To maintain clarity and focus, we believe that our current statements sufficiently address this issue.

  7. In Section 3.1 of the updated version (as well as in the first version), we aim to clarify why the (0,2) triality breaks down when moving from the U(N) gauge group to the SU(N) gauge group. As the referee also pointed out this issue, we recognize that there are various misleading statements in the literature regarding whether (0,2) triality holds for SU(N) gauge groups. However, the condition (3.3) for duality between the two SU gauge theories in Figure SU-dual is inconsistent with the triality condition. The failure of (0,2) triality for SU gauge theories can be attributed to the absence of the Fayet-Iliopoulos (FI) term, as explained in Section 3.1. We hope that this clarification helps resolve any misunderstandings in the literature.

  8. In the new triality presented in Figure AS-dual, the AS-2 theory involves an SU(2N) gauge theory with one anti-symmetric chiral and $2N+2$ fundamental chiral multiplets, rather than anti-fundamental chirals. The superpotential for this theory is now explicitly provided in Equations (3.12) and (3.13). If we were to replace these with $2N+2$ anti-fundamental chirals and adopt the superpotential suggested by the referee, the elliptic genus would acquire an additional factor of $N$, as explained in Equation (C.22). Consequently, for the duality to hold, we would need to consider the disjoint union of $N$ copies of the Landau-Ginzburg models.

9 & 10. We appreciate the referee’s insightful comments regarding the superpotential, which led us to identify new trialities in Figure AS-dual and Figure Sym-dual. In the revised version, we have explicitly provided the superpotential for each theory. We hope this revision sufficiently addresses the referee’s concerns.

Eleven. We clarify this point in the text above Equation (3.9). In this duality, the target space is a vector bundle over a Grassmannian, where the Grassmannian itself is compact. However, while the Grassmannian is compact, the total space of the determinant line bundle (as described in Equation (3.8)) is non-compact due to the presence of $N_1$ (anti-)fundamental chirals. Since these $N_1$ (anti-)fundamental chirals transform with the same charge under the $\mathrm{U}(1)_1$ flavor symmetry, their $R$-charges must all be set to zero. We hope this clarification resolves the concern.

Twelve. As suggested by the referee, we have included relevant references on brane constructions of 2d $\mathcal{N}=(0,2)$ theories in the Conclusion section (Section 4), including the works of Franco-Lee-Seong as well as lectures on integrable models.

We sincerely appreciate the referee’s feedback, which has helped us improve the manuscript further. We hope that our revisions address all concerns.

---

## Round 2 · Referee Report · Anonymous (Referee 1) · 2025-3-21

Report

The authors have significantly revised the manuscript so to address the comments from my previous report and the one of the other referee, and also to include numerous new results. Most changes look good to me, however since the manuscript has changed substantially I have some more comments.

  • I am also confused (like the other referee) by the superpotentials that the authors have added (upon request from the previous reports) to the 2d (0,2) dualities of section 3.

1) Compared to the first version of the manuscript, the authors have expanded the duality between $SU(2N)$ with one antisymmetric $X$ and $2N+2$ fundamental chirals $Y$ and the LG model of one $U(2N+2)$ antisymmetric chiral $A$ to a triality, by exploiting the fact that the latter theory is known to be dual also to the $Sp(2N)$ gauge theory with $2N+2$ chirals. However, for this known duality to hold one also have to add a Fermi field $\Psi$ to the LG theory which flips $\mathrm{Pf}\,A$, so such a deformation should also be turned on in the $SU(2N)$ gauge theory. In the first version, the authors identified $A$ with $YXY$, so shouldn't the superpotential (3.12) be of the form $\Psi\, \mathrm{Pf}(YXY)$?

2) In the SO duality on pages 14-15 there is now an additional Fermi singlet $\Psi$ appearing in the superpotential (3.22) compared to the first version of the manuscript. Which of the two versions is the correct one? I share the concern of the other referee on the superpotential (3.22), but if there was not such a Fermi field then there would be no need for a superpotential.

  • The logic in section 3.3 is a bit confusing and hard to follow. What is the purpose of going through the complicated dualizations of figure 6 if the goal is just to explain that the SU theory Higgses to the Sp theory when $x\to 1$ (I would write that $x\to 1$ is achieved by giving a VEV to X)? These dualizations would be useful to get to the Sp theory without studying explicitly the Higgsing, if somehow the duality in figure 7 was already known by some other means, but this doesn't seem to be the case. Also the last paragraph of this section 3.3 is quite cryptic. I would suggest to the authors to improve the presentation of this section and clarify its logic and purpose.

Recommendation

Ask for minor revision

  • validity: top
  • significance: top
  • originality: high
  • clarity: good
  • formatting: excellent
  • grammar: good

Author:  Satoshi Nawata  on 2025-04-08  [id 5355]

(in reply to Report 2 on 2025-03-21)

Above all, we sincerely thank the referee for the insightful and professional comments and suggestions.

Author:  Satoshi Nawata  on 2025-04-08  [id 5350]

(in reply to Report 2 on 2025-03-21)
Category:
answer to question
correction

1) In the AS-2 theory, the field $Y $ transforms in the fundamental representation of the $ SU(2N)$ gauge group, not in the anti-fundamental representation. Consequently, the operator $ YXY $ is not gauge invariant, since $ X $ transforms in the rank-two antisymmetric representation. This issue was initially pointed out by the other referee in the first round of review. The correct superpotential is now given in equation (3.12):

$$ W = \Psi\, \mathrm{Pf}(X)~. $$
The theory involving anti-fundamental chiral multiplets is instead discussed in Appendix C.4. In comparison with the AS-2 theory in Section 3.2, the elliptic genus receives an additional factor of $ \frac{n-2}{2} $, as shown in equation (C.20). We note that even in this theory, the operator $ \mathrm{Pf}(YXY) $ vanishes, since $\mathrm{rank}(YXY) \leq \mathrm{rank}(X) \leq n-2$. This point was raised by the other referee in their second round of comments.

2) For the triality shown in Figure Sym-dual to hold, it is necessary to include the Fermi multiplet $ \Psi $ with the corrected superpotential given in equation (3.21) in the SO gauge theory.

Regarding the Higgsing procedure, we have incorporated the referee’s suggestion into the revised version of Section 3.3. We believe the updated explanation now provides clarity and better aligns with the referee’s expectations.

---

## Round 2 · Referee Report · Bruno Le Floch (Referee 2) · 2025-3-21

Strengths

1- Many new two-dimensional dualities/trialities between non-abelian gauge theories with low (0,2) supersymmetry and various gauge groups including Sp, SO, SU, and Landau-Ginzburg models.

2- Derivation of these dualities from 4d N=1 and 4d N=2 dualities. Dimensional reduction is a robust way to anchor the dualities in the vast network of checks of higher-dimensional dualities.

3- Proposals for new dualities between (2,2) quiver theories.

Weaknesses

1- Superpotentials in (3.12), (3.22) seem to be incorrectly determined as certain Pfaffians or determinants simply vanish. See specific comments in the 'Changes' section of this report. If the hope was for these superpotentials to force various fugacities to be equal, then this hope breaks down. However, for some of these theories I don't see why the superpotentials would be needed in the first place.

2- Relatedly, the mechanism responsible for eliminating some fugacities, such as in (3.11), is not understood since the superpotential does not do this. There is an attempt around (3.33) for one such fine-tuning of fugacities (x→1) but it is not clear what triggers the Higgsing. Concretely, this means that symmetries of the putative dual theories are not shown to be the same in these cases. In principle this could be seen at the appropriate level in the index.

3- Since the analysis of dualities is centered around elliptic genera, which do not depend on moduli, it is impossible to specify the dictionary between 4d parameters (masses, gauge couplings) and 2d parameters (masses, R-charges, FI parameters, theta angles).

4- The paper could be clearer somewhere about the list of 4d dualities that are being dimensionally reduced. The current presentation makes it appear quite ad hoc.

Report

The authors approach of twisted reduction of 4d dualities down to 2d is sound, and I find particularly interesting their 2d (0,2) dualities and trialities derived from Seiberg duality. The paper advances significantly the state of knowledge of 2d dualities with low supersymmetry and constitutes good work.

Requested changes

1- Intro: 'of the duality frame' would be clearer than 'of the frame'

2- Section 2: regarding 'if these charges are non-negative integers', the situation (see the end of section 2 in ref [9]) seems to be that if all charges are non-negative then the 4d theory reduces cleanly to a single 2d theory. Under this condition it is confusing to talk about getting (1-r) chiral multiplets for r less than 1, since the only non-negative integer less than 1 is simply zero.

3- 'it becomes 2d' → 'it becomes a 2d'

4- Equation (2.2) has a stray word 'equivalent'

5- It's hard to understand when reading the text that between (2.8) and Figure 3 the authors turn to a new 2d theory. Maybe a more liberal use of the paragraph LaTeX commands would help clarify, or simply a clearer sentence at the start of the list of theories stating how the section 2 is organized?

6- In (2.11) and nearby, $U_2$ should be $U_N$? In (2.11) there are two $,V$ superscripts missing.

7- The identity (2.12) is an immediate consequence of gauging $U(1)$ in the identity (2.8) for $n=2$. There is probably a TQFT interpretation of all this.

8- The authors do not explain the condition (3.3) ($N_1\geq N_2+N_3$) imposed for the 2d (0,2) SU duality. A wrong explanation is given below (3.7): the truth is that the Grassmannians are defined whenever $N_1\geq|N_2-N_3|$, weaker than (3.3). I didn't locate in the literature an analysis of the IR behaviour of 2d (0,2) SU(n) SQCD theories, for instance ref [20] seems to focus on U(n), the authors may have better luck. Maybe, contrarily to U(n) theories which can flow to two different NLSMs depending on FI, the SU(n) theories can only flow to a single NLSM, at least in the regime $N_1\geq N_2+N_3$. Beyond this bound it may be that we get several NLSMs. For instance, in the left side of Figure SU-dual, going from the large~$N_1$ regime to the $N_2\geq N_1+N_3$ regime, the NLSM should go from giving vevs to the $N_1$ fundamental chirals to giving vevs to the $N_2$ antifundamental chirals. This whole discussion is probably beyond the scope of this work, unless some reference already does it.

9- When citing [21,22] write the words 'c-extremization' to orient the reader.

10- As far as I can tell, the Pfaffian of $A$ in (3.12) vanishes, because $A$ has rank at most 2. Indeed, in $\mathbb{C}^{2N+2}$, if the $2N$ vectors $Y_\alpha=(Y_\alpha^i)_{i=1,\dots,2N+2}$ for $\alpha=1,\dots 2N$ are not linearly independent, $A$ vanishes altogether, and otherwise we check that $A_{ij}Y_\alpha^i=0$ for all $\alpha$, hence $A$ is orthogonal to a $2N$-dimensional subspace of $\mathbb{C}^{2N+2}$. Maybe the correct superpotential mixes $X$ and $A$? Same comment for (C.7)--(C.8).

11- In (3.23) $A$ vanishes I think. The combination $\epsilon^{\alpha_1\dots\alpha_N} Y_{\alpha_1}^{j_1}\dots Y_{\alpha_N}^{j_N}$ vanishes because it is antisymmetrizing $N$ vectors $Y_\alpha\in\mathbb{R}^{N-2}$. In contrast, (3.25) is fine because $A$ is a projection of an $N\times N$ symmetric matrix~$Z$ to an $(N-2)\times(N-2)$ symmetric matrix, which can very well have full rank. But (C.18)--(C.19) seems to have a problem since ${\rm rank}(A)\leq{\rm rank}(X)\leq n-2$ so ${\rm Pf}(A)=0$. Same problem for (C.23)--(C.25).

12- I don't understand what the two paragraphs after (3.57) are doing there in particular. It seems we are in the middle of studying reductions of 4d ${\cal N}=4$, and suddenly we switch back to some comments about reductions of 4d ${\cal N}=2$ theories.

13- Comments around pages 27--30 about fields that contribute to the central charge but not to the elliptic genus seem very surprising. I think that the elliptic genus is a Jacobi form of weight zero and index related to the central charges, but maybe it is a combination of central charges that is not affected by these extra fields?

14- Below (B.2) it is stated that for a non-compact vacuum moduli space, the right-moving central charge is determined by the dimension, but this is wrong: in the product of a non-compact and compact theories, the central charge should simply add up, so c-extremization has to be done in the compact part of the theory. My understanding is that the non-compact chiral multiplets are forced to have R-charge zero, and then within the class of R-symmetries that respect this, one should do c-extremization, but I am unsure.

15- Just before section C.2 the reference to Fig Sym-dual should be AS-dual maybe?

16- In various places (in appendix C) 'one fundamental chirals' is incorrectly plural.

Recommendation

Ask for minor revision

  • validity: high
  • significance: good
  • originality: ok
  • clarity: good
  • formatting: excellent
  • grammar: excellent

Author:  Satoshi Nawata  on 2025-04-08  [id 5351]

(in reply to Report 1 by Bruno Le Floch on 2025-03-21)

We sincerely thank the referee for the thoughtful and constructive comments. We are especially grateful for pointing out the inaccuracies in our superpotentials and linear algebraic arguments. Below, we summarize the key changes and clarifications made in response to the referee's suggestions:

7- Below equations (2.12) and (2.14), we now briefly mention the underlying TQFT structure of the 2d $(2,2)$ theories. Since this structure is emphasized in reference [7], we have kept the explanation concise.

8- We agree with the referee’s observation. To the best of our knowledge, this point has not been addressed in the literature. We believe it warrants a more thorough study, and we leave a detailed analysis of the IR non-linear sigma model for future work.

10- We have corrected the superpotential in equation (3.12). As rightly pointed out by the referee, there is no consistent way to construct the previously proposed term involving the fundamental chiral multiplet $Y$. We have also corrected a similar issue in equation (C.17).

11- As the referee noted, the previous superpotentials in these examples were incorrect and the proposed terms would in fact vanish. The field $A$ can be constructed via the projection of $\Box \otimes \Box$ onto the trivial representation for the $SO$ gauge group. Accordingly, the superpotentials in equations (C.17) and (C.21) have been corrected.

12- We have moved the relevant comments to the conclusion in Section 4 for better flow and emphasis.

13- While these fields do contribute to the elliptic genus, they cancel each other. We have improved the phrasing below equation (3.62) to clarify this point. A similar cancellation occurs in the $(2,2)$ $SU(N)$ pure Yang-Mills theory, which is dual to $N-1$ $(2,2)$ twisted chiral multiplets, as discussed in the work of Aharony–Razamat–Seiberg–Willett. The central charges are $c_L = c_R = N - 1$, but the elliptic genus simplifies significantly due to cancellations between Fermi and chiral multiplets:

$$ \mathcal{I}(q, y) = \prod_{n=2}^N \frac{\theta(y^{1-n}; q)}{\theta(y^{-n}; q)} = \frac{\theta(y^{-1}; q)}{\theta(y^{-N}; q)}~. $$

14- Thank you for pointing out the imprecise explanation. We have revised the discussion below equation (B.2) accordingly. Furthermore, below equation (3.9), we added a comparison to the evaluation of the central charge in the context of $(0,2)$ triality for unitary gauge groups. In particular, we emphasize the necessity of performing $c$-extremization, even when the IR non-linear sigma model has a non-compact target space.

For the remaining comments and suggestions, we have made appropriate changes and improvements throughout the manuscript.

---

## Round 2 · Author Response

First of all, we would like to express our sincere gratitude for the valuable comments provided by the referees. In response to the feedback, we have made major revisions to the manuscript. In the first version, Section 3 focused on the duality of (0,2) SU gauge theories, while Section 4 addressed (0,2) Gauge/Landau-Ginzburg dualities. In the second version, we have combined these two sections into a single, unified section. As a result of further exploration, we have discovered additional new dualities, some of which extend to new trialities. Consequently, the length of the paper has approximately doubled.

---

## Round 2 · List of Changes

As noted above, the entire structure of the paper has been reorganized.

Section 2 has undergone a minor revision:
- In Section 2.3, we clarify that the term discussed is the scalar potential, not the superpotential. In an (\mathcal{N}=(2,2)) theory, the scalar potential is present even when there is no superpotential.
- Below equation (2.6) for the elliptic genus, we mention that the coordinate ring of the target space ((\mathrak{t} \times \mathrak{t}) / S_N) is generated by (\text{Tr}(\phi^i \sigma^j)). However, we are unclear about the precise relation between these generators, and thus do not fully understand the cancellation mechanism that leads to such a simple expression for the elliptic genus.

Section 3 has undergone a major revision with the addition of new dualities:
- Section 3.1 provides a more detailed discussion of the duality of SU gauge theories originally found by Gadde-Razamat-Willet [arXiv:1506.08795], addressing various subtleties.
- Section 3.2 generalizes the (0,2) Seiberg-like duality for the Sp gauge group into a new triality. In a similar vein, we propose a new triality involving SO gauge theory, SU+1Sym gauge theory, and a Landau-Ginzburg model. Based on these results, we uncover new (0,2) dualities in the following subsections:
- Section 3.3 proposes a new duality between Sp gauge theory and SU quiver gauge theory.
- Section 3.4 presents a duality between two SU+1AS gauge theories.
- Section 3.5 introduces a duality between two SU+1Sym gauge theories.
- Section 3.6 investigates SO and Sp gauge theories with adjoint chiral matter, exploring their dualities with free chiral theories.
- Section 3.7 explores theories that contain both symmetric and anti-symmetric chiral multiplets, presenting a duality to a Landau-Ginzburg model.

We have added Appendix C to provide additional technical details on the (0,2) dualities that are too specialized for the main text:
- Appendix C.1 explains the twisted compactification of the 4d (\mathcal{N}=1) Intriligator-Seiberg duality on (S^2), which leads to a duality between (0,2) SO gauge theory and a Landau-Ginzburg model.
- Appendix C.2 proposes a triality similar to the one discussed in Section 3.2.
- Appendices C.3 and C.4 cover dualities not included in Section 3.5.

---

## Round 3 · Referee Report · Bruno Le Floch (Referee 2) · 2025-4-28

Report
Recommendation
Publish (meets expectations and criteria for this Journal)

---

## Round 3 · Author Response

We extend our sincere gratitude to the referees for their meticulous and thoughtful feedback, as well as their time and care in evaluating our manuscript. Their expertise and constructive suggestions have profoundly strengthened the quality of this work.
In addressing the referees’ comments, we have implemented revisions to the manuscript, including:
- We have corrected the superpotentials of some (0,2) theories in Section 3.2 and Appendices C.2–C.4.
- We have also refined the discussion of Higgsing in Section 3.3 for clarity.
- Based on the referees' suggestions, we have incorporated further explanatory details and improved the overall presentation and phrasing.
We thank the referees once more for their invaluable guidance, which has been pivotal in elevating the level and clarity of our paper.

---

## Round 3 · List of Changes

We would like to thank the referees for their valuable feedback. In response to their comments and suggestions, we have made the following changes and improvements in the revised manuscript:
- Below equations (2.12) and (2.14), we now highlight and clarify the underlying TQFT structure of the 2d (2,2) theories.
- Below equation (3.9), we have added a discussion comparing our evaluation of the central charge to that of the (0,2) triality for unitary gauge groups. In particular, we emphasize the necessity of performing c-extremization for unitary gauge groups, even though the IR non-linear sigma model has a non-compact target space.
- We corrected the superpotential in equation (3.12), as one of the referees pointed out.
- Similarly, the superpotential in equation (3.21) has been corrected. In particular, we revised the construction of the field A in the Sym-1 theory in response to the referee’s comment.
- Per the referees’ suggestions, we improved the explanation of the Higgsing procedure in Section 3.3. Specifically, we clarified that the specialization (x \to 1) in the elliptic genus corresponds to giving a vacuum expectation value (VEV) to the chiral field (X) in the bottom theory of Figure 6. This leads directly to the duality depicted in Figure 7.
- In Section 4, we now provide a list of 4d theories and the corresponding 2d dualities discussed in the paper. We have also expanded the summary and added clarifying remarks in the conclusion.
- Below equation (B.2), we revised the explanation of the application of c-extremization.
- The superpotentials in equations (C.7), (C.17), and (C.21) have been corrected as per the referee's observation.
- In equations (C.19) and (C.23), we removed the field (B), which was not present in the rest of the text.

---

## Editorial Decision

published